# Decentralized Matrix Sensing:
# Statistical Guarantees and Fast Convergence

**Marie Maros**
School of Industrial Engineering
Purdue University
mmaros@purdue.edu

**Gesualdo Scutari**
School of Industrial Engineering
Purdue University
gscutari@purdue.edu

## Abstract

We explore the matrix sensing problem from near-isotropic linear measurements, distributed across a network of agents modeled as an undirected graph, with no server. We provide the first study of statistical, computational/communication guarantees for a decentralized gradient algorithm that solves the (nonconvex) Burer-Monteiro type decomposition associated to the low-rank matrix estimation. With small random initialization, the algorithm displays an approximate two-phase convergence: (i) a *spectral phase* that aligns the iterates' column space with the underlying low-rank matrix, mimicking centralized spectral initialization (not directly implementable over networks); and (ii) a *local refinement phase* that diverts the iterates from certain degenerate saddle points, while ensuring swift convergence to the underlying low-rank matrix. Central to our analysis is a novel "in-network" Restricted Isometry Property which accommodates for the decentralized nature of the optimization, revealing an intriguing interplay between sample complexity, network connectivity & topology, and communication complexity.

## 1 Introduction

Matrix sensing–the estimation of a low-rank matrix from a set of linear measurements–finds applications in diverse fields such as image reconstruction (e.g., [48, 29]), object detection (e.g., [33, 52]) and array processing (e.g., [21]), to name a few. It also serves as a benchmark for determining the statistical and computational guarantees achievable in deep learning theory, since it retains many of the key phenomena in deep learning while being simpler to analyze. Despite significant progress in understanding the convergence and generalization properties of various solution methods for training such learning models, a majority of these advances focus on a centralized paradigm, aggregating data at a central location with vast computing resources–good tutorials on the topic include [7, 4]. This centralized approach, however, is increasingly unsuitable for modern applications due to server bottlenecks, inefficient communication, and power usage. Therefore, the development of statistical learning methods for massively decentralized networks without servers is timely and crucial.

This paper tackles the matrix sensing problem from data distributed over networks. We contemplate a network of $m$ agents modeled as an undirected graph with no servers, where agents can communicate with their immediate neighbors–these architectures are also known as *mesh* networks. The collective objective is to estimate a ground-truth matrix $\bar{Z}^\star \in \mathbb{R}^{d \times d}$, based on $N = m \cdot n$ total observations $y_1, \ldots, y_N$, equally split into $n$-sized, disjoint datasets $\mathcal{D}_1, \mathcal{D}_2, \ldots, \mathcal{D}_m$. Each agent's signal model is thus given by

$$y_j = \langle A_j, \bar{Z}^\star \rangle := \texttt{trace}(A_j \bar{Z}^\star), \quad \text{for } j \in \mathcal{D}_i. \tag{1}$$

Here, $A_j \in \mathbb{R}^{d \times d}$, with $j \in \mathcal{D}_i$, are the known symmetric measurement matrices to agent $i$; $\bar{Z}^\star$ is assumed symmetric, positive semidefinite, and low-rank, i.e., $r^\star := \texttt{rank}(\bar{Z}^\star) << d$.

37th Conference on Neural Information Processing Systems (NeurIPS 2023).

To minimize communication overhead and avoid the need for $d \times d$ matrix transmission, we employ the Burer-Monteiro-type decomposition of the estimate $\bar{Z}$ of $\bar{Z}^\star$, that is, $\bar{Z} = \bar{U}\bar{U}^\top$, with $\bar{U} \in \mathbb{R}^{d \times r}$, and seek to minimize the squared loss $F(\bar{U})$, defined as

$$\min_{\bar{U} \in \mathbb{R}^{d \times r}} F(\bar{U}) := \frac{1}{m} \sum_{i=1}^{m} \underbrace{\frac{1}{4n} \sum_{j \in \mathcal{D}_i} \left( y_j - \langle A_j, \bar{U}\bar{U}^\top \rangle \right)^2}_{:= f_i(\bar{U})}, \tag{2}$$

where $f_i(\bar{U})$ is the loss function of agent $i$. Ideally, the number $r$ of columns of $\bar{U}$ should be set to $r^\star$. However, $r^\star$ might not be known in advance. In this study, we consider the so-called *over-parameterized* regime where $r \geq r^\star$.

The formulation (2) poses multiple challenges. Firstly, $F$ is nonconvex, lacks global smoothness (i.e., global Lipschitz continuity of $\nabla F$), and is not entirely known to the agents. Secondly, the over-parameterized regime may intuitively suggest a risk of overfitting. However, recent studies (e.g., [34, 20]) have compellingly revealed that when *centralized* Gradient Descent (GD) is applied to (2) with a small random initialization, it induces an implicit bias towards simpler solutions with favorable generalization properties. This bias–often referred to as the *simplicity bias* or the *incremental learning* behavior of GD(/Stochastic Gradient Descent)–assists in the exact or approximate recovery of the ground truth $\bar{Z}^\star$. Interestingly, this phenomenon serves as a hidden mechanism in various other (deep) learning tasks that mitigates overfitting in highly over-parameterized models (e.g., [13, 30, 31]). However, the direct implementation of GD in mesh networks is not feasible due to the lack of access to $\nabla F$ by the agents or of a server collecting agents' gradients $\nabla f_i$.

The goal of this paper is to uncover the possible simplicity bias of a *decentralized* instance of GD, solving the matrix sensing problem (2) over mesh networks. To the best of our knowledge, this study is unique in the realm of decentralized optimization, establishing the first sample, convergence rate, and generalization guarantees of a decentralized gradient-based algorithm tailored for matrix sensing over mesh networks. We delve into the relevant existing literature in the subsequent section.

## 1.1 Related works

The literature offers numerous decentralized algorithms which could in principle be used to tackle the matrix sensing problem, directly or indirectly. However, the accompanying statistical and computational guarantees fall short, being either non-existent or inadequate, as we elaborate next.

• **Off-the-shelf decentralized algorithms for nonconvex problems:** The matrix sensing problem (2) naturally invites the application of decentralized algorithms specifically designed for nonconvex losses in the form $F = (1/m) \sum_{i=1}^{m} f_i$ (summation of agent functions). Noteworthy examples of such algorithms include **(i)** decentralizations of the GD that merge local gradient updates with (push-sum) consensus algorithms [47, 2, 37], **(ii)** decentralized first-order methods employing gradient tracking strategies [8, 32, 42, 17], and **(iii)** decentralized algorithms grounded on primal-dual decomposition or penalization of lifted reformulations incorporating explicitly consensus constraints [15, 14, 50]. However, despite their initial appeal, when applied to (2), these algorithms either lack of any convergence guarantee–the requirement that $F$ is *globally smooth* [47, 2, 37, 8, 32, 42, 15, 14, 50] and has a (uniformly) *bounded* gradient [47, 2, 37, 8, 42] is not met by the matrix sensing loss in (2)–or they converge at sublinear rate to *some* critical points of $F$ (which may not be the global minimizers), whose generalization properties remain unexplored and obscure [17].

• **Ad-hoc decentralized algorithms for some matrix recovery problems:** This line of works comprises decentralized schemes designed *specifically* for the *structured* matrix-related optimization problem under consideration. Relevant examples are briefly highligthed next.
**(i) Dictionary learning & matrix factorization problems [5, 51, 49]:** In [5], convergence of a decentralized gradient tracking method for certain dictionary learning problems is established; a similar problem class is further investigated in [51], where generalization properties of a penalized consensus algorithm are studied, albeit without a convergence rate analysis. Despite their differences, these studies share a common premise of a *full* observation model, leading to a loss in the form of $F(\bar{U}\bar{V}^\top) = \|Y - \bar{U}\bar{V}^\top\|^2$. Here, $Y = \bar{Z}^\star + N$ is the data matrix with $N$ denoting noise. This full observation model contrasts with the matrix sensing model in (2), which is based on *partial* (noiseless) measurements. Lastly, [49] proposes a distributed Frank-Wolfe algorithm to address a low-rank matrix factorization problem, formulated as a trace (nuclear) norm *convex* minimization

problem (the nonconvex rank constraint is substituted by a nuclear norm constraint).

**(ii) Distributed spectral methods [45, 19, 11, 12, 10, 44, 43]:** Spectral methods have been established as effective strategies for obtaining reliable estimates of leading eigenvectors of a specified data matrix, as well as for providing a promising "warm start" for numerous iterative nonconvex matrix factorization algorithms [4, 7]. Recent developments [19, 11, 12, 10, 44, 43] have successfully extended spectral methods–particularly principal component analysis–to decentralized contexts, achieving linear convergence rates, communications per iteration on the order of $\mathcal{O}(dr)$, and precise recovery up to a desired accuracy. A good tutorial on this subject can be found in [45]. These methods, in principle, can tackle the decentralized matrix sensing problem as formulated in this work through the estimation of the leading eigenspace of the surrogate matrix $Y = \sum_{i=1}^{m} \sum_{j \in \mathcal{D}_i} y_j A_j$, where $\sum_{j \in \mathcal{D}_i} y_j A_j$ is held by agent $i$. In fact, under suitable RIP on the linear mapping associated with $Y$, one has $(1/N) \sum_{i=1}^{m} \sum_{j \in \mathcal{D}_i} y_j A_j \approx \bar{Z}^\star$ [34]. While such an approach can yield valuable insights about the ground truth $\bar{Z}^\star$, *exact* recovery of $\bar{Z}^\star$ to arbitrary precision cannot be guaranteed [38]. This limitation starkly contrasts with the robust guarantees attainable by the gradient algorithm applied to the centralized matrix sensing problem with small random initialization (e.g., [34, 20]).

• **Generic saddle-escaping decentralized algorithms:** Under a sufficiently small RIP constant of the linear mapping associated with the signal model (1), the matrix sensing loss in (2) is shown to have no spurious local minima and all strict saddle points [1, 22]. Consequently, the task becomes escaping strict saddle points and computing second-order critical points. In the distributed optimization context, recent works studied the escape properties of several decentralized algorithms. Early works showed that certain decentralized schemes–the deterministic DGD [6, 18], the subgradient-flow [36], gradient-tracking algorithms [6], and primal-dual based methods [16, 24]–with random initialization, converge *asymptotically* towards a second-order critical point of a smooth function (subject to mild regularity conditions), with high probability. However, this near-certain convergence does not necessarily imply fast convergence. There exist non-pathological functions for which randomly initialized GD requires exponential time (in the ambient dimension) to escape saddle points [9]. It remains uncertain whether the inherent structure of the matrix sensing problem could yield superior convergence guarantees. Subsequent research has investigated the impact of decaying, additive noise perturbation on the agents' gradients of (stochastic) DGD in the Adapt-then-Combine (ATC) form [40, 39, 41]. While convergence to approximately second-order stationary points is assured within a polynomial number of iterations, the prerequisite that the loss has a *globally* Lipschitz gradient and Hessian matrix is not met by the matrix sensing loss in (2). This leaves decentralized saddle-escaping methods bereft of convergence rate *and* generalization guarantees when applied to (2).

## 1.2  Major contributions

We establish the first *convergence rate and generalization guarantees* of a *decentralized* gradient algorithm solving the matrix sensing problem via (2) over mesh networks. We borrow the following decentralized gradient descent [25],[46]: for each agent $i = 1, \ldots, m$,

$$\bar{U}_i^{t+1} = \sum_{j=1}^{m} w_{ij} \bar{U}_j^{t+1/2} \quad \text{and} \quad \bar{U}_i^{t+1/2} = \sum_{j=1}^{m} w_{ij} \bar{U}_j^t - \alpha \nabla f_i \left( \sum_{j=1}^{m} w_{ij} \bar{U}_j^t \right), \qquad (3)$$

Here, $\bar{U}_i^t$ is an estimate at iteration $t$ of the optimization, common matrix $\bar{U}$ in (2) held by agent $i$; $\alpha \in (0, 1]$ is the stepsize; and $w_{ij}$'s are appropriately chosen nonnegative weights. We have $w_{ii} > 0$, $i = 1, \ldots, m$, and $w_{ij} > 0$ if agents $i$ and $j$, $i \neq j$, can communicate; otherwise $w_{ij} = 0$. The algorithm employs two communication steps/iteration, aiming to enforce an agreement on both iterates $\bar{U}_i^t$ and local gradients $\nabla f_i$. One could reduce the communication steps to *one* per iteration via a suitable variable change, resulting in the DGD-ATC form [46]. However, for the sake of clarity and ease of analysis, we opt to keep the form in (3), without any loss of generality.

• **Guarantees:** Our study presents a thorough statistical and convergence analysis of (3), yielding the following key insights. **(i)** *Convergence to low-rank solutions*: We demonstrate that, regardless the degree of overparametrization $r$, the iterates generated by (3) from a small random initialization converge towards low-rank solutions. We also provide an estimate of the worst-case iteration complexity. Improving results of the GD in a centralized setting (e.g., [34]), we specify an *entire* interval for algorithm termination, within which the generalization error is guaranteed to remain below the desired accuracy. This interval expands as the initialization becomes smaller. **(ii)** *Two-phase convergence:* Our analysis reveals a two-phase convergence behavior of (3). The initial

"spectral" phase sees the iterates mimic a theoretical centralized power method with full data access, while the subsequent "refinement" phase steers the trajectory towards the ground-truth solution. To the best of our knowledge, this provides the first evidence of a *simplicity* bias in a decentralized algorithm, aligning with the observed behavior of centralized GD [34]. **(iii)** *Generalization error and communication complexity:* The generalization error is shown to scale polynomially with the initialization size while the communication complexity scales logarithmically. Consequently, one can achieve arbitrarily small estimation errors with only a modest increase in communication and computation cost. **(iv)** *RIP and network connectivity*: Our findings hold under the conditions that the centralized measurement operator satisfies the standard RIP, and that network connectivity is sufficiently small. The former condition implies that our algorithm operates under the *same* sample complexity requested in the centralized setting. The latter is distinctive of the decentralized settings and shown to be unavoidable. **(v)** *Almost performance invariance with the network size*: We demonstrate that with an increase in the network size, the generalization error maintains its consistency whereas the communication cost grows logarithmically. Thus, the algorithm ensures effective error control with a marginal increase in communication overhead as the network expands.

• **Convergence analysis:** Although our analysis draws some insights from [34], the proof techniques employed diverge from those used therein. The decentralized nature of our setting introduces additional error terms, thereby making the analysis substantially more complex. Our methodology hinges on a newly introduced concept of RIP, termed *in-network* RIP. This concept harnesses the RIP of the measurement operator, much like the centralized GD, and intertwines it with the network's connectivity to derive favorable attributes of the new, overarching network-wide measurement operator. Furthermore, it reveals the interplay between sample complexity, network connectivity & topology, and communication complexity towards achieving statistical and computational guarantees over networks. Although we have defined the in-network RIP in the context of our specific algorithm dynamics, we posit that it possesses independent significance and could potentially pave the way for performance analysis of other distributed schemes.

## 2 Preliminaries

In this section, we first list the notations used in the paper, and then provide details of our theoretical setup and necessary preliminary results.

### 2.1 Notations

For any positive integer $m$, we define $[m] \triangleq \{1, \ldots, m\}$; $1_m$ is the $m$-dimensional vector of all ones; $I_d$ is the $d \times d$ identity matrix; $\otimes$ denotes the Kronecker product; and $\texttt{range}(M)$ (resp. $\texttt{rank}(M)$) denotes the range space (resp. rank) of the matrix $M$. When considering a matrix $M \in \mathbb{R}^{md \times md}$, partitioned into blocks of size $d \times d$, we will denote the block at the $i$-th row and $j$-th column as $[M]_{ij}$, for $i, j \in [m]$. Here, $i$ and $j$ indices increment by $d$, reflecting the size of the blocks.

We use $\|\cdot\|$ to denote the Euclidean norm. When applied to matrices, $\|\cdot\|$ is the operator norm induced by $\|\cdot\|$, and $\|\cdot\|_F$ denotes the Frobenius norm of the argument matrix. We order the eigenvalues of any symmetric matrix $M \in \mathbb{R}^{d \times d}$ in nonincreasing fashion, i.e., $\lambda_1(M) \geq \ldots \geq \lambda_d(M)$. The singular values of (a rectangular) matrix $M$ of rank $r$ are denoted as $\sigma_1(M) \geq \sigma_2(M) \geq \cdots \sigma_r(M) > 0$.

*Truncated SVD:* For any given matrix $M \in \mathbb{R}^{d_1 \times d_2}$, with rank $r^\star > 0$, we write the truncated SVD as $M = V_M \Lambda_M Q_M^\top$, where $V_M \in \mathbb{R}^{d_1 \times r^\star}$ and $Q_M \in \mathbb{R}^{d_2 \times r^\star}$ satisfy $V_M^\top V_M = I_{d_1}$ and $Q_M^\top Q_M = I_{d_2}$, and $\Lambda_M \in \mathbb{R}^{r^\star \times r^\star}$ is a diagonal matrix.

*Augmented matrices:* It is convenient to introduce the following "augmented" matrices suitable to rewrite the decentralized algorithm (3) in a concise block-stacked form:

$$\mathcal{W} := W \otimes I_d, \quad \mathcal{J} := (1/m)1_m 1_m^\top \otimes I_d. \tag{4}$$

where $W \in \mathbb{R}^{m \times m}$ is the matrix of the gossip weights in (3), defined as $[W]_{ij} = w_{ij}$.

### 2.2 Basic definitions and assumptions

We develop our theoretical analysis under the following standard assumptions on the matrix sensing problem, algorithm parameters, and network connectivity.

• **On the matrix sensing problem:** Given the signal model (1), we decompose the ground-truth matrix as $\bar{Z}^\star = \bar{X}\bar{X}^\top$, for some $\bar{X} \in \mathbb{R}^{d \times r^\star}$ (recall, $r^\star$ is the rank of $\bar{Z}^\star$).

**Definition 1** (condition number). *We define the condition number of $\bar{X} \in \mathbb{R}^{d \times r^\star}$ as $\kappa = \frac{\|\bar{X}\|}{\sigma_{r^\star}(\bar{X})}$.*

We associate to the signal model (1) the measurement linear operator $\mathbb{R}^{d \times d} \ni \bar{Z} \mapsto \bar{\mathcal{A}}(\bar{Z}) \in \mathbb{R}^N$ and its adjoint $\mathbb{R}^N \ni w \mapsto \bar{\mathcal{A}}^*(w) \in \mathbb{R}^{d \times d}$, defined as

$$\bar{\mathcal{A}}(\bar{Z}) := \frac{1}{\sqrt{N}} \left( \langle A_j, \bar{Z} \rangle \right)_{j \in \mathcal{D}_i, i \in [m]} \quad \text{and} \quad \bar{\mathcal{A}}^*(w) = \frac{1}{\sqrt{N}} \sum_{i=1}^{m} \sum_{j \in \mathcal{D}_i} w_j A_j. \tag{5}$$

A standard assumption in the matrix sensing literature is requiring the RIP for the operator $\bar{\mathcal{A}}$.

**Definition 2** (RIP). *The measurement operator $\bar{\mathcal{A}} : \mathbb{R}^{d \times d} \to \mathbb{R}^N$ satisfies the $(\delta, r)$-RIP condition if*

$$(1 - \delta)\|\bar{Z}\|_F^2 \leq \|\bar{\mathcal{A}}(\bar{Z})\|^2 \leq (1 + \delta)\|\bar{Z}\|_F^2, \tag{6}$$

*for all matrices $\bar{Z} \in \mathbb{R}^{d \times d}$ with $rank(\bar{Z}) \leq r$.*

The RIP condition is the key to ensure the ground truth $\bar{Z}^\star$ to be recoverable with partial observations. In fact, an important consequence of RIP is that $\bar{\mathcal{A}}^*\bar{\mathcal{A}}(\bar{Z}) = (1/N) \sum_{i=1}^{m} \sum_{j \in \mathcal{D}_j} \langle A_j, \bar{Z} \rangle A_j \approx \bar{Z}$, for all $\bar{Z}$ low-rank (see, e.g., [34]). Notice that when all entries of the matrices $A_j$ are drawn i.i.d. with distribution $\mathcal{N}(0, 1)$ on the off-diagonal entries and distribution $\mathcal{N}(0, 1/\sqrt{2})$ on the diagonal, the $(\delta, r)$-RIP holds with high probability, if the number of observations $N = \Omega(dr/\delta^2)$ (e.g., [3]).

• **Network setup and gossip matrices:** Agents are embedded in a communication network, modelled as an undirected graph $\mathcal{G} = \{\mathcal{V}, \mathcal{E}\}$, where the vertices $\mathcal{V} = [m] \triangleq \{1, \ldots, m\}$ correspond to the agents and $\mathcal{E}$ is the set of edges of the graph; $(i, j) \in \mathcal{E}$ if and only if there is a communication link between agents $i$ and $j$. We study the decentralized algorithm (3) using gossip weight matrices satisfying the following standard assumption in the literature of distributed optimization.

**Assumption 1.** $W = [w_{ij}]_{ij=1}^{m}$ *satisfies: **(i)** $w_{ij} > 0$, if $(i, j) \in \mathcal{E}$; otherwise $w_{ij} = 0$; furthermore, $w_{ii} > 0$, for all $i \in [m]$; **(ii)** $W = W^\top$ and $W1 = 1$ (stochastic); **(iii)** $W$ is positive semidefinite; and **(iv)** there holds $\rho \triangleq \|W - 1_m 1_m^\top/m\|_2 < 1$.*

Assumption 1 is standard in the literature of distributed algorithms and is satisfied by several weight matrices; see, e.g., [26]. Note that $\rho < 1$ holds true by construction for connected graphs. Roughly speaking, $\rho$ measures how fast the network mixes information; the smaller $\rho$, the faster the mixing.

## 2.3 Augmented mapping and in-network RIP

Fundamental to our analysis is a novel RIP-like property associated with an augmented linear mapping tied to the decentralized algorithm (3). This new property effectively captures the admissible "degree of distortion" on the signal information $\bar{Z}^\star$, taking into account both the partial observability of $\bar{Z}^\star$ as postulated in (1) (through the measurement operator $\bar{\mathcal{A}}$ defined in (5)) and the intricacies of the in-network optimization process (regulated by the network operator $\mathcal{W}$, as defined in (4)).

We begin rewriting the decentralized algorithm (3) in a compact form. To do so, we define the following quantities: (i) the stacked block matrices $U^t \in \mathbb{R}^{md \times r}$ and $Z^\star \in \mathbb{R}^{md \times md}$:

$$U^t := (\bar{U}_i^t)_{i \in [m]} \quad \text{and} \quad Z^\star := 1_m 1_m^\top \otimes \bar{Z}^\star, \tag{7}$$

respectively; (ii) the augmented mapping $\mathcal{A} : \mathbb{R}^{md \times md} \to \mathbb{R}^N$ and its adjoint $\mathcal{A}^* : \mathbb{R}^N \to \mathbb{R}^{md \times md}$:

$$[\mathcal{A}(Z)]_\ell \triangleq \frac{1}{\sqrt{mn}} \left\langle m \left( w_{\mathcal{V}(\ell)} w_{\mathcal{V}(\ell)}^\top \right) \otimes A_\ell, Z \right\rangle, \quad \mathcal{A}^*(q) \triangleq \sum_{\ell=1}^{N} \frac{q_\ell}{\sqrt{mn}} (m w_{\mathcal{V}(\ell)} w_{\mathcal{V}(\ell)}^\top) \otimes A_\ell, \tag{8}$$

where $\mathcal{V}(\ell) : \cup_{i=1}^{m} \mathcal{D}_i \to \mathcal{V}$ returns the index $i$ such that $\ell \in \mathcal{D}_i$, $\ell \in [N]$. Note that these operators depend on both data measures (via $\bar{\mathcal{A}}$) and the network (via $\mathcal{W}$). Using the definitions in (4), (7), and (8), it is not difficult to check that (3) can be rewritten equivalently as:

$$U^{t+1} = \left( \mathcal{W}^2 + \frac{\alpha}{m} \mathcal{A}^* \mathcal{A}(Z^\star - U^t(U^t)^\top) \right) U^t. \tag{9}$$

For the convergence of (9) towards low-rank matrices with strong generalization properties, we anticipate certain conditions to be imposed on the operator $\mathcal{A}$. The algorithmic mapping structure in (9) provides some insights in this regard. Since $\mathcal{W}^2\mathcal{J} = \mathcal{J}$ (due to Assumption 1(ii)), the linear network operator $\mathcal{W}^2$ effectively functions as the identity map on matrices $Z \in \mathbb{R}^{dm \times dm}$ with range$(Z) \subset$ range$(\mathcal{J})$. This is in particular true for ($d \times d$ block) consensual matrices $Z = \mathcal{J}Z\mathcal{J}$, including the (augmented) ground-truth $Z^\star$, as defined in (7). This implies that to accomplish precise reconstructions of $Z^\star$, the operator $\mathcal{A}$ ought to exhibit some RIP-like regularity. Postponing to Sec. 3.1 a more rigorous and comprehensive argument, we claim that the following property suffices.

**Definition 3** (In-network RIP). *The operator $\mathcal{A} : \mathbb{R}^{md \times md} \to \mathbb{R}^N$ defined in (8) satisfies the in-network $(\delta, r)-$RIP property with tolerance $\Delta \geq 0$, if*

$$(1 - \delta)\|\mathcal{J}Z\mathcal{J}\|_F^2 - \Delta\|Z - \mathcal{J}Z\mathcal{J}\|_F^2 \leq \|\mathcal{A}(Z)\|_2^2 \leq (1 + \delta)\|\mathcal{J}Z\mathcal{J}\|_F^2 + \Delta\|Z - \mathcal{J}Z\mathcal{J}\|_F^2, \quad (10)$$

*for any matrix $Z \in \mathbb{R}^{md \times md}$ such that each of its $d \times d$ blocks $[Z]_{i,j}$ and its block-average $\bar{Z} = \frac{1}{m^2}\sum_{i=1}^m \sum_{j=1}^m [Z]_{i,j}$ are of rank at most $r$.*

The condition (10) reads as an "exact" RIP property of $\mathcal{A}$ along (block) consensual directions, allowing for some "perturbation" in the form of consensus errors $\|Z - \mathcal{J}Z\mathcal{J}\|_F^2$. The following result shows that the tolerance error can be controlled by the network connectivity $\rho$ (see Assumption 1(iv)).

**Lemma 1.** *Suppose $\bar{\mathcal{A}}$ satisfies the $(\delta_{2r}, 2r)$-RIP, and the gossip matrix $W$ is chosen according to Assumption 1. Then, the augmented operator $\mathcal{A}$ satisfies the in-network $(2\delta_{2r}, r)$-RIP with tolerance*

$$\Delta = \rho^2 \cdot \frac{4m^5(1 + 2\delta_{2r})}{\delta_{2r}}(1 + \delta_{2r}). \quad (11)$$

Clearly, the smaller $\rho$, the smaller $\Delta$, revealing an unexplored interplay between (potential) generalization properties and network characteristics. Since small $\rho$ can be enforced also by employing multiple rounds of communications per iteration (see Sec. 3 for details), the communication complexity enters in the tradeoff equation. Our theory in the next section will quantify this interplay, revealing conditions and tuning recommendations to achieve fast convergence and strong generalization properties.

## 3 Main Results

We are ready to state the convergence results of Algorithm (3). Here we consider the overparametrized case $r \geq 2r^\star$ while the other ranges of $r$ are discussed in the supplementary material.

**Theorem 1.** *Consider the matrix sensing problem (1), with augmented ground-truth $Z^\star$, under $r \geq 2r^\star$, and the measurement operator $\bar{\mathcal{A}}$ satisfying the $(4(r^\star + 1), \delta)-$RIP, with $\delta \precsim \kappa^{-4}(r^\star)^{-1/2}$. Let $\{U^t\}_t$ be the (augmented) sequence generated by Algorithm (3), under the following tuning: (i) the stepsize $\alpha \precsim \kappa^{-4}\|\bar{X}\|^{-2}$; (ii) the gossip matrix $W$ is chosen to satisfy Assumption 1, with*

$$\rho \precsim \frac{\delta^2}{m^6\kappa^4 r^\star}; \quad (12)$$

*and (iii) the initialization $U^0$ is chosen as $U^0 = \mu U$, where $U \in \mathbb{R}^{md \times r}$ has i.i.d. $\mathcal{N}(0, \sqrt{m/r})$ distributed entries, and $\mu$ satisfies*

$$\mu^2 \precsim \min\left\{\frac{\sqrt{rm}}{d\sqrt{d}\kappa^9}, \frac{\sqrt{r}}{d\sqrt{d}}\left(\kappa^2\sqrt{\frac{d}{r}}\right)^{-96\kappa^2}\right\}. \quad (13)$$

*Then, after*

$$\hat{t} \precsim \frac{1}{\alpha\,\sigma_{r^\star}^2(\bar{X})}\left(\ln\left(\kappa^2\sqrt{\frac{d}{r}}\right) + \ln\left(\frac{\sigma_{r^\star}(\bar{X})}{\mu}\right) + \ln\left(\max\left\{1, \kappa\frac{r^\star}{r - r^\star}\right\}\frac{\|\bar{X}\|}{\mu}\right)\right) \quad (14)$$

*iterations, there holds*

$$\frac{\|U^{\hat{t}}(U^{\hat{t}})^\top - Z^\star\|_F}{\|Z^\star\|} \precsim \left((r - r^\star)^{7/8}(r^\star)^{1/8}\|\bar{X}\|^{-21/16}\mu^{21/16}\left(\kappa^2\frac{d}{r}\right)^{21/16}\right), \quad (15)$$

*with probability at least $1 - c_1 e^{-c_2 r}$, where $c_1, c_2 > 0$ are universal constants.*

- *Statistical guarantees:* (15) demonstrates that, in the setting above, the iterates $U^t(U^t)^\top$ converge to an estimate of the low-rank solution $Z^\star$ within a precision that can be made arbitrarily small by reducing the size $\mu$ of the random initialization. The test error's dependence on $\mu$ is polynomial, whereas the worst-case convergence time only increases logarithmically with $\mu$ (see (14)), indicating that significant test error reductions can be achieved with moderate increases in communication and local computations. These guarantees are established under the RIP of the measurement operator $\bar{\mathcal{A}}$, and thus operate under the same sample complexity as the centralized setting. For instance, for Gaussian measurement matrices, $N \gtrsim d(r^\star)^2\kappa^8$. While the dependence on $d$ is optimal, the scaling on $(r^\star)^2$ and $\kappa^8$ is less favorable compared to convex approaches based on nuclear norm minimization [3]. However, decentralized methods solving such formulations directly (e.g., [23]) would entail a communication cost of $\mathcal{O}(d^2)$, which is significantly less favorable than the $\mathcal{O}(rd)$ of Algorithm 3.

- *On the number of iterations:* Interestingly, the worst-case iteration complexity aligns with what observed for the GD in the centralized setting, following thus the same interpretation [34]. The first term in (14) represents the duration of the *spectral alignment* phase: beginning from a small initialization, the iterates $U^t(U^t)^\top$ progressively align with the $r^\star$ leading eigenvectors of the mapping $\mathcal{W}^2 + \alpha/m\mathcal{A}^*\mathcal{A}(Z^\star)$. Under the in-network RIP (Lemma 1), which requires the RIP of $\bar{\mathcal{A}}$ and a sufficiently small $\rho$, we establish that this operator approximates the mapping of the power method applied to $Z^\star$ (see the sketch of the proof in Sec. 3.1). The remaining two terms in (14) represent the duration of the subsequent *refinement* phase. This phase steers the iterates away from certain degenerate saddle points while ensuring convergence towards the low-rank matrix $Z^\star$.

In line with the findings for centralized GD [34], the test accuracy achieved at time $\hat{t}$ might not persist for larger iterations. The corollary below refines these results by establishing a nonempty time interval within which the estimation error is guaranteed to stay within the desired accuracy.

**Corollary 1.** *Under the conditions of Theorem 1, it holds*

$$\frac{\|U^t(U^t)^\top - Z^\star\|_F}{\|Z^\star\|} \precsim (r - r^\star)^{7/8}(r^\star)^{1/8}\mu^{1/8}\|\bar{X}\|^{-21/16}\left(\kappa^2\frac{d}{r}\right)^{1/8}, \tag{16}$$

*for any $t \in [\hat{t}, T]$, with*

$$T - \hat{t} \gtrsim \frac{1 - \frac{d\kappa^2\mu}{r}}{\alpha\left(\kappa^2\sigma^2\mu^{1/8}\right)} \quad and \quad \sigma := c_3(r^\star)^{1/8}(r - r^\star)^{7/8}\|\bar{X}\|^{11/16}, \tag{17}$$

*where $c_3 > 0$ is an universal constant.*

The corollary ensures that the test error remains proportional to $\mu^{1/8}$ throughout the interval $T - \hat{t}$. Notably, the duration of this interval increases as $\mu$ approaches zero. This result is quite desirable, especially in distributed settings where coordinating termination at a specific time may be challenging.

- *On the condition (12) on $\rho$ and network scalability:* The stipulation on $\rho$ signifies the need of a well-connected network—the larger the network size $m$ or the condition number $\kappa$ of the ground truth, the smaller $\rho$. This is a non-negotiable condition essential for managing consensus errors through the tolerance $\Delta$, thus ensuring an adequate in-network RIP for the algorithm operator $\mathcal{A}$. When coupled with the RIP of the measurement operator $\bar{\mathcal{A}}$, it suffices for a sufficient alignment of the iterates $U^t(U^t)^\top$ with the signal subspace from the early stages of the algorithm. Our numerical experiments (see Sec. 4) indeed demonstrate that maintaining such a constraint on $\rho$ is indispensable for securing convergence and favorable estimation errors. When the network graph is predetermined (with given $W$), one can meet the condition (12) (if not a-priori satisfied) by employing at each agent's side multiple rounds of communications per gradient evaluation. This is a common practice [27] that in our case results in a communication overhead that is only logarithmic in $m$ and $\kappa$.

Notice that the generalization error (see (15) and (16)) is independent of $\rho$ or $m$. This demonstrates that the algorithm's performance scales favorably with $m$. As $m$ increases, the generalization error remains unchanged, whereas the communication cost grows only modestly (logarithmically with $m$).

## 3.1 Sketch of the proof of Theorem 1

This section provides some insights on the proof of the theorem, highlighting the challenges and the differences with existing centralized and decentralized techniques.

The goal is to establish that $U^t(U^t)^\top \approx Z^\star$ as the algorithm progresses. Following [34], we decompose the iterates as

$$U^t = \underbrace{U^t Q^t (Q^t)^\top}_{\triangleq \text{signal}} + \underbrace{U^t Q^{t,\perp}(Q^{t,\perp})^\top}_{\triangleq \text{noise}}, \tag{18}$$

where $Q^t \in \mathbb{R}^{r \times r^\star}$ contains the right singular vectors of $V_{Z^\star}^\top U^t$, i.e., $V_{Z^\star}^\top U^t = V^t \Lambda^t (Q^t)^\top$; and $Q^{t,\perp} \in \mathbb{R}^{(r \times r - r^\star)}$ is the orthonormal complement of $Q^t$. By construction, $\text{span}(Q^t) \cup \text{span}(Q^{t,\perp}) = \mathbb{R}^r$, allowing for the decomposition (18). Further, notice that the noise term is orthogonal to the signal space, i.e. $V_{Z^\star}^\top U^t Q^{t,\perp} = 0$, which implies that once $U^t$ is projected onto the signal space, the only relevant term left is $V_{Z^\star}^\top U^t = V_{Z^\star}^\top U^t Q^t (Q^t)^\top$, hence the name "signal".

Based on (18), and under the assumptions of the theorem, we establish that: **(i)** $U^t Q^t (Q^t)^\top$ is full rank and the signal-term grows as the algorithm progresses. **(ii)** The noise-term grows slower than the signal and remains sufficiently small. **(ii)** The error can be bounded by a polynomial proportional to the initialization size. Similar to [34], the analysis is organized in two phases.

**Phase I (power-like method):** The goal of this phase is to establish that after sufficiently long time $t_\star$ since the initialization ($t = 0$), $\sigma_{\min}(U^{t_\star} Q^{t_\star}) > c\|U^{t_\star} Q^{t_\star,\perp}\|$ and and that $\|(V_{Z^\star}^\perp)^\top V_{U^{t_\star} Q^{t_\star}}\|$ is small, which means that the iterates are better aligned with the signal space than the noise space. Therefore, we are to identify in (9) a mechanism that allows $V_{U^t Q^t}$ to become aligned with $V_{Z^\star}$.

Given the initialization $U^0 = \mu U$, at iteration $t = 1$, we have

$$U^1 = \left(\mathcal{W}^2 + \frac{\alpha}{m}\mathcal{A}^*\mathcal{A}(Z^\star)\right) U^0 + \mu^2 \frac{\alpha}{m}\mathcal{A}^*\mathcal{A}(UU^\top)U^0. \tag{19}$$

Consequently, if $\mu$ is sufficiently small, for the first few iterations $t$, one can write

$$U^t \approx \left(\mathcal{W}^2 + \frac{\alpha}{m}\mathcal{A}^*\mathcal{A}(Z^\star)\right)^t U^0 + \mathcal{O}\left(\mu^2 \|U^0\|\|U\|^2\right). \tag{20}$$

Under the assumption that $\bar{\mathcal{A}}$ fulfills the $\delta_{r^\star}$ RIP, we can establish using the in-network RIP that if $\rho \leq \mathcal{O}\left(\frac{\delta_{2(r^\star+1)}}{m^2\sqrt{m}}\right)$, $\mathcal{A}^*\mathcal{A}(Z^\star) = Z^\star + \varepsilon$, with $\|\varepsilon\| \leq \mathcal{O}(\Delta_{2r^\star}\|Z^\star\|)$. Further, by construction $\mathcal{W}^2 Z^\star = Z^\star \mathcal{W}^2$, implying that $\mathcal{W}^2$ and $Z^\star$ share the same eigenspace. Also, $\lambda_i(\mathcal{W}^2) = 1$, for all $i = 1, \ldots, d$. Consequently $\lambda_i\left(\mathcal{W}^2 + \frac{\alpha}{m}Z^\star\right) = 1 + \frac{\alpha}{m}\lambda_i(Z^\star)$ for $i = 1, \ldots, d$. Therefore, we can further approximate (20) as

$$U^t \approx \left(\mathcal{W}^2 + \frac{\alpha}{m}Z^\star\right)^t U^0 + \mathcal{O}\left(\mu^2 \|U^0\|\|U\|^2\right), \tag{21}$$

where we have disregarded $\varepsilon$, for simplicity of exposition. Leveraging perturbation theory arguments, in the proof we demonstrate that $\varepsilon$ can be properly controlled. Using the above arguments, we have

$$V_{Z^\star} V_{Z^\star}^\top U^t \approx V_{Z^\star}\left(\mathcal{I} + \frac{\alpha}{m}\Lambda_{Z^\star}\right)^t V_{Z^\star}^\top U^0 + V_{Z^\star} V_{Z^\star}^\top \left(\mathcal{W}^2\right)^t V_{Z^\star}^\perp \left(V_{Z^\star}^\perp\right)^\top U^0 + \mathcal{O}\left(\mu^2\|U^0\|\|U\|^2\right) \tag{22}$$

$$V_{Z^\star}^\perp (V_{Z^\star}^\perp)^\top U^t \approx V_{Z^\star}^\perp (\mathcal{W}^2)^t (V_{Z^\star}^\perp)^\top U^0 + \mathcal{O}(\mu^2\|U^0\|\|U\|^2), \tag{23}$$

where the first term in the RHS of (22) corresponds to the power method on the matrix $Z^\star$. Further, we see that the mentioned term grows faster than any other. Consequently, at the time $t_\star$ at which we exit phase I, $U^{t_\star}$ is sufficiently aligned with the signal space as compared to the noise space.

**Phase II (refinement):** In this phase we establish that, given $\sigma_{\min}(U^{t_\star} Q^{t_\star}) \geq c_0\|U^{t_\star} Q^{t_\star,\perp}\|$ and $\|(V_{Z^\star}^\perp)^\top V_{U^{t_\star} Q^{t_\star}}\| \leq c_1$ : **(i)** the alignment $\sigma_{\min}(V_{Z^\star}^\top U^{t_\star} Q^{t_\star})$ grows and stabilizes away from zero, **(ii)** the error $\|U^t Q^{t,\perp}\|$ grows slower than the alignment $\|U^t Q^t\|$ and, **(iii)** the error $\|V_{Z^\perp}^\top V_{U^t Q^t}\|$ remains sufficiently small. A careful study and balance of these quantities yield the final convergence.

**Challenges with respect to the centralized case:** The main challenge with respect to analyses of the GD (e.g., [34]) comes from the distributed nature of the algorithm generating extra error terms (e.g., consensus errors), which significantly complicate the analysis. Our analysis builds on a newly introduced notion of RIP for the algorithm operator $\mathcal{A}$. This is substantially different from the classical RIP or GD mapping [34], which lack of the network gossip matrix $\mathcal{W}^2$.

**Challenges with respect to existing distributed optimization approaches:** The standard approach in the distributed optimization literature typically takes the route of splitting the algorithm dynamics

in its average and consensus error. We deviated from such decomposition, because controlling such errors on $U^t(U^t)^\top$ would result in bounds of the type

$$\|U^t(U^t)^\top - \mathcal{J}U^t(U^t)^\top \mathcal{J}\| \leq \mathcal{O}\left(\rho\|U^t(U^t) - Z^\star\|\right) \tag{24}$$

which are insufficient to understand for example the dynamics of $\|(V_{Z^\star}^\perp)^\top V_{U^tQ^t}\|$. Furthermore, the split into signal and noise subspaces allows us to invoke the in-network RIP with $r^\star$, which would not be the case if splitting the iterates along consensus and non-consensus spaces.

Specifically regarding to phase I of the scheme, another mode classical approach would be "centering" the dynamics around the centralized trajectory of the power method, i.e.,

$$U^1 = \left(\mathcal{J} + \frac{\alpha}{m}\mathcal{J}\mathcal{A}^*\mathcal{A}(Z^\star)\mathcal{J}\right)U^0 - \mu^2\frac{\alpha}{m}\mathcal{A}^*\mathcal{A}(UU^\top)U^0 \tag{25}$$

$$+ (\mathcal{W}^2 - \mathcal{J})U^0 + \frac{\alpha}{m}\left(\mathcal{A}^*\mathcal{A}(Z^\star) - \mathcal{J}\mathcal{A}^*\mathcal{A}(Z^\star)\mathcal{J}\right)U^0, \tag{26}$$

which, using the in-network RIP and the fact that $\mu^2$ and $t$ are sufficiently small, would yield

$$U^t \approx \left(\mathcal{J} + \frac{\alpha}{m}\mathcal{J}\mathcal{A}^*\mathcal{A}(Z^\star)\mathcal{J}\right)^t U^0 + \mu^2\mathcal{O}\left(\|U\|^2\|U^0\|\right) + \mathcal{O}\left(\rho^2\|U^0\|\right). \tag{27}$$

Here, the degree of freedom to control the error terms (second and third) are $\rho$ and $\mu$. This however would enforce the undesirable condition $\rho \leq \mathcal{O}(\mu)$, which couples the network connectivity with the size of the initialization.

## 4 Numerical experiments

We discuss some preliminary experiments validating our theoretical findings. All simulations are performed on a Apple M2 Pro @ 3.5 GHz computer, using 32 GB RAM running macOS Ventura 13.3.1. We generate a random matrix $\bar{X} \in \mathbb{R}^{d \times r^\star}$, with $d = 50$ and $r^\star = 2$, which we use through all experiments. The symmetric measurement matrices are generated as $A_i = (1/2)(S_i + S_i^\top)$, where $S_i \in \mathbb{R}^{d \times d}$ have i.i.d. standard Gaussian elements. The communication networks are generated as Erdős-Rényi graphs, with link activation probability $p = 0.05$. and different sizes $m$ (specified in each experiment below). For any generated graph, we set $\bar{W}$ according to Metropolis weights [28], and then let $W = \bar{W}^K$, with the integer $K$ chosen to meet the condition (12) on $\rho$, resulting in $K$ communication rounds per agent/iteration. Finally, we choose $\alpha = 1/4$ and $\mu = d^{-3}$.

**(i) Validating Theorem 1:** This experiment shows that under the conditions of Theorem 1, the test error behaves predictably as that of the centralized GD (up to constant factors). Furthermore, the invariance of such an error with the network size $m$ is also confirmed, as long as $\rho \leq \mathcal{O}(1/m^6)$ (as requested in (12)). In the experiments, the total sample size is $N = 1000$, split equally among agents $m = \{5, 100, 500, 1000\}$. **Fig 1a** plots the normalized test error; **Fig 1b** shows $\|(V_M^\perp)^\top V_{U^t}\|$ which measures the missalignment of $U^t$ with the power method matrix; and and **Fig 1c** displays the $\sigma_{r^\star}(U^t)/\sigma_{r^\star}(X)$ which combined with Fig 1b allows us to claim that the signal $U^tQ^t$ is well aligned with $V_{Z^\star}$ and full ranked. The curves show that the behavior of the decentralized algorithm is close to that of the centralized GD (blue lines). As predicted, the error decays quickly after the correct subspace has been identified. Furthermore, convergence rate and generalization error are almost invariant to network-size scaling, as long as $\rho \leq \mathcal{O}(m^{-6})$.

**(ii) Validating condition (12) on $\rho$:** We showcase the necessity of decreasing $\rho$ while increasing $m$. Given a connected base graph with associated $\bar{W}$, for the sequence of graphs generated with increasing $m = \{10, 50, 100, 500\}$, we let $\bar{W} = \bar{W}^T$, with $T$ such that $\rho \approx 0.85$, for all $m$. This eventually violates (12). **Fig. 2** (resp. **Fig. 2b**) plots the normalized generalization error versus the iterations, for $m = 1$, $m = 10$ and $m = 50$ (resp. $m = 100$ and $m = 500$, where the two curves are only up to the iterations $t = 70$ and $t = 17$, respectively). The figures demonstrate the necessity of $\rho$ scaling down with $m$ increasing. In fact, both the rate and achievable estimation errors degrade (and eventually break down) as the network size increases while keeping $\rho$ fixed. We claim that this stems from the fact that if the network is not sufficiently well-connected, the in-network RIP does not hold with sufficiently small tolerance, yielding to a failure of the power method early stage and consequently producing an unrecoverable missalignment with the signal subspace.

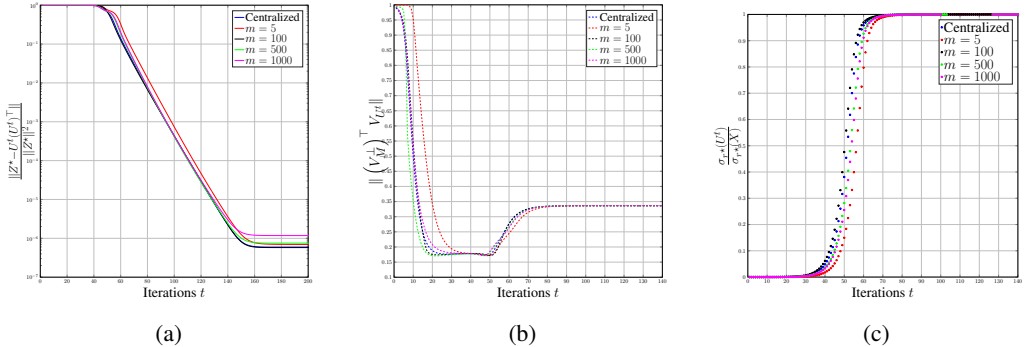

(a)          (b)          (c)

Figure 1: Performance of Algorithm 3, for different network size $m$, with $\rho = \mathcal{O}(m^{-6})$.

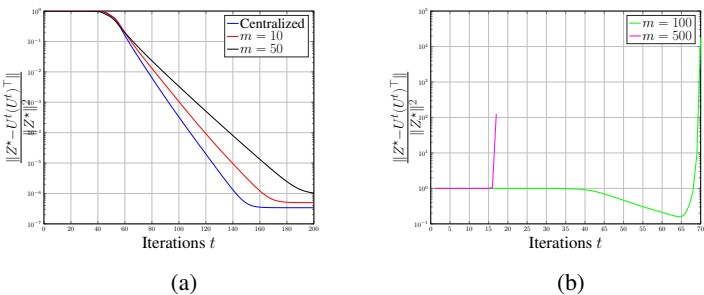

(a)          (b)

Figure 2: Performance of Algorithm 3, for different network size $m$, and fixed $\rho \approx 0.85$.

## Acknowledgments and Disclosure of Funding

Funding in direct support of this work: ONR Grant # N00014-21-1-267.

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
