# Supplementary Material

## Contents

## A  Preliminaries: On the In-Network RIP and its Consequences

This section collects some preliminary results on the introduced In-network RIP (Definition 3) and related properties/consequences, which will be used through the rest of the appendix. The proof of these intermediate results can be found in Appendix G.

**Lemma 2.** *Suppose that $\bar{\mathcal{A}}$ satisfies the $(\delta_{2(r+1)}, 2(r+1))-$RIP, and the gossip matrix is chosen according to Assumption 1 with*

$$\rho^2 \leq \frac{2\delta_{2(r+1)}^2}{4m^5(1+2\delta_{2(r+1)})^2}.$$

*Then, for any matrix $Z \in \mathbb{R}^{md \times md}$ such that each of its $d \times d$ blocks $[Z]_{ij}$ and its block average $\bar{Z} = \frac{1}{m^2}\sum_{i=1}^m \sum_{j=1}^m Z_{ij}$ are of rank at most $r$, it holds*

$$\|\mathcal{W}Z\mathcal{W} - \mathcal{A}^*\mathcal{A}(Z)\| \leq 2\delta_{2r}\|\mathcal{J}Z\mathcal{J}\|_F + 2\left(\rho + (1+2\delta_{2(r+1)})\hat{\Delta}_{2(r+1)}\right)\|Z - \mathcal{J}Z\mathcal{J}\|_F,$$

*where*

$$\hat{\Delta}_{2(r+1)} \triangleq 4m^5\rho^2\frac{(1+2\delta_{2(r+1)})^2}{\delta_{2(r+1)}}.$$

**Lemma 3.** *Suppose that $\bar{\mathcal{A}}$ satisfies the $(\delta_4, 4)-$RIP, and the gossip matrix is chosen according to Assumption 1 with*

$$\rho^2 \leq \frac{2\delta_4^2}{4m^4\left(1+2\delta_4\right)^2}.$$

*Then,*

$$\|\mathcal{W}Z\mathcal{W} - \mathcal{A}^*\mathcal{A}(Z)\| \leq \left(2\delta_4 + 2\left(\rho + (1+2\delta_4)\hat{\Delta}_4\right)\right)\|Z\|_*,$$

*for any matrix $Z \in \mathbb{R}^{md \times md}$. Here, $\|Z\|_\star$ denotes the nuclear norm of $Z$.*

**Lemma 4.** *Suppose that $\bar{\mathcal{A}}$ satisfies the $(\delta_{2r^\star}, 2r^\star)-$RIP, and the gossip matrix is chosen according to Assumption 1, with*

$$\rho \leq \frac{2\delta_{2r^\star}^2}{4m^5(1+2\delta_{2r^\star})^2}.$$

*Then, the following hold:*

$$\|(\mathcal{I} - \mathcal{J})\mathcal{A}^*\mathcal{A}(Z^\star)\| \leq \rho^{1/2}\left(\|Z^\star\| + \delta_{2r^\star}\|Z^\star\|_F\right),$$
$$\|(\mathcal{I} - \mathcal{J})\mathcal{A}^*\mathcal{A}(Z^\star)(\mathcal{I} - \mathcal{J})\| \leq \rho\left(\|Z^\star\| + \delta_{2r^\star}\|Z^\star\|_F\right).$$

# B    Proof of Theorem 1

As preliminary sketched in Sec. 3.1, the proof of the theorem in organized in two phases. Phase I (Appendix B.1) shows that the iterates $U^t$ stay in close proximity to a carefully selected reference sequence, $\tilde{U}^t$. This reference sequence exhibits desirable alignment with the signal space, a key factor to ensure progresses of $U^t$ toward the true solution. Phase II (Appendix B.2) demonstrates that this established alignment of the iterates to the signal space remains steady while the generalization error progressively lessens.

## B.1    Phase I

This section is devoted to the analysis of the spectral phase of the algorithm, with detailed proof of the supporting lemmata deferred to Appendix E.

Recalling the signal plus noise decomposition of the iterates $U^t$ as in (18), our primary objective is to demonstrate that, after a certain elapsed time $t$ post-initialization ($t = 0$), the signal components $\sigma_{\min}(U^tQ^t)$ and $\|V_{Z^\star}^\top U^tQ^t\|$ "outweigh" the noise-related terms $\|U^tQ^{t,\perp}\|$ and $\|(V_{Z^\star}^\perp)^\top U^tQ^{t,\perp}\|$, respectively. Essentially, this indicates a significant alignment of the iterates $U^t$ with the signal eigenspace. The proof is organized in the following three steps:

• **Step 1 (approximating the trajectory):** We show that, for any given initialization $U^0 = \mu U$ with small $\mu$, and sufficiently connected network ($\rho$ small), the iterates $U^t$ can be closely approximated by the following dynamics in the first few iterations $t$ :

$$\tilde{U}^t = \left(\mathcal{W}^2 + \frac{\alpha}{m}\mathcal{A}^*\mathcal{A}(Z^\star)\right)^t U^0 =: \mathcal{M}^t U^0, \quad \text{with} \quad \mathcal{M} := \mathcal{W}^2 + \frac{\alpha}{m}\mathcal{A}^*\mathcal{A}(Z^\star). \quad (28)$$

We refer to Sec. 3.1 for further insight behind the choice of such a reference trajectory;

• **Step 2 (bounding the signal- and noise-terms):** Employing the eigen-properties of the mapping $\mathcal{M}$ in (28) and the signal-plus-noise decomposition (18) of $U^t$, we provide suitable bounds of the quantities $\sigma_{\min}(U^t Q^t)$, $\|U^t Q^{t,\perp}\|$, and $\|(V_{Z^\star}^\perp)^\top V_{U^t Q^t}\|$. From these, we determine conditions on the free parameters to achieve the desired alignment of the iterates $U^t$ with the signal eigenspace.

• **Step 3 (from deterministic to random initialization):** The derivations in Steps 1 and 2 hold for any given initialization matrix $U$. In this step, we conclude the analysis by stating the final result when $U$ is randomly selected, with Gaussian entries.

### B.1.1 Step 1: Approximating the trajectory

Employing (28) and following the same steps as in the proof of [34, Lemma 8.1] applied to $\tilde{U}^t$, we can express $U^t$ as a function of $\tilde{U}^t$ as follows:

$$U^t = \tilde{U}^t + E^t, \tag{29}$$

where the error sequence $E^t$ reads

$$E^t = \sum_{i=1}^{t} \left( \mathcal{W}^2 + \frac{\alpha}{m} \mathcal{A}^* \mathcal{A}(Z^\star) \right)^{t-i} \frac{\alpha}{m} \left( \mathcal{A}^* \mathcal{A}(U^{i-1}(U^{i-1})^\top) U^{i-1} \right). \tag{30}$$

Next we quantify the quality of the approximation of $U^t$ by $\tilde{U}^t$. Specifically, defining

$$t^\star \triangleq \min \left\{ t \in \mathbb{N}_+ : \|\tilde{U}^{t-1} - U^{t-1}\| > \|\tilde{U}^{t-1}\| \right\},$$

Lemma 5 below shows that $\tilde{U}^t$ is a good approximation of $U^t$ ($\|E^t\|$ is uniformly bounded) as long as $t \in [t^\star]$. Lemma 6 shows that $t^\star$ is bounded away from zero, providing thus an estimate of the time interval within which $U^t$ is "well approximated" by $\tilde{U}^t$.

**Lemma 5.** *Suppose that **(i)** $\bar{A}$ satisfies the $(\delta_{2(r^\star+1)}, 2(r^\star+1))-$RIP property and the $(\delta_4, 4)-$RIP; and **(ii)** the gossip matrix $W$ is chosen according to Assumption 1, with*

$$\rho^2 \le \frac{2\delta_{2(r^\star+1)}^2}{4m^5(1+2\delta_{2(r^\star+1)})^2}.$$

*Then, $E^t$ can be bounded as*

$$\|E^t\| \le \frac{4m^{-1}}{(1+2\sqrt{r^\star}\delta_{2(r^\star+1)})\lambda_1(\bar{Z}^\star)} \mu^3 \min\{r, md\} \left( 1 + 2\delta_4 + 2\left( \rho + (1+2\delta_4)\hat{\Delta}_4 \right) \right) \times$$

$$\times \left( 1 + \alpha(1+2\sqrt{r^\star}\delta_{2(r^\star+1)})\lambda_1(\bar{Z}^\star) \right)^{3t} \left( \|\mathcal{J}U\| + \rho^{1/2}\sqrt{r^\star}\|(\mathcal{I}-\mathcal{J})U\| \right)^3, \quad \forall t \in [t^\star]. \tag{31}$$

**Lemma 6.** *In the setting of Lemma 5, the following lower bound of $t^\star$ holds:*

$$t^\star \ge \left\lceil \frac{\ln \left( \frac{(1+2\sqrt{r^\star}\delta_{2(r^\star+1)})\lambda_1(\bar{Z}^\star)}{4\left(1+2\delta_4+2\left(\rho+(1+2\delta_4)\hat{\Delta}_4\right)\right)\mu^2 \frac{\left(\|\mathcal{J}U\|+\rho^{1/2}\sqrt{r^\star}\|(\mathcal{I}-\mathcal{J})U\|\right)^3}{m\|U^\top v_1\|}} \right)}{\ln \left( \frac{(1+\alpha\lambda_1(\bar{X}^\star)(1+2\sqrt{r^\star}\delta_{r^\star+1}))^3}{1+\alpha\lambda_1(\bar{X}^\star)(1-2\sqrt{r^\star}\delta_{2(r^\star+1)})} \right)} \right\rceil.$$

### B.1.2 Step 2: Bounding the signal- and noise-terms

Through the section we will use the following notation, which simplifies the statement of our results:

$$\nu \triangleq \lambda_1(\bar{Z}^\star)(1 + \sqrt{r^\star}\delta_{2r^\star}),$$

$$\delta_\rho \triangleq \frac{\sqrt{r^\star}\rho^{1/2}\nu}{\lambda_{r^\star}(\bar{Z}^\star) - 3\delta_{2(r^\star+1)}\sqrt{r^\star}\lambda_1(\bar{Z}^\star)},$$

$$\eta_r \triangleq 2\delta_r + 2(\rho + (1 + 2\delta_r)\hat{\Delta}_r),$$

$$\sigma \triangleq 2\kappa^2 \frac{(1 + \alpha 2\sqrt{r^\star}\delta_{2(r^\star+1)})\lambda_1(\bar{Z}^\star)\|\mathcal{J}U\| + (\rho^2 + \alpha\rho^{1/2}\nu)\|(\mathcal{I} - \mathcal{J})U\|}{c\sigma_{\min}\left(V_{\mathcal{J}M\mathcal{J}}^\top U\right)\left(1 + \alpha 2\sqrt{r^\star}\delta_{2(r^\star+1)}\right)\lambda_1(\bar{Z}^\star)}, \tag{32}$$

$$\gamma_t \triangleq \frac{\mu\sigma_{r^\star+1}(\mathcal{M}^{t-1})\left(\mu\sigma_{r^\star+1}(\mathcal{M})\|\mathcal{J}U\| + (\rho^2 + \alpha\rho^{1/2}\nu)\|(\mathcal{I} - \mathcal{J})U\|\right) + \|E^t\|}{(1 - \delta_\rho)\mu\sigma_{r^\star}(\mathcal{M}^t)\sigma_{\min}(V_{\mathcal{J}M\mathcal{J}}^\top U)}. \tag{33}$$

The upcoming lemma delves into the characteristics of the dynamics $\tilde{U}^t$ (through the eigenproperties of $\mathcal{M}$) to bound the signal and noise quantities of interest $\sigma_{\min}(U^t Q^t)$, $\|\tilde{U}^t Q^{t,\perp}\|$, and $\|(V_{\bar{Z}^\star}^\perp)^\top V_{U^t Q^t}\|$. These bounds will shed light on the conditions ensuring that $U^t$ exhibits sufficient alignment to the signal space (in the sense previously discussed).

**Lemma 7.** *Suppose that **(i)** $\bar{\mathcal{A}}$ satisfies the $(\delta_{2(r^\star+1)}, 2(r^\star+1))-RIP$, with*

$$\delta_{2(r^\star+1)} \leq \frac{1}{128\sqrt{r^\star}}\kappa^{-4};$$

**(ii)** *$W$ is chosen according to Assumption 1, with*

$$\rho \leq \min\left\{\frac{\delta_{2(r^\star+1)}}{2m^2\sqrt{m}(1 + 2\delta_{2(r^\star+1)})}, \frac{(\lambda_{r^\star}(\bar{Z}^\star) - 3\delta_{2(r^\star+1)}\sqrt{r^\star}\lambda_1(\bar{Z}^\star))^2}{3r^\star\lambda_1(\bar{Z}^\star)^2(1 + \sqrt{r^\star}\delta_{2r^\star})^2}\right\}; \tag{34}$$

*and **(iii)** $\gamma_t$ defined in (33) satisfies*

$$\gamma_t \leq \frac{\kappa^{-2}}{17}. \tag{35}$$

*Then, $\sigma_{\min}(U^t Q^t)$, $\|U^t Q^{t,\perp}\|$ and $\|(V_{\bar{Z}^\star}^\perp)^\top V_{U^t Q^t}\|$ can be bounded respectively as*

$$\sigma_{\min}(U^t Q^t) \geq \frac{\mu(1 - \delta_\rho)}{2}\sigma_{r^\star}(\mathcal{M}^t)\sigma_{\min}(V_{\mathcal{J}M\mathcal{J}}^\top U) - \|E^t\| \tag{36a}$$

$$\|U^t Q^{t,\perp}\| \leq \frac{\kappa^{-2}}{8}\mu\sigma_{r^\star}(\mathcal{M}^t)\sigma_{\min}(V_{\mathcal{J}M\mathcal{J}}^\top U), \tag{36b}$$

$$\|(V_{\bar{Z}^\star}^\perp)^\top V_{U^t Q^t}\| \leq 56\left(\sqrt{r^\star}\kappa^2\delta_{2(r^\star+1)} + \gamma_t\right). \tag{36c}$$

The desiderata is a sufficient large $\sigma_{\min}(U^t Q^t)$ (persistent signal component) along with small noise-terms $\|U^t Q^{t,\perp}\|$ and $\|(V_{\bar{Z}^\star}^\perp)^\top V_{U^t Q^t}\|$. A close look at equations (36) suggests that this is achievable by guaranteeing sufficiently small $\rho$ (quantified by (34)), $\|E^t\|$ and $\gamma_t$. The latter represents a suitable noise-to-signal ratio measure, which accounts for the deviation of $U^t$ from $\tilde{U}^t$ as well as the consensus discrepancies in $U$.

In Step 1 (Lemma 5) we showed that $\|E^t\|$ remains "small" as long as $t \in [t^\star]$. The subsequent lemma proves that, after a certain number of iterations nestled within $[t^\star]$, one can confidently ensure that $\gamma_t$ also is small, so that (36) will hold.

**Lemma 8.** *Reinstate Assumptions (i)-(ii) of Lemma 7. Choose $\mu$ such that*

$$\mu^2 \leq \frac{(1 + 2\sqrt{r^\star}\delta_{2(r^\star+1)})\lambda_1(\bar{Z}^\star)}{4(1 + \eta)\min\{r, md\}}\min\left\{\frac{m\|U^\top v_1\|}{(\|\mathcal{J}U\| + \sqrt{r^\star}\rho^{1/2}\|(\mathcal{I} - \mathcal{J})U\|)^3}(\sigma)^{-96\kappa^2},\right.$$

$$\left., \frac{c(1 - \delta_\rho)m\sigma_{\min}(V_{\mathcal{J}M\mathcal{J}}^\top U)}{8\kappa^2\left(\|\mathcal{J}U\| + \rho^{1/2}\sqrt{r^\star}\|(\mathcal{I} - \mathcal{J})U\|\right)^3}(\sigma)^{-12\kappa^2}\right\}, \tag{37}$$

*where $c$ is a strictly positive constant $c < 1/2$ and $\sigma$ is defined in (32).*

*Then after*

$$t_\star \triangleq \left\lceil \ln(\sigma) \left( \ln \left( \frac{1 + \alpha\lambda_{r^\star}(\bar{Z}^\star) - \alpha 2\sqrt{r^\star}\delta_{2(r^\star+1)}\lambda_1(\bar{Z}^\star)}{1 + \alpha 2\sqrt{r^\star}\delta_{2(r^\star+1)}\lambda_1(\bar{Z}^\star)} \right) \right)^{-1} \right\rceil$$

*iterations, it holds*

$$\gamma_{t_\star} \leq c\kappa^{-2}.$$

Lemma 8 establishes the existence of a time $t_\star$ at which $\gamma_{t^\star}$ is sufficiently small. Notice that, in the setting of the lemma, $t_\star \leq t^\star$. As such, we can apply also Lemma 5 at $t = t_\star$, ensuring that $\|E^{t_\star}\|$ concurrently resides under the desired limit. This, coupled with a judicious choice of $\mu$, results in the ultimate bounds on the quantities in (36), as detailed next.

Using Lemma 8 in (36), yields

$$\sigma_{\min}(U^{t_\star}Q^{t_\star}) \geq (1 - \delta_\rho)\frac{\mu}{4}\beta - \|E^t\|, \tag{38}$$

$$\|U^{t_\star}Q^{t_\star,\perp}\| \leq \frac{\kappa^{-2}}{8}\mu\beta, \tag{39}$$

$$\left\| (V_{\bar{Z}^\star}^\perp)^\top V_{U^{t_\star}Q^{t_\star}} \right\| \leq 56 \left( \sqrt{r^\star}\kappa^2\delta_{2(r^\star+2)} + c\kappa^{-2} \right),$$

where for convenience, we defined

$$\beta \triangleq \sigma_{r^\star}(\mathcal{M}^{t_\star})\sigma_{\min}(V_{\mathcal{J}M\mathcal{J}}^\top U).$$

We proceed to further bound the RHS of (38) and (39).

By definition of $\gamma_t$ and choosing $c \leq 1/17$, it holds

$$\|E^t\| \leq \frac{1}{17}(1 - \delta_\rho)\mu\beta.$$

Consequently

$$\sigma_{\min}(U^{t_\star}Q^{t_\star}) \geq \frac{\mu}{2}(1 - \delta_\rho)\beta - \frac{\mu}{17}(1 - \delta_\rho)\mu\beta \overset{(a)}{\geq} \frac{\mu\beta}{4},$$

whereby in (a) we used $\delta_\rho \leq 1/3$, which holds under (34).

It remains to upper and lower bound $\beta$. Under the postulated assumption on $\delta_{2r^\star}$, $\lambda_{r^\star}(\mathcal{M}^t) > 1$. Therefore,

$$\beta \geq \sigma_{\min}\left(V_{\mathcal{J}M\mathcal{J}}\right).$$

Similarly, using Weyl's inequality,

$$\sigma_{r^\star}(\mathcal{M}^{t_\star}) \leq \left( 1 + \alpha\lambda_{r^\star}(\bar{Z}^\star) + \frac{\alpha}{m}\|\mathcal{W}Z^\star\mathcal{W} - \mathcal{A}^\ast\mathcal{A}(Z^\star)\| \right)^t$$

$$\overset{(a)}{\leq} \left( 1 + \alpha\lambda_{r^\star}(\bar{Z}) + \alpha 2\sqrt{r^\star}\delta_{2r^\star}\lambda_1(\bar{Z}^\star) \right)^{t_\star}$$

$$\leq \left( 1 + \alpha\lambda_{r^\star}(\bar{Z}^\star) + \alpha 2\sqrt{r^\star}\delta_{2r^\star}\lambda_1(\bar{Z}^\star) \right)^{\frac{\ln(\sigma)}{\ln(1+\alpha\lambda_{r^\star}(\bar{Z}^\star)+\alpha 2\sqrt{r^\star}\delta_{2r^\star}\lambda_1(\bar{Z}^\star))} \times \frac{\ln(1+\alpha\lambda_{r^\star}(\bar{Z}^\star)+\alpha 2\sqrt{r^\star}\delta_{2r^\star}\lambda_1(\bar{Z}^\star))}{\ln\left(\frac{1+\alpha\lambda_{r^\star}-\alpha 2\sqrt{r^\star}\delta_{2r^\star}\lambda_1(\bar{Z}^\star)}{1+\alpha 2\sqrt{r^\star}\delta_{2(r^\star+1)}\lambda_1(\bar{Z}^\star)}\right)} + 1$$

$$\overset{(b)}{\leq} 2\sigma^{\frac{\ln\left(1+\alpha\lambda_{r^\star}(\bar{Z}^\star)+\alpha 2\sqrt{r^\star}\delta_{2r^\star}\lambda_1(\bar{Z}^\star)\right)}{\ln\left(\frac{1+\alpha\lambda_{r^\star}-\alpha 2\sqrt{r^\star}\delta_{2r^\star}\lambda_1(\bar{Z}^\star)}{1+\alpha 2\sqrt{r^\star}\delta_{2(r^\star+1)}\lambda_1(\bar{Z}^\star)}\right)}}$$

where in (a) we used Lemma 2 and (b) follows from $1 + \alpha\lambda_{r^\star}(\bar{Z}^\star) + 2\alpha\sqrt{r^\star}\delta_{2r^\star}\lambda_1(\bar{Z}^\star) \leq 2$, due to the assumption on $\alpha$ and $\delta_{2r^\star}$. Finally, we bound the exponent as

$$\frac{\ln\left(1 + \alpha\left(\lambda_{r^\star}(\bar{Z}^\star) + \alpha\sqrt{r^\star}\delta_{2r^\star}\lambda_1(\bar{Z}^\star)\right)\right)}{\ln\left(\frac{1+\alpha\lambda_{r^\star}-\alpha 2\sqrt{r^\star}\delta_{2r^\star}\lambda_1(\bar{Z}^\star)}{1+\alpha 2\sqrt{r^\star}\delta_{2(r^\star+1)}\lambda_1(\bar{Z}^\star)}\right)} \leq \frac{\alpha\left(\lambda_{r^\star}(\bar{Z}^\star) + 2\sqrt{r^\star}\delta_{2r^\star}\lambda_1(\bar{Z}^\star)\right)}{\frac{\alpha\left(\lambda_{r^\star}(\bar{Z}^\star) - 4\sqrt{r^\star}\delta_{2r^\star}\lambda_1(\bar{Z}^\star)\right)}{1+\alpha\lambda_{r^\star}(\bar{Z}^\star)-2\sqrt{r^\star}\delta_{2(r^\star)}\lambda_1(\bar{Z}^\star)}}$$

$$\leq \frac{(1 + \alpha\lambda_{r^\star}(\bar{Z}))(1 + 1/64)}{(\bar{Z}^\star)(1 - 1/32)} \overset{(a)}{\leq} \frac{(1 + 3/4)(1 + 1/64)}{1 - 1/32} \overset{(b)}{\leq} 2,$$

where in (a) we used $\alpha \leq \frac{3}{4}\|\bar{Z}^\star\|^{-1}\kappa^{-2}$.

We conclude the analysis, proving the following bound for $U^{t_\star}$: $\|U^{t_\star}\| \leq 3\sqrt{m}\|\bar{X}\|$.

Following a similar procedure as in the proof of Lemma 5, we can write

$$\|U^{t_\star}\| \leq \mu\|M^{t_\star}U\| + \|E^{t_\star}\| \leq \mu\left(1 + \alpha(1 + 2\sqrt{r^\star}\delta_{2(r^\star+1)})\lambda_1(\bar{Z}^\star)\right)^{t_\star}$$
$$\times (\|\mathcal{J}U\| + \rho^{1/2}\sqrt{r^\star}\|(\mathcal{I} - \mathcal{J})U\|) + \frac{\tilde{c}_3}{2}(1 - \delta_\rho)\mu\sigma_{r^\star}(M^{t_\star})\sigma_{\min}(V_{\mathcal{J}M\mathcal{J}}^\top U)\kappa^{-2}.$$

The bound $\|U^{t_\star}\| \leq 3\sqrt{m}\|\bar{X}\|$ is obtained by using the bound of $\mu$ in (37).

We summarize next the obtained properties of the signal and noise component at time $t_\star$, which represents the "exist" of Phase I.

**Proposition 1.** *Restate the conditions of Lemma 8, choosing a strictly positive constant $c \leq \frac{1}{17}$, and assume $\alpha \leq \frac{3}{4}\kappa^{-2}\|\bar{Z}^\star\|^{-1}$. The following hold:*

$$\sigma_{\min}\left(U^{t_\star}Q^{t_\star}\right) \geq \frac{\mu\beta}{4}, \tag{40a}$$

$$\|U^{t_\star}Q^{t_\star,\perp}\| \leq \frac{\kappa^{-2}}{8}\mu\beta,, \tag{40b}$$

$$\|(V_{\bar{Z}^\star}^\perp)^\top V_{U^{t_\star}Q^{t_\star}}\| \leq 56\left(\sqrt{r^\star}\kappa^2\delta_{2(r^\star+1)} + c\kappa^{-2}\right), \tag{40c}$$

$$\|U^{t_\star}\| \leq 3\sqrt{m}\|\bar{X}\|, \tag{40d}$$

*with*

$$\sigma_{\min}(V_{\mathcal{J}M\mathcal{J}}^\top U) \leq \beta \leq 2\sigma_{\min}(V_{\mathcal{J}M\mathcal{J}}^\top)\sigma^2. \tag{41}$$

### B.1.3 Step 3: From deterministic to random initialization

We conclude the analysis of Phase I allowing for $U$ to be random with Gaussian entries. We obtain the following counterpart of Proposition 1.

**Proposition 2.** *Reinstate the conditions of Proposition 1, and choose a strictly positive constant $c \leq \frac{1}{17}$. Additionally, suppose that $U \in \mathbb{R}^{md \times r}$ is a random matrix such that each of its elements is i.i.d. and distributed according to $\mathcal{N}(0, \sqrt{m}/\sqrt{r})$. Then, (40) holds with probability at least $1 - \exp(-r/4) - \exp(-c_2 r)$, wherein $\mu$ and $\beta$ satisfy*

$$\mu^2 \leq \frac{(1 + 2\alpha\sqrt{r^\star}\delta_{2(r^\star+1)}\lambda_1(\bar{Z}^\star))}{4(1 + \eta)\min\{r, md\}}\min\left\{\frac{r\sqrt{r}}{2d\sqrt{d}(1 + (\rho r^\star m)^{1/2})}\sigma^{-96\kappa^2},\right.$$
$$\left.\frac{c(1 - \delta_\rho)e^{-1/2}(2 - \sqrt{2})r\sqrt{r}}{16c_2\kappa^2 d\sqrt{d}(1 + (\rho r^\star m)^{1/2})}\sigma^{-12\kappa^2}\right\},$$

*with*

$$\sigma \leq 4c_2\kappa^2\frac{(1 + 2\sqrt{r^\star}\alpha\delta_{2(r^\star+1)}\lambda_1(\bar{Z}))\sqrt{\frac{d}{r}} + (\rho^2 + \alpha\rho^{1/2}\Delta)\sqrt{\frac{dm}{r}}}{ce^{-1/2}(2 - \sqrt{2})(1 + 2\alpha\sqrt{r^\star}\delta_{2(r^\star+1)}\lambda_1(\bar{Z}^\star))},$$

*and*

$$\sqrt{m}\frac{e^{-1/2}(2 - \sqrt{2})}{2c_2} \leq \beta \leq 2\sqrt{m}\sigma^2.$$

## B.2 Phase II

We exploit now the properties of the iterates at time $t_\star$ to establish further refinements on the dynamics of the signal and noise terms.

We begin establishing some bounds of the quantities of interest at time $t + 1$, conditioned to the following events at time $t$:

**(a)** $\|U^t\| \leq 3\sqrt{m}\|\bar{X}\|$;

**(b)** $\|(V_{Z^\star}^\perp)^\top V_{U^t Q^t}\| \le c_1 \kappa^{-1}$;

**(c)** $\|\mathcal{W}\Delta^t \mathcal{W} - \mathcal{A}^* \mathcal{A}(\Delta^t)\| \le c_2 m \lambda_{r^\star}(\bar{Z}^\star)$.

We then proceed by induction to show that such events hold.

In the following we define for convenience

$$\Delta^t \triangleq Z^\star - U^t (U^t)^\top. \tag{42}$$

Also $\|\!|\cdot|\!\|$ denotes any matrix norm, thus fulfilling $\|\!|ABC|\!\| \le \|A\|\|\!|B|\!\|\|C\|$.

**Lemma 9.** *Assume that* $\alpha \le \frac{1}{81000}\|\bar{Z}^\star\|^{-1}\kappa^{-2}$. *The we have the following:*

*(i) If (a) holds, with $c_2 \le 1$, then*

$$\|U^{t+1}\| \le 3\sqrt{m}\|\bar{X}\|. \tag{43}$$

*(ii) If (a)-(c) hold, with $c_1, c_2 \le 1/110$, and $V_{Z^\star}^\top U^t$ has full rank, then*

$$\sigma_{\min}(V_{Z^\star}^\top U^{t+1} Q^t) \ge \left(1 + \frac{\alpha}{4}\lambda_{r^\star}(\bar{Z}^\star) - \frac{\alpha}{m}\sigma_{\min}^2\left(V_{Z^\star}^\top U^t\right)\right)\sigma_{\min}(V_{Z^\star}^\top U^t). \tag{44}$$

*(iii) If (a), (b) hold, with $c_1 \le 1/1200$, $V_{Z^\star}^\top U^{t+1} W^t$ is full row rank, and*

$$\alpha \le (1/1200)m\|\mathcal{W}\Delta^t \mathcal{W} - \mathcal{A}^* \mathcal{A}(\Delta^t)\|^{-1}, \tag{45}$$

*then*

$$\|U^{t+1}Q^{t+1,\perp}\| \le \left(1 - \frac{2\alpha}{m}\left(\frac{\|U^t Q^{t,\perp}\|}{4} - \|\mathcal{W}\Delta^t \mathcal{W} - \mathcal{A}^* \mathcal{A}(\Delta^t)\|\right) + \right.$$
$$\left. + 9\alpha\|\bar{Z}^\star\|\|(V_{Z^\star}^\perp)^\top V_{U^t Q^t}\|\right)\|U^t Q^{t,\perp}\|. \tag{46}$$

*(iv) If (a), (b) hold, with $c_1 \le 1/(24 \cdot 800)$, and*

$$\|U^t Q^{t,\perp}\| \le \min\left\{2\sigma_{\min}(U^t Q^t), c_0 \kappa^{-2}\sqrt{m}\|\bar{X}\|\right\}, \tag{47}$$

*then,*

$$\|(V_{Z^\star}^\perp)^\top V_{U^{t+1}Q^{t+1}}\| \le \left(1 - \alpha\frac{\lambda_{r^\star}(\bar{Z}^\star)}{4}\right)\|(V_{Z^\star}^\perp)^\top V_{U^t Q^t}\|$$
$$+ \frac{100\alpha}{m}\|\mathcal{W}\Delta^t \mathcal{W} - \mathcal{A}^* \mathcal{A}(\Delta^t)\| + \frac{500\alpha^2}{m^2}\|\Delta^t\|^2. \tag{48}$$

*(v) If (a), (b) hold, with $c_1 \le 1/(81000)$, and further*

$$\sigma_{\min}(U^t Q^t) \ge \sqrt{\frac{m}{10}}\sigma_{r^\star}(\bar{X}), \tag{49}$$

$$\max\left\{\|\!|V_{Z^\star}^\top\left(\mathcal{W}\Delta^t \mathcal{W} - \mathcal{A}^* \mathcal{A}(\Delta^t)\right)|\!\|, \|\!|V_{U^t Q^t}^\top(\mathcal{W}\Delta^t \mathcal{W} - \mathcal{A}^* \mathcal{A}(\Delta^t))|\!\|, \right.$$
$$\left. , \|\mathcal{W}\Delta^t \mathcal{W} - \mathcal{A}^* \mathcal{A}(\Delta^t)\|\right\} \le c_0 \kappa^{-2}\|\!|\Delta^t|\!\|, \tag{50}$$

*then,*

$$\|\!|V_{Z^\star}^\top\left(\Delta^{t+1}\right)|\!\| \le \left(1 - \frac{\alpha}{200}\lambda_{r^\star}(\bar{Z}) + \alpha 9\|\bar{Z}^\star\|\rho\right)\|\!|V_{Z^\star}^\top \Delta^t|\!\| + \frac{\alpha\lambda_{r^\star}(\bar{Z}^\star)}{100}\|\!|U^t Q^{t,\perp}(U^t Q^{t,\perp})^\top|\!\|. \tag{51}$$

*Proof.* See Appendix F. $\square$

Now, we strive to establish, via induction, that the events outlined in (a)-(c), alongside the additional conditions utilized in Lemma 9, are upheld. The proof is organized in two steps. **Step 1** delves into the dynamics of the quantities under scrutiny, initiating from the iterates at time $t_\star$ (which marks the exit from Phase I), hence possessing a favorable alignment with the signal subspace (in the sense of Proposition 1). Consequently, it is proven that throughout the progression of iterations, the alignment, as quantified by $\sigma_{\min}(V_{Z^\star}^\top U^t)$, accelerates at a pace that outstrips the misalignment, gauged via $\|U^t Q^{t,\perp}\|$ and $\|(V_{Z^\star})^\top V_{U^t Q^t}\|$, eventually attaining a suitably elevated value. Upon reaching an adequate growth in $\sigma_{\min}(V_{Z^\star}^\top U^t)$, we demonstrate in **Step 2** that (i) $\sigma_{\min}(V_{Z^\star}\bar{Z}^\star)$ establishes a stable lower limit, while (ii) $\|U^t Q^{t,\perp}\|$ exhibits a gradual increase, (iii) $\|(V_{Z^\star}^\perp)^\top V_{U^t Q^t}\|$ remains stable, and (iv) the generalization error continually diminishes to yield the final result.

### B.2.1 Step 1: Growing alignment phase

Select a time $t_\star$ such that there holds

$$\|U^{t_\star} Q^{t_\star,\perp}\| \leq 2\tau,$$
$$\|U^{t_\star}\| \leq 3\sqrt{m}\|\bar{X}\|,$$
$$\|(V_{Z^\star}^\perp)^\top V_{U^t Q^t}\| \leq \bar{c}_2 \kappa^{-2}, \tag{52}$$
$$\sigma_{\min}(U^{t_\star} Q^{t_\star}) \geq \tau.$$

We will anticipate that we will later identify such a $t_\star$ as that in Lemma 8 with $\sigma$ as in Proposition 2 and $\nu = \frac{\mu\beta}{4}$ as in Proposition 1 with bounds on $\beta$ as in Proposition 2.

We prove next by induction that the following hold:

$$\sigma_{\min}(V_{Z^\star}^\top U^t) \geq \frac{1}{2}\left(1 + \frac{\alpha}{8}\lambda_{r^\star}(\bar{Z}^\star)\right)^{t-t_\star}\tau, \tag{53a}$$

$$\|U^t Q^{t,\perp}\| \leq 2\left(1 + 80\bar{c}_1\lambda_{r^\star}(\bar{Z})\right)^{t-t_\star}\tau, \tag{53b}$$

$$\|U^t\| \leq 3\sqrt{m}\|\bar{X}\|, \tag{53c}$$

$$\|(V_{Z^\star}^\perp)^\top V_{U^t Q^t}\| \leq \bar{c}_2 \kappa^{-2}, \tag{53d}$$

for all $t \in [t_\star, t_1]$, where

$$t_1 \triangleq \min\left\{t \geq t_\star : \sigma_{\min}(V_{Z^\star}^\top U^t) \geq \frac{\sigma_{r^\star}(\bar{X})\sqrt{m}}{\sqrt{10}}\right\}.$$

Notice that (53) holds at time $t = t_\star$. It is sufficient to note for (53a) that

$$\sigma_{\min}(V_{Z^\star}^\top U^{t_\star}) \geq \sigma_{\min}(V_{Z^\star}^\top V_{U^{t_\star} Q^{t_\star}})\sigma_{\min}(U^{t_\star} Q^{t_\star}) \overset{(a)}{\geq} (1 - \bar{c}_2 \kappa^{-2})\sigma_{\min}(U^{t_\star} Q^{t_\star}) \overset{(b)}{\geq} \frac{\tau}{2},$$

where in (a) we used (52) and in (b) we set $\bar{c}_2 \leq 1/2$.

We begin proving that $\|\mathcal{W}\Delta^t\mathcal{W} - \mathcal{A}^*\mathcal{A}(\Delta^t)\|$ is sufficiently small (event **(c)**) for any time $t$ within the interval of interest. Using

$$\Delta_S^t \triangleq Z^\star - U^t Q^t (U^t Q^t)^\perp \quad \text{and} \quad \Delta_\perp^t \triangleq -U^t Q^{t,\perp}(U^t Q^t)^\perp,$$

we have

$$\|\mathcal{A}^*\mathcal{A}(\Delta^t) - \mathcal{W}\Delta^t\mathcal{W}\| = \|\mathcal{A}^*\mathcal{A}(\Delta_S^t + \Delta_\perp^t) - \mathcal{W}(\Delta_S^t + \Delta_\perp^t)\mathcal{W}\|$$
$$\leq \|\mathcal{A}^*\mathcal{A}(\Delta_S^t) - \mathcal{W}\Delta_S^t\mathcal{W}\| + \|\mathcal{A}^*\mathcal{A}(\Delta_\perp^t) - \mathcal{W}\Delta_\perp^\top\mathcal{W}\|.$$

Observe that each of the $d \times d$ blocks of $\Delta_S^t$ is of rank at most $2r^\star$ and so is its average block. Hence, under the assumption that $\bar{\mathcal{A}}$ satisfies the $(\delta_{4(r^\star+1)}, 4(r^\star+1))-$RIP, we can invoke Lemma 2 and obtain

$$\|\mathcal{A}^*\mathcal{A}(\Delta_S^t) - \mathcal{W}\Delta_S^t\mathcal{W}\| \leq 2\delta_{4(r^\star+1)}\|\mathcal{J}\Delta_S^t\mathcal{J}\|_f + 2\eta_{4(r^\star+1)}\|\Delta_S^t - \mathcal{J}\Delta_S^t\mathcal{J}\|_F,$$

where

$$\eta_{4(r^\star+1)} \triangleq 2\left(\rho + (1 + 2\delta_{4(r^\star+1)})\hat{\Delta}_{4(r^\star+1)}\right).$$

Consequently,

$$\|\mathcal{A}^*\mathcal{A}(\Delta_S^t) - \mathcal{W}\Delta_S^t\mathcal{W}\| \leq \left(2\sqrt{2r^\star}\delta_{4(r^\star+1)} + 2\sqrt{2r^\star}m\eta_{4(r^\star+1)}\right)\|\Delta_S^t\|.$$

Further, $-\Delta_\perp^t$ is a p.s.d matrix with rank at most $r - r^\star$. Thus, we may invoke Lemma 3

$$\|\mathcal{W}\Delta_\perp^t\mathcal{W} - \mathcal{A}^*\mathcal{A}(\Delta_\perp^t)\| \leq (2\delta_4 + 2\eta_4)\|\Delta_\perp^t\|_*,$$

where

$$\eta_4 \triangleq 2\rho + 2\left(\rho + (1 + 2\delta_4)\right)\hat{\Delta}_4.$$

Combining we obtain

$$\|\mathcal{A}^*\mathcal{A}(\Delta^t) - \mathcal{W}\Delta^t\mathcal{W}\| \leq 2\sqrt{2r^\star}\left(\delta_{4(r^\star+1)} + m\eta_{4(r^\star+1)}\right)\|\Delta_S^t\| + 2(r - r^\star)(\delta_4 + \eta_4)\|\Delta_\perp^t\|.$$

Using the induction hypothesis (53c), we can write

$$\|\mathcal{A}^*\mathcal{A}(\Delta^t) - \mathcal{W}\Delta^t\mathcal{W}\| \leq 2\sqrt{2r^\star}\left(\delta_{4(r^\star+1)} + m\eta_{4(r^\star+1)}\right)10m\|\bar{Z}^\star\|$$
$$+ 2(r - r^\star)(\delta_4 + \eta_4)\|U^tQ^{t,\perp}\|^2.$$

Further, under the induction hypothesis (53b), it holds

$$\|\mathcal{A}^*\mathcal{A}(\Delta^t) - \mathcal{W}\Delta^t\mathcal{W}\| \leq 2\sqrt{2r^\star}\left(\delta_{4(r^\star+1)} + m\eta_{4(r^\star+1)}\right)10m\|\bar{Z}^\star\|$$
$$+ 8(r - r^\star)(\delta_4 + \eta_4)\left(1 + 80\alpha\bar{c}_1\lambda_{r^\star}(\bar{Z}^\star)\right)^{2(t-t_\star)}\tau^2.$$

By (53a), there must hold

$$t_1 - t_\star \leq \frac{16}{\alpha\lambda_{r^\star}(\bar{Z}^\star)}\ln\left(\sqrt{\frac{5}{2}}\frac{\sigma_{r^\star}(\bar{X})\sqrt{m}}{\tau}\right). \tag{54}$$

Given that

$$\max\left\{2\sqrt{2}(\delta_{4(r^\star+1)} + m\eta_{4(r^\star+1)}), (\delta_4 + \eta_4)\right\} \leq \frac{\bar{c}_3\kappa^{-4}}{\sqrt{r^\star}},$$

we have that for all $t \in [t_\star, t_1]$

$$\|\mathcal{A}^*\mathcal{A}(\Delta^t) - \mathcal{W}\Delta^t\mathcal{W}\| \leq 10\bar{c}_3 m\kappa^{-4}\|\bar{Z}^\star\| + (r - r^\star)\frac{\bar{c}_3}{\sqrt{r^\star}}\kappa^{-4}\left(1 + \alpha80\bar{c}_1\lambda_{r^\star}(\bar{Z}^\star)\right)^{2(t_1-t_\star)}\tau^2.$$

Assuming $r \leq md$, using (54), with

$$\tau \leq \bar{c}_4\frac{\sigma_{r^\star}(\bar{X})\sqrt{m}}{\min\{r, md\}\kappa^2},$$

and choosing $8c_4^{7/4} \leq 30$, it follows that

$$\|\mathcal{A}^*\mathcal{A}(\Delta^t) - \mathcal{W}\Delta^t\mathcal{W}\| \leq \left(10\bar{c}_3 + 8\bar{c}_3\bar{c}_4^{7/4}\right)m\kappa^{-2}\lambda_{r^\star}(\bar{Z}^\star) \leq 40\bar{c}_3 m\kappa^{-2}\lambda_{r^\star}(\bar{Z}^\star). \tag{55}$$

Under the assumption that $\alpha \leq \bar{c}_5\|\bar{Z}^\star\|^{-1}\kappa^{-4}$ and $\bar{c}_5 < 1/110$, $\bar{c}_2 < 1/110$ and $40\bar{c}_3 < 1/110$, we can invoke Lemma 9(ii), and obtain

$$\sigma_{\min}(V_{Z^\star}^\top U^{t+1}Q^{t+1}) = \sigma_{\min}(V_{Z^\star}^\top U^{t+1}) \geq \sigma_{\min}(V_{Z^\star}^\top U^{t+1}Q^t)$$
$$\geq \left(1 + \frac{\alpha}{4}\lambda_{r^\star}(\bar{Z}^\star) - \frac{\alpha}{m}\sigma_{\min}^2(V_{Z^\star}^\top U^t)\right)\sigma_{\min}(V_{Z^\star}^\top U^t).$$

Because $t \leq t_1$,

$$\sigma_{\min}(V_{Z^\star}^\top U^t) \leq \frac{\sqrt{m}\sigma_{\min}(\bar{X})}{\sqrt{10}},$$

and therefore

$$\sigma_{\min}(V_{Z^\star}^\top U^{t+1}) \geq \left(1 + \frac{3\alpha}{20}\lambda_{r^\star}(\bar{Z}^\star)\right)\sigma_{\min}(V_{Z^\star}^\top U^t).$$

Using the induction hypothesis (53a) there holds

$$\sigma_{\min}\left(V_{Z^\star}^\top U^{t+1}\right) \geq \frac{\tau}{2}\left(1 + \frac{\alpha}{8}\lambda_{r^\star}(\bar{Z}^\star)\right)^{t+1-t_\star}, \tag{56}$$

and thus (53a) holds.

Observe that (56) implies that $V_{Z^\star}^\top U^{t+1}Q^t$ is full rank. Consequently, we can invoke Lemma 9(iii): under $\bar{c}_5 \leq \frac{\bar{c}_3}{40}$, $\frac{\bar{c}_3}{40} < \frac{1}{1200}$, and $\bar{c}_2 < 1/1200$, it holds

$$\|U^{t+1}Q^{t+1,\perp}\| \leq \left(1 - \frac{\alpha}{m}\left(\frac{\|U^tQ^{t,\perp}\|}{2} - 2\|\mathcal{A}^*\mathcal{A}(\Delta^t) - \mathcal{W}\Delta^t\mathcal{W}\|\right) + 9\alpha\|\bar{Z}^\star\|\|(V_{Z^\star}^\perp)^\top V_{U^tQ^t}\|\right)\|U^tQ^{t,\perp}\|$$
$$\overset{(a)}{\leq} \left(1 + 80\alpha\bar{c}_1\lambda_{r^\star}(\bar{Z}^\star)\right)\|U^tQ^{t,\perp}\|,$$

where in (a) we used $80\bar{c}_3 + 9\bar{c}_2 < 80\bar{c}_1$.

Under the induction hypothesis (53b), we then have

$$\|U^{t+1}Q^{t+1,\perp}\| \leq 2\left(1 + \alpha 80\bar{c}_1\lambda_{r^\star}(\bar{Z}^\star)\right)^{t+1-t_\star}\tau,$$

and thus (53b) is established.

Under the induction hypothesis (53c) and (55) we can invoke Lemma 9(i) yielding

$$\|U^{t+1}\| \leq 3\sqrt{m}\|\bar{X}\|.$$

Next, we invoke Lemma 9(iv) under

$$\bar{c}_5 \leq 1/(24 \cdot 800), \quad 40\bar{c}_3 \leq 1/(24 \cdot 800), \quad \text{and} \quad \bar{c}_2 \leq 1/(24 \cdot 800).$$

In fact, since

$$\|U^t Q^{t,\perp}\| \leq 2\sqrt{2}c_4^{7/8}\frac{\sigma_{r^\star}(\bar{X})\sqrt{m}}{r^{7/8}\kappa^{14/8}}, \quad \forall t \in [t_1, t_\star],$$

choosing $2\sqrt{2}c_4^{7/4} \leq 1/(24 \cdot 800)$, all conditions of Lemma 9(iv) are satisfied. Therefore,

$$\|(V_{\bar{Z}^\star}^\perp)^\top V_{U^{t+1}Q^{t+1}}\| \leq \left(1 - \frac{\alpha}{4}\lambda_{r^\star}(\bar{Z}^\star)\right)\|(V_{\bar{Z}^\star}^\perp)^\top V_{U^tQ^t}\|$$
$$+ \frac{100\alpha}{m}\|\mathcal{W}\Delta^t\mathcal{W} - \mathcal{A}^*\mathcal{A}(\Delta^t)\| + \frac{500\alpha^2}{m^2}\|\Delta^t\|^2,$$

whereby (55), (53c) and induction hypothesis (53d) yields

$$\|(V_{\bar{Z}^\star}^\perp)^\top V_{U^{t+1}Q^{t+1}}\| \leq \left(\bar{c}_2(1 - \alpha/4\lambda_{r^\star}(\bar{Z})) + 4000\bar{c}_3\alpha\lambda_{r^\star}(\bar{Z}^\star) + 5000\bar{c}_5\alpha\lambda_{r^\star}(\bar{Z}^\star)\right)\kappa^{-2}.$$

Thus, requesting $\bar{c}_2/4 \geq 4000\bar{c}_3 + 5000\bar{c}_5$ yields

$$\|(V_{\bar{Z}^\star}^\perp)^\top V_{U^{t+1}Q^{t+1}}\| \leq \bar{c}_2\kappa^{-2}$$

concluding the proof of (53).

### B.2.2 Step 2: Refinement phase

Define

$$t_a \triangleq t_1 + \left\lfloor \frac{300}{\alpha\lambda_{r^\star}(\bar{Z}^\star)}\ln\left(\frac{\kappa^{1/4}}{8}\sqrt{\frac{r^\star}{r - r^\star}}\frac{m^{7/8}\|\bar{X}\|^{7/4}}{\tau^{7/4}}\right)\right\rfloor,$$
$$t_b \triangleq \min\{t : (\sqrt{r - r^\star} + 1)\|\Delta_\perp^t\|_F \geq \|\Delta^t\|_F, \ t \geq t_1\},$$
$$t_2 \triangleq \min\{t_a, t_b\}.$$

The proof follows in a similar fashion as Step 1. More specifically, it is shown inductively that

$$\sigma_{\min}(U^tQ^t) \geq \sigma_{\min}(V_{Z^\star}^\top U^t) \geq \frac{\sigma_{r^\star}(\bar{X})\sqrt{m}}{\sqrt{10}}, \tag{57a}$$

$$\|U^tQ^{t,\perp}\| \leq \left(1 + 80\alpha\bar{c}_1\lambda_{r^\star}(\bar{Z}^\star)\right)^{t-t_1}\|U^{t_1}Q^{t_1,\perp}\|, \tag{57b}$$

$$\|U^t\| \leq 3\sqrt{m}\|\bar{X}\| \tag{57c}$$

$$\|(V_{\bar{Z}^\star}^\perp)^\top V_{U^tQ^t}\| \leq \bar{c}_2\kappa^{-2}, \tag{57d}$$

$$\|V_{Z^\star}^\top\Delta^t\|_F \leq 10\sqrt{r^\star}\left(1 - \frac{\alpha}{300}\lambda_{r^\star}(\bar{Z}^\star) + 9\alpha\rho\|\bar{Z}^\star\|\right)^{t-t_1}m\|\bar{Z}^\star\|, \tag{57e}$$

for all $t \in [t_1, t_2]$.

Observe that the above inequalities but (57e) readily hold at $t = t_1$. Thus, we start showing that (57e) holds as well at time $t_1$.

Observe that

$$\|V_{Z^\star}^\top\Delta^{t_1}\|_F \leq \|V_{Z^\star}^\top\Delta_S^{t_1}\|_F + \|V_{Z^\star}^\top\Delta_\perp^{t_1}\|_F.$$

By definition $V_{Z^\star}^\top \Delta_\perp^{t_1} = 0$. Therefore,

$$\|V_{Z^\star}^\top \Delta^{t_1}\|_F \leq \|V_{Z^\star}^\top \Delta_S^{t_1}\|_F \leq \|Z^\star\|_F + \|U^{t_1}Q^{t_1}(U^{t_1}Q^{t_1})^\top\|_F,$$

whereby (57c) at $t = t_1$ (57e) at $t = t_1$ follows.

We prove now (57), following a similar approach as in Step 1. We need to invoke Lemma 9. For this, we proceed similarly, and obtain

$$\|\mathcal{A}^*\mathcal{A}(\Delta^t) - \mathcal{W}\Delta^t\mathcal{W}\| \leq \left(2\sqrt{2}\delta_{4(r^\star+1)} + 2\sqrt{2}m\eta_{4(r^\star+1)}\right)10m\sqrt{r^\star}\|\bar{Z}^\star\|$$
$$+ 2(r - r^\star)(\delta_4 + \eta_4)\|U^t Q^{t,\perp}\|^2.$$

At evaluating $t_1$ the end of Phase II it holds that

$$\|U^{t_1}Q^{t,\perp}\|^2 \leq 8\sigma_{\min}^{1/4}(\bar{X})\tau^{7/4}m^{1/8}.$$

Consequently, under the assumption that

$$\max\{2\sqrt{2}\delta_{4(r^\star+1)} + 2\sqrt{2}m\eta_{4(r^\star+1)}, 2(\delta_4 + \eta_4)\} \leq \frac{\bar{c}_3\kappa^{-4}}{\sqrt{r^\star}}, \tag{58}$$

there holds

$$\|\mathcal{A}^*\mathcal{A}(\Delta^t) - \mathcal{W}\Delta^t\mathcal{W}\| \leq 10m\bar{c}_3\kappa^{-2}\lambda_{r^\star}(\bar{Z}^\star)$$
$$+ \frac{r - r^\star}{\sqrt{r^\star}}\bar{c}_3\kappa^{-4}\left(1 + 80\alpha\bar{c}_1\sigma_{\min}^2(\bar{X})\right)^{t_a - t_1}8\sigma_{\min}^{1/4}(\bar{X})\tau^{7/4}m^{1/8},$$

for all $t \leq t_2$. Given that

$$\frac{1}{8}\kappa^{1/4}\sqrt{\frac{r^\star}{r - r^\star}}\frac{m^{7/8}\|\bar{X}\|^{7/4}}{\tau^{7/4}} > 1,$$

if $\bar{c}_1 \leq \bar{c}_3$, we have

$$(1 + \alpha\bar{c}_1 80\sigma_{\min}^2(\bar{X}))^{t_a - t_1} \leq \left(\frac{\kappa^{1/4}}{8}\sqrt{\frac{r^\star}{r - r^\star}}\frac{m^{7/8}\|\bar{X}\|^{7/4}}{\tau^{7/4}}\right)^{24000\bar{c}_3}.$$

Further requesting $c_3 \leq 1/(48000)$, yields

$$\|\mathcal{A}^*\mathcal{A}(\Delta^t) - \mathcal{W}\Delta^t\mathcal{W}\| \leq 10m\bar{c}_3\kappa^{-2}\lambda_{r^\star}(\bar{Z}^\star)$$
$$+ \frac{r - r^\star}{\sqrt{r^\star}}\bar{c}_3\kappa^{-4}\sqrt{\frac{1}{8}\frac{(r^\star)^{1/4}}{(r - r^\star)^{1/4}}}\kappa^{-6/8}\|\bar{X}\|^{7/8}\sigma_{\min}^{9/8}(\bar{X})m.$$

Using $r \geq 1$ and $r^\star \geq 1$, yields

$$\|\mathcal{A}^*\mathcal{A}(\Delta^t) - \mathcal{W}\Delta^t\mathcal{W}\| \leq 40m\bar{c}_3\kappa^{-2}\sigma_{r^\star}^2(\bar{X}). \tag{59}$$

Given (59) and the induction hypotheses (57c),(57d) we can invoke Lemma 9(ii) and infer

$$\sigma_{\min}(U^{t+1}Q^{t+1}) \geq \sigma_{\min}(V_{Z^\star}^\top U^{t+1}) \geq \sigma_{\min}(V_{Z^\star}^\top)\left(1 + \frac{\alpha}{4}\lambda_{r^\star}(\bar{Z}^\star) - \frac{\alpha}{m}\sigma_{r^\star}^2(V_{Z^\star}^\top U^t)\right).$$

Following [34, Theorem 9.6], where we consider the case in which $\sigma_{\min}(V_{Z^\star}^\top U^{t+1}Q^{t+1}) \geq \frac{1}{2}\sigma_{\min}(X)$ and the complement to yield

$$\sigma_{\min}(V_{Z^\star}^\top U^{t+1}Q^{t+1}) \geq \frac{\sigma_{\min}(\bar{X})\sqrt{m}}{\sqrt{10}}$$

as long as $\bar{c}_5 \leq 1/45$. Observe that this implies that $V_{Z^\star}^\top U^{t+1}Q^{t+1}$ is invertible. Consequently, (57b) can be established in the same way as in Step 1. The same approach applies also to prove (57c) and (57d).

We have left to establish (57e). For this, we invoke Lemmas 3 and 2, and write

$$\|V_{Z^\star}^\top(\mathcal{W}\Delta^t\mathcal{W} - \mathcal{A}^*\mathcal{A}(\Delta^t))\|_F \leq \|V_{Z^\star}^\top(\mathcal{W}\Delta_S^t\mathcal{W} - \mathcal{A}^*\mathcal{A}(\Delta_S^t))\|_F + \|V_{Z^\star}^\top(\mathcal{W}\Delta_\perp^t\mathcal{W} - \mathcal{A}^*\mathcal{A}(\Delta_\perp^t))\|_F$$
$$\leq \left(2\delta_{4(r^\star+1)} + 2\eta_{4(r^\star+1)}\right)\|\Delta_S^t\|_F + (2\delta_4 + 2\eta_4)\|\Delta_\perp^t\|_*,$$

whereby if $t \leq t_2$, it holds

$$(\sqrt{r - r^\star} + 1)\|U^t W^t\|_F^2 \leq \|\Delta^t\|_F.$$

Therefore, under the assumptions on the RIP that we have imposed in (58), it holds

$$\|V_{Z^\star}^\top (\mathcal{W}\Delta^t \mathcal{W} - \mathcal{A}^* \mathcal{A}(\Delta^t))\|_F \leq \frac{2\bar{c}_3}{\kappa^2}\|\Delta^t\|_F.$$

Following a similar procedure we can establish the same bound with the difference in $V_{Z^\star}$ to $V_{U^t W^t}^\top$ or no projection at all. Given that $\bar{c}_3 \leq 1/(2 \cdot 81000)$, $\bar{c}_5 \leq 1/81000$, and $\bar{c}_2 \leq 1/81000$, we can invoke Lemma 9(v), yielding

$$\|V_{Z^\star}^\top \Delta^{t+1}\|_F \leq \left(1 - \frac{\alpha}{200}\lambda_{r^\star}(\bar{Z}^\star) + \alpha 9\|\bar{Z}^\star\|\rho\right)\|V_{Z^\star}^\top \Delta^t\|_F + \frac{\alpha\lambda_{r^\star}(\bar{Z}^\star)}{100}\|U^t Q^{t,\perp}(U^t Q^{t,\perp})^\top\|_F.$$

Using the induction hypothesis (57e) and $\|U^{t_1} Q^{t_1,\perp}\| \leq 8\sigma_{\min}^{1/4}\tau^{7/4}m^{1/8}$, we obtain

$$\|U^t Q^{t,\perp}(U^t Q^{t,\perp})^\top\|_F \leq \sqrt{r - r^\star}\|U^t Q^{t,\perp}\|^2$$
$$\leq \sqrt{r - r^\star}\left(1 + \alpha\bar{c}_1 80\lambda_{r^\star}(\bar{Z}^\star)\right)^{2(t_2 - t_1)} 8\sigma_{\min}^{1/4}(\bar{X})\tau^{7/4}m^{1/8}.$$

This is implied by choosing

$$80\bar{c}_2\bar{c}_5 \leq \frac{\sqrt{400} - \sqrt{399}}{\sqrt{399}};$$

which yields

$$\|V_{Z^\star}^\top \Delta^{t+1}\|_F \leq \left(1 - \frac{\alpha\lambda_{r^\star}(\bar{Z}^\star)}{200} + 9\alpha\|\bar{Z}^\star\|\rho\right)\|V_{Z^\star}^\top \Delta^t\|_F$$
$$+ \frac{\alpha\lambda_{r^\star}(\bar{Z}^\star)}{100}\sqrt{r^\star}\left(1 - \frac{\alpha}{400}\lambda_{r^\star}(\bar{Z}^\star)\right)^{t-t_1} m\|\bar{Z}^\star\|,$$

whereby invoking the induction hypothesis we have

$$\|V_{Z^\star}^\top \Delta^{t+1}\| \leq 10\sqrt{r^\star}m\|\bar{Z}^\star\|\left(1 - \frac{\alpha}{300}\lambda_{r^\star}(\bar{Z}^\star) + 9\alpha\rho\|\bar{Z}^\star\|\right)^{t-t_1} \times$$
$$\times \left(1 - \frac{\alpha}{200}\lambda_{r^\star}(\bar{Z}^\star) + 9\alpha\rho\|\bar{Z}^\star\|\right) + \frac{\alpha\lambda_{r^\star}(\bar{Z}^\star)}{1000}\sqrt{r^\star}m 10\|\bar{Z}^\star\|\left(1 - \frac{\alpha}{400}\lambda_{r^\star}(\bar{Z}^\star)\right)^{t-t_1}.$$

This concludes the proof by induction.

We have now all the ingredients to show that by time $t_2$ the generalization error is small.

From [34, Lemma B.4] it follows that

$$\|U^{t_2}(U^{t_2})^\top - Z^\star\|_F \leq 4\|V_{Z^\star}^\top(Z^\star - U^t(U^t)^\top)\|_F + \|U^{t_2}Q^{t_2,\perp}(U^{t_2}Q^{t_2,\perp})^\top\|_F.$$

To upper bound the RHS, we look separately at the two cases in which $t_2 = t_a$ and $t_2 = t_b$.

We start with $t_2 = t_a$: by setting $\rho \leq \frac{\lambda_{r^\star}(\bar{Z}^\star)}{1200 \cdot 9\|\bar{z}^\star\|}$ we obtain

$$\|\Delta^{t_2}\|_F \leq 40\sqrt{r^\star}\left(1 - \frac{\alpha\lambda_{r^\star}(\bar{Z})}{400}\right)^{t_2 - t_1} m\|\bar{Z}^\star\| + \sqrt{r^\star}\left(1 - \frac{\alpha\lambda_{r^\star}(\bar{Z})}{400}\right)^{t_2 - t_1} m\|\bar{Z}^\star\|,$$

evaluating

$$\|\Delta^{\hat{t}}\|_F \leq 8(r^\star)^{1/8}(r - r^\star)^{3/8}\|\bar{X}\|^{11/16}m^{11/32}\kappa^{-3/16}\tau^{21/16}.$$

The case $t_2 = t_b$: Requesting $\bar{c}_1 \leq 1/(19200)$ and following the procedure in [35, Theorem 9.6, Phase III] yields the same result for the case $t_2 = t_b$.

Summing up the times $t_1$ and $t_2$ yields the following result.

### B.2.3 Summary

We summarize the established result in the following proposition.

**Proposition 3.** *Consider the matrix sensing problem (1), with augmented ground-truth $Z^\star$, with $r^\star + 1 \le r \le md$, and the measurement operator $\bar{\mathcal{A}}$ satisfying the $4(r^\star + 1, \delta)-RIP$, with $\delta \precsim \frac{\kappa^{-4}}{\sqrt{r^\star}}$. Let $\{U^t\}_t$ be the (augmented) sequence generated by Algorithm 3, under the following tuning: (i) the stepsize $\alpha \precsim \kappa^{-4}\|\bar{X}\|^{-2}$, (ii) the gossip matrix is chosen to satisfy Assumption 1, with*

$$\rho \precsim \frac{\kappa^{-2}\delta}{\sqrt{r^\star}m^2\sqrt{m}}.$$

*Then, assume the existence of an iteration count $t_\star$ such that for $\tau > 0$ the following holds*

$$\|U^{t_\star}\| \le 3\sqrt{m}\|\bar{X}\|, \qquad\qquad \sigma_{\min}(U^{t_\star}Q^{t_\star}) \ge \tau,$$
$$\|U^{t_\star}Q^{t_\star,\perp}\| \le 2\tau, \qquad\qquad \|(V_{Z^\star}^\perp)^\top V_{U^{t_\star}Q^{t_\star}}\| \le \bar{c}_2\kappa^{-2},$$

*with*

$$\tau \precsim \frac{\sigma_{r^\star}(\bar{X})\sqrt{m}}{\min\{r, md\}\kappa^2}.$$

*Then, after*

$$t_2 - t_\star \le \frac{2}{\alpha\lambda_{r^\star}(\bar{Z}^\star)}\left(8\ln\left(\sqrt{\frac{5}{2}}\frac{\sigma_{r^\star}(\bar{X})\sqrt{m}}{\tau}\right) + 150\ln\left(\frac{\kappa^{1/4}}{8}\sqrt{\frac{r^\star}{r - r^\star}}\frac{m^{7/8}\|\bar{X}\|^{7/4}}{\tau^{7/4}}\right)\right)$$

*iterations,*

$$\|\Delta^{t_2}\|_F \le 50\left(1 + \sqrt{r - r^\star}\right)(r^\star)^{1/8}(r - r^\star)^{3/8}\|\bar{X}\|^{11/16}m^{11/32}\tau^{21/16}\kappa^{-3/16}$$

**Remark 1.** *The statement in Proposition 3 is made only for $r > r^\star$. The case $r = r^\star$ can be obtained by setting $r = r^\star + 1$ and setting the last column of each $\bar{U}_i^0 = 0$. Observe that this will column remain zero for all $t$, and is thus equivalent to the case of $r = r^\star$.*

We are now ready to establish that Theorem 1 holds. For this we identify the $t_\star$ in Proposition 3 with $t_\star$ in Lemma 8. This implies through Proposition 1 that the time $t_\star$ exists with $\tau = \frac{\mu\beta}{4}$. Consequently, $\sigma$, and $\beta$, $\rho$ and $\mu$ behave as stated and required in Proposition 2, and the statement holds with the same probability.

## C  Convergence of Algorithm 3 in the setting $r \in [r^\star, 2r^\star]$

We provide the counterpart of Theorem 1 in the setting $r \in [r^\star, 2r^\star]$.

**Theorem 2.** *Consider the matrix sensing problem (1), with augmented ground-truth $Z^\star$, under $r^\star \le r \le 2r^\star$, and the measurement operator $\bar{\mathcal{A}}$ satisfying the $4(r^\star + 1, \delta)-RIP$ with $\delta \precsim \kappa^{-4}(r^\star)^{-1/2}$. Let $\{U^t\}_t$ be the (augmented) sequence generated by Algorithm 9, under the following tuning: (i) the stepsize $\alpha \precsim \kappa^{-4}\|\bar{X}\|^{-2}$; (ii) the gossip matrix $W$ chosen to satisfy Assumption 1 with*

$$\rho \precsim \frac{\delta^2}{m^6\kappa^4 r^\star}$$

*and (iii) the initialization $U^0$ is chosen as $U^0 = \mu U$, where $U \in \mathbb{R}^{md\times r}$ has i.i.d. $\mathcal{N}(0, \sqrt{m/r})$ distributed entries, and $\mu$ satisfies*

$$\mu^2 \precsim \min\left\{\frac{\varepsilon^2\sigma_{r^\star}(\bar{X})}{\kappa^6 dr\min\{r, md\}}, \frac{\sqrt{r}}{d\sqrt{d}}\left(\kappa^2\frac{\sqrt{dr}}{\varepsilon}\right)^{-96\kappa^2}\right\}.$$

*Then, after*

$$\hat{t} \precsim \frac{1}{\alpha\sigma_{r^\star}(\bar{X})}\left(\ln\left(\frac{\kappa^2\sqrt{r}d}{\varepsilon}\right) + \ln\left(\frac{\sigma_{r^\star}(\bar{X})}{\varepsilon\mu}\right) + \ln\left(\max\left\{1, \kappa\frac{r^\star}{\max\{1, r - r^\star\}}\right\}\frac{r\|\bar{X}\|}{\mu\varepsilon}\right)\right)$$

*iterations, there holds*

$$\frac{\|Z^\star - U^{\hat{t}}(U^{\hat{t}})\|_F}{\|Z^\star\|} \precsim (\max\{r - r^\star, 1\})^{7/8}(r^\star)^{1/8}\|\bar{X}\|^{-21/16}\mu^{21/16}\left(\kappa^4\frac{dr}{\varepsilon}\right)^{21/16}.$$

*with probability at least $1 - (c_1\varepsilon)^{r-r^\star+1} - \exp(-c_2 r)$ where $c_1 > 0$, $c_2 > 0$ are universal constants.*

Similar to Theorem 1, Theorem 2 can be seen as an extension to the centralized counterpart in [34], with a degradation on the allowed initialization size that stems from effectively having a larger RIP due to the network. Observe that in opposition to Theorem 1 we allow $\hat{t}$ to be increasing with $r$ as per assumption, $r \leq 2r^\star$. Comparing the statements in Theorems 1 and 2 allows us to conclude that overparametrizing does not yield a performance loss in terms of computational complexity but will incur additional communication costs due to the exchange of additional variables.

## C.1 Proof of Theorem 2

**Proposition 4.** *Assume that $2r^\star \geq r \geq r^\star$, and instantiate here the assumptions in Proposition 1. Then, if $U$ is initialized such that each element of $U$ is i.i.d. and distributed according to $\mathcal{N}(0, \sqrt{m}/\sqrt{r})$, with probability at least $1 - (c_1\varepsilon)^{r-r^\star+1} - exp(-c_2 r)$ the statements in Proposition 1 hold with*

$$\mu^2 \leq \frac{(1 + 2\alpha\sqrt{r^\star}\delta_{2(r^\star+1)}\lambda_1(\bar{Z}^\star))}{4(1+\eta)\min\{r, md\}} \min \left\{ \frac{r\sqrt{r}}{2d\sqrt{d}(1 + (\rho r^\star m)^{1/2})} \sigma^{-96\kappa^2} \right.$$
$$\left. \frac{\tilde{c}_3(1 - \delta_\rho)\varepsilon\sqrt{r}}{8c_2\kappa^2 d\sqrt{d}(1 + (\rho r^\star m)^{1/2})} \sigma^{-12\kappa^2} \right\}$$

*with*

$$\sigma \leq 2\kappa^2 \frac{\sqrt{dr}\left((1 + 2\alpha\sqrt{r^\star}\delta_{2(r^\star+1)}\lambda_1(\bar{Z}^\star)) + (\rho^2 + \alpha\rho^{1/2}\Delta)m^{1/2}\right)}{\tilde{c}_3\varepsilon(1 + 2\alpha\sqrt{r^\star}\delta_{2(r^\star+1)}\lambda_1(\bar{Z}^\star))}$$

*and*

$$\sqrt{m}\frac{\varepsilon}{r} \leq \beta \leq 2\sqrt{m}\sigma^2.$$

*Proof.* See Appendix E. $\qquad\square$

Observe that Proposition 5 already accommodates $r > r^\star$. Consequently, we are only missing the choice $r = r^\star$, which we cover in the following Proposition.

**Proposition 5.** *Consider the matrix sensing problem (1), with augmented ground-truth $Z^\star$, with $r = r^\star$ and the measurement operator $\bar{A}$ satisfying the $4(r^\star + 1, \delta) - RIP$ with $\delta \precsim \frac{\kappa^{-4}}{\sqrt{r^\star}}$. Let $\{U^t\}_t$ be the (augmented) sequence generated by Algorithm 3, where $\bar{U}_i^t \in \mathbb{R}^{md \times r^\star+1}$ where w.l.o.g. the last column is set to all zeros for all $i \in [m]$. Under the following tuning: (i) the stepsize $\alpha \precsim \kappa^{-4}\|\bar{X}\|^{-2}$, (ii) the gossip matrix is chosen to satisfy Assumption 1 with*

$$\rho \precsim \frac{\kappa^{-2}\delta}{\sqrt{r^\star}m^2\sqrt{m}}.$$

*Then, choose $\tau > 0$ for which*

$$\|U^{t_\star}\| \leq 3\sqrt{m}\|\bar{X}\|, \qquad\qquad \sigma_{\min}(U^{t_\star}Q^{t_\star}) \geq \tau,$$
$$\|U^{t_\star}Q^{t_\star,\perp}\| \leq 2\tau, \qquad\qquad \|(V_{\bar{Z}^\star}^{\perp})^\top V_{U^{t_\star}Q^{t_\star}}\| \leq \bar{c}_2\kappa^{-2},$$

*and*

$$\tau \precsim \frac{\sigma_{r^\star}(\bar{X})\sqrt{m}}{\min\{r, md\}\kappa^2}.$$

*Then, after*

$$t_2 - t_\star \leq \frac{2}{\alpha\lambda_{r^\star}(\bar{Z}^\star)} \left( 8\ln\left(\sqrt{\frac{5}{2}}\frac{\sigma_{r^\star}(\bar{X})\sqrt{m}}{\tau}\right) + 150\ln\left(\frac{\kappa^{1/4}}{8}\sqrt{\frac{r^\star}{\max\{r - r^\star, 1\}}}\frac{m^{7/8}\|\bar{X}\|^{7/4}}{\tau^{7/4}}\right) \right)$$

*iterations, it holds*

$$\|\Delta^{t_2}\|_F \leq 50\left(1 + \max\{\sqrt{r - r^\star}, 1\}\right)(r^\star)^{1/8}(r - r^\star)^{3/8}\|\bar{X}\|^{11/16}m^{11/32}\tau^{21/16}\kappa^{-3/16}$$

*Proof.* The proof follows by applying Proposition 5 to the iterates defined with the extra column of zeros. This yields an equivalent scheme to using Algorithm (3) with $U^t \in \mathbb{R}^{md \times r^\star}$. $\qquad \square$

We are now ready to establish that Theorem 2 holds. For this we identify the $t_\star$ in Proposition 5 with $t_\star$ in Lemma 8. This implies through Proposition 1 that the time $t_\star$ exists with $\tau = \frac{\mu\beta}{4}$. Consequently, $\sigma$, and $\beta$, $\rho$ and $\mu$ behave as stated and required in Proposition 4, and the statement holds with the same probability.

## D  Proof of Corollary 1

We begin with the following Corollary.

**Corollary 2.** *Under the conditions of Theorem 3, it holds*

$$\frac{\|U^t(U^t) - Z^\star\|_F}{\|Z^\star\|^2} \precsim \left( \max\{(r - r^\star)^{5/8}, 1\}(r^\star)^{1/8} \|\bar{X}\|^{-21/16} \left( \frac{\tau}{\sqrt{m}} \right)^{13/16} \right), \qquad (60)$$

*for any $t \in [T, \hat{t}]$, where*

$$T \succsim \frac{1 - \frac{\sqrt{m}}{\tau}}{\alpha\kappa\|\bar{Z}^\star\| \max\{r - r^\star, 1\} \left( \frac{\tau}{\sqrt{m}} \right)^{1/4}}. \qquad (61)$$

Observe that then, we may choose $\tau = \frac{\mu\beta}{4}$ with $\mu$ and $\beta$ as in Proposition 1 and Lemma 8, yielding the result in Corollary 1. We now proceed to prove Corollary 2. The proof consists of two parts. In **Step I** we start off by the iterates' properties at time $\hat{t}$. We then proceed to establish the result by induction. More specifically, we establish that the alignment $\sigma_{\min}(U^t Q^t)$ remains bounded away from zero while the error $\|U^t Q^{t,\perp}\|$ grows slowly and the error $\|(V_{Z^\star}^\perp)^\top V_{U^t Q^t}\|$ shrinks so as to become $\mathcal{O}(\tau^k)$, for some $k > 0$. This is a critical requirement for **Step II** as the error $\|(V_{Z^\star}^\perp)^\top V_{U^t Q^t}\|$ together with the estimation error are what dictate the *rate of growth* of $\|U^t Q^{t,\perp}\|$ (c.f. Lemma 9 (iii)). Consequently, in **Step II** we part from the iterates' properties at the end of **Step I**, and exploit the fact that both the estimation error and $\|(V_{Z^\star}^\perp)^\top V_{U^t Q^t}\| \leq \mathcal{O}(\tau^k)$ to establish that the rate of growth of the remaining error is $\mathcal{O}(\tau^k)$. This allows us to claim that the estimation error will grow extremely slowly, thus providing with a large window throughout which it is small.

Recall that we may write

$$\|\mathcal{W}\Delta^t\mathcal{W} - \mathcal{A}^*\mathcal{A}(\Delta^t)\| \leq \|\mathcal{W}\Delta_S^t\mathcal{W} - \mathcal{A}^*\mathcal{A}(\Delta_S^t)\| + \|\mathcal{W}\Delta_\perp^t\mathcal{W} - \mathcal{A}^*\mathcal{A}(\Delta_\perp^t)\|.$$

Invoking Lemmas 2 and 3 yields

$$\|\mathcal{W}\Delta^t\mathcal{W} - \mathcal{A}^*\mathcal{A}(\Delta^t)\| \leq \left( 2\delta_{4(r^\star+1)} + \eta_{4(r^\star+1)} \right) \|\Delta_S^t\|_F + (2\delta_4 + \eta_4) \|\Delta_\perp^t\|_*.$$

For convenience we define

$$\Delta \triangleq \max\{2\delta_{4(r^\star+1)} + \eta_{4(r^\star+1)}, 2\delta_4 + \eta_4\}.$$

Under the assumption that

$$\|(V_{Z^\star}^\perp)^\top V_{U^t Q^t}\| \leq \kappa^{-2},$$

$$\sigma_{\min}(V_{Z^\star}^\top U^t Q^t) \geq \frac{\sigma_{r^\star}(\bar{X})\sqrt{m}}{\sqrt{10}},$$

$$\|U^t\| \leq 3\sqrt{m}\|\bar{X}\|$$

hold at time $t$, we can invoke Lemma B.4 in [34] and claim

$$\||U^t(U^t)^\top - Z^\star|\| \leq 4\||V_{Z^\star}(U^t(U^t)^\top - Z^\star)|\| + \||U^t Q^{t,\perp}(U^t Q^{t,\perp})^\top|\|.$$

For further convenience, we define the quantities

$$A_1 \triangleq 50\sqrt{1 + \sqrt{r - r^\star}}(r^\star)^{1/8}(r - r^\star)^{3/8}\|\bar{X}\|^{11/16}m^{11/32}\kappa^{-3/16},$$

$$A_2 \triangleq \|\bar{X}\|^{11/32}m^{11/64} \left( \frac{r^\star}{r - r^\star} \right)^{1/16}.$$

**Step I:** In this phase we prove by induction that the following is true:

$$\|U^t\| \leq 3\sqrt{m}\|\bar{X}\| \tag{62a}$$

$$\sigma_{\min}(U^t Q^t) \geq \frac{\sigma_{r^\star}(\bar{X})\sqrt{m}}{\sqrt{10}} \tag{62b}$$

$$\|(V_{\bar{Z}^\star}^\perp)^\top V_{U^t Q^t}\| \leq \left(1 - \alpha\frac{\lambda_{r^\star}(\bar{Z}^\star)}{4}\right)^{t-t_2} \bar{c}_2\kappa^2 + A_3\tau^{1/2} \tag{62c}$$

$$\|U^t Q^{t,\perp}\| \leq \left(1 + \alpha(1/6)\lambda_{r^\star}(\bar{Z}^\star)\right)^{t-t_2} A_2\tau^{21/32} \tag{62d}$$

$$\|V_{Z^\star}^\top(Z^\star - U^t(U^t)^\top)\|_F \leq \left(1 - \frac{\alpha}{400}\lambda_{r^\star}(\bar{Z}^\star)\right)^{t-t_2} A_1\tau^{21/16} + A_4\tau^{1/2} \tag{62e}$$

for all $t \in [t_2, t_3]$ where

$$t_3 \triangleq t_2 + \left\lfloor \frac{4\log(\sqrt{m}\tau^{-1}(r - r^\star)^{-1})}{16\log(1 + \alpha(1/6)\lambda_{r^\star}(\bar{Z}^\star))} \right\rfloor.$$

Observe that (62) hold at time $t = t_2$. Recall that Lemma 9 requires a bound on $\|\mathcal{W}\Delta^t\mathcal{W} - \mathcal{A}^*\mathcal{A}(\Delta^t)\|$ to invoke most of its consequences. Consequently we start off by establishing that given that (62) holds at time $t$, $\|\mathcal{W}\Delta^t\mathcal{W} - \mathcal{A}^*\mathcal{A}(\Delta^t)\|$ is sufficiently small.

As argued prior,

$$\|\mathcal{W}\Delta^t\mathcal{W} - \mathcal{A}^*\mathcal{A}(\Delta^t)\| \leq 4\Delta\|V_{Z^\star}^\top(Z^\star - U^t(U^t)^\top)\|_F + (4\Delta + 1)\sqrt{r - r^\star}\|U^t Q^{t,\perp}\|^2$$
$$\|\Delta^t\| \leq 4\|V_{Z^\star}^\top(Z^\star - U^t(U^t)^\top)\|_F + \sqrt{r - r^\star}\|U^t Q^{t,\perp}\|^2$$

Invoking the induction hypothesis

$$\|\mathcal{W}\Delta^t\mathcal{W} - \mathcal{A}^*\mathcal{A}(\Delta^t)\| \leq \left(4\Delta A_1 + (4\Delta + 1)A_2^2\tau^{-1/2}m^{1/4}\right)\tau^{21/16} + 4\Delta A_4\tau^{1/2}$$
$$\|\Delta^t\| \leq \left(4\Delta + A_2^2\tau^{-1/2}m^{1/4}\right)\tau^{21/16} + 4A_4\tau^{1/2}.$$

Thus, if

$$\left(4\Delta A_1 + (4\Delta + 1)A_2^2\tau^{-1/2}m^{1/4}\right)\tau^{21/16} + 4\Delta A_4\tau^{1/2} \leq \frac{m\lambda_{r^\star}(\bar{z}^\star)}{24 \cdot 800}$$

the condition on the quantity $\|\mathcal{A}^*\mathcal{A}(\Delta^t) - \mathcal{W}\Delta^t\mathcal{W}\|$ holds for all statements that can be invoked in Lemma 9.

Consequently, we can invoke Lemma 9.(i) yielding

$$\|U^{t+1}\| \leq 3\sqrt{m}\|\bar{X}\|.$$

To invoke Lemma 9.(ii) we require that $\|(V_{\bar{Z}^\star}^\perp)^\top V_{U^t Q^t}\| \leq \frac{\kappa^{-1}}{110}$ which is fulfilled by the induction hypothesis as long as

$$\bar{c}_2\kappa^{-2} + A_3\tau^{1/2} \leq \frac{\kappa^{-2}}{110}.$$

If the above holds, we may invoke Lemma 9.(ii) and following the same argument as in Phase II we obtain

$$\sigma_{\min}(U^{t+1}Q^{t+1}) \geq \frac{\sigma_{r^\star}(\bar{X})\sqrt{m}}{\sqrt{10}}.$$

Next, our goal is to invoke Lemma 9.(iv). For this we require that

$$(1 + \alpha(1/6)\lambda_{r^\star}(\bar{Z}^\star))^{t-t_2} A_2\tau^{21/32} \leq \frac{\sqrt{m}\|\bar{X}\|}{\kappa^2 24 \cdot 800}.$$

Then, we may invoke Lemma 9.(iv) yielding

$$\|(V_{\bar{Z}^\star}^\perp)^\top V_{U^{t+1}Q^{t+1}}\| \leq \left(1 - \alpha \frac{\lambda_{r^\star}(\bar{Z}^\star)}{4}\right)^{t-t_2} \|(V_{\bar{Z}^\star}^\perp)^\top V_{U^t Q^t}\|$$
$$+ \frac{100\alpha}{m}\|\mathcal{W}\Delta^t\mathcal{W} - \mathcal{A}^*\mathcal{A}(\Delta^t)\| + \frac{500\alpha^2}{m^2}\|\Delta^t\|^2$$
$$\leq \left(1 - \alpha \frac{\lambda_{r^\star}(\bar{Z}^\star)}{4}\right)^{t-t_2} \|(V_{\bar{Z}^\star}^\perp)^\top V_{U^{t+1}Q^{t+1}}\|$$
$$+ \frac{100\alpha}{m}\left(\left(4\Delta A_1 + (4\Delta + 1)A_2^2\tau^{-1/2}m^{1/4}\right)\tau^{21/16} + 4\Delta A_4\tau^{1/2}\right)$$
$$+ \frac{\alpha^2 500}{m^2}\left(\left((4\Delta + A_2^2\tau^{-1/2}m^{1/4})\tau^{21/16} + 4A_4\tau^{1/2}\right)^2\right).$$

Applying the induction hypothesis yields

$$\|(V_{\bar{Z}^\star}^\perp)^\top V_{U^{t+1}Q^{t+1}}\| \leq \left(1 - \alpha \frac{\lambda_{r^\star}(\bar{Z}^\star)}{4}\right)^{t+1-t_2} \bar{c}_2\kappa^{-2} + \left(1 - \alpha \frac{\lambda_{r^\star}(\bar{Z}^\star)}{4}\right) A_3\tau^{1/2}$$
$$+ \frac{100\alpha}{m}\left(\left(4\Delta A_1 + (4\Delta + 1)A_2^2\tau^{-1/2}m^{1/4}\right)\tau^{21/16} + 4\Delta A_4\tau^{1/2}\right)$$
$$+ \frac{\alpha^2 500}{m^2}\left(\left((4\Delta + A_2^2\tau^{-1/2}m^{1/4})\tau^{21/16} + 4A_4\tau^{1/2}\right)^2\right),$$

and thus for the bound to hold for $t + 1$ we require that

$$+ \frac{100}{m}\left(\left(4\Delta A_1 + (4\Delta + 1)A_2^2\tau^{-1/2}m^{1/4}\right)\tau^{21/16} + 4\Delta A_4\tau^{1/2}\right)$$
$$+ \frac{\alpha 500}{m^2}\left(\left((4\Delta + A_s^2\tau^{-1/2})\tau^{21/16} + 4A_4\tau^{1/2}\right)^2\right) \leq \frac{\lambda_{r^\star}(\bar{Z}^\star)}{4}A_3\tau^{1/2}.$$

All conditions to invoke Lemma 9.(iii) hold and thus

$$\|U^{t+1}Q^{t+1,\perp}\| \leq$$
$$\left(1 - \frac{\alpha}{2m}\|U^t Q^{t,\perp}\|^2 + 9\alpha\|\bar{Z}^\star\|\|(V_{\bar{Z}^\star}^\perp)^\top V_{U^t Q^t}\| + 2\frac{\alpha}{m}\|\mathcal{W}\Delta^t\mathcal{W} - \mathcal{A}^*\mathcal{A}(\Delta^t)\|\right)\|U^t Q^{t,\perp}\|.$$

Observe that we have established conditions for which $\|(V_{\bar{Z}^\star}^\perp)^\top V_{U^t Q^t}\| \leq \frac{\kappa^{-2}}{110}$ and consequently

$$\|U^{t+1}Q^{t+1,\perp}\| \leq$$
$$\left(1 + 9\alpha\|\bar{Z}^\star\|\frac{\kappa^{-2}}{110} + 2\frac{\alpha}{m}\left(\left(4\Delta A_1 + (4\Delta + 1)A_2^2\tau^{-1/2}m^{1/4}\right)\tau^{21/16} + 4\Delta A_4\tau^{1/2}\right)\right)\|U^t Q^{t,\perp}\|,$$

where we apply the induction hypothesis to yield

$$\|U^{t+1}Q^{t+1,\perp}\| \leq \left(1 + \alpha 80\lambda_{r^\star}(\bar{Z}^\star)\right)^{t-t_2} A_2\tau^{21/32} \times$$
$$\times \left(1 + 9\alpha\|\bar{Z}^\star\|\frac{\kappa^{-2}}{110} + 2\frac{\alpha}{m}\left(\left(4\Delta A_1 + (4\Delta + 1)A_2^2\tau^{-1/2}m^{1/4}\right)\tau^{21/16} + 4\Delta A_4\tau^{1/2}\right)\right)$$

where we request that

$$\frac{9}{110}\kappa^{-2}\|\bar{Z}^\star\| + \frac{2}{m}\left(\left(4\Delta A_1 + (4\Delta + 1)A_2^2\tau^{-1/2}\right)\tau^{21/16} + 4\Delta A_4\tau^{1/2}\right) \leq 1/6\lambda_{r^\star}(\bar{Z}^\star)$$

to establish that the statement holds for $t + 1$. Finally, we have left to bound (62e). For this, we can invoke Lemma 9.(v) yielding

$$\|V_{\bar{Z}^\star}^\top(\Delta^{t+1})\|_F \leq \left(1 - \frac{\alpha}{200}\lambda_{r^\star}(\bar{Z}^\star) + 9\alpha\|\bar{Z}^\star\|\rho\right)\|V_{\bar{Z}^\star}^\top\Delta^t\| + \frac{\alpha\lambda_{r^\star}(\bar{Z}^\star)}{100}\sqrt{r - r^\star}\|U^t Q^{t,\perp}\|$$
$$\leq \left(1 - \frac{\alpha\lambda_{r^\star}(\bar{Z}^\star)}{200} + 9\alpha\|\bar{Z}^\star\|\rho\right)\|V_{\bar{Z}^\star}^\top\Delta^t\|_F + \frac{\alpha\lambda_{r^\star}(\bar{Z}^\star)}{100}A_2^2\tau^{21/16}\tau^{-1/2}m^{1/4}.$$

Invoking the induction hypothesis yields

$$\|V_{Z^\star}^\top \Delta^{t+1}\|_F \leq \left(1 - \frac{\alpha \lambda_{r^\star}(\bar{Z})}{200}\right)^{t-t_2}\left(1 - \frac{\alpha \lambda_{r^\star}(\bar{Z}^\star)}{200} + 9\alpha\|\bar{Z}^\star\|\rho\right)A_1\tau^{21/16}$$

$$+ \left(1 - \frac{\alpha}{200}\lambda_{r^\star}(\bar{Z}^\star) + 9\alpha\|\bar{Z}^\star\|\rho\right)A_4\tau^{1/2} + \frac{\alpha\lambda_{r^\star}(\bar{Z}^\star)}{100}A_2^2\tau^{21/16}\tau^{-1/2}m^{1/4}.$$

Thus, by requesting

$$\left(\frac{1}{200}\lambda_{r^\star}(\bar{Z}^\star) - 9\|\bar{Z}^\star\|\rho\right)A_4\tau^{1/2} \geq \frac{\lambda_{r^\star}(\bar{Z}^\star)}{100}A_2^2\tau^{21/16}\tau^{-1/2}m^{1/4}$$

$$\rho \leq \frac{\lambda_{r^\star}(\bar{Z}^\star)}{3600\|\bar{Z}^\star\|}$$

yields the desired result, and consequently the proof of this part is concluded, by setting $A_4 = 4A_2^2\tau^{5/16}m^{1/4}$, and $A_3 = \frac{\bar{c}_2}{2m^{1/4}}\kappa^{-2}$, with $\bar{c}_2 \leq \frac{1}{220}$, with the assumption that $\tau < 1$. A sufficient condition for all conditions to hold is given by

$$\rho \leq \frac{1}{3600\kappa^{-2}}$$

$$\left(4\Delta A_1\tau^{1/2} + A_2^2m^{1/4}(20\Delta+1)\right)\tau^{6/16} + \frac{5}{m}\alpha\left(4A_1\tau^{1/2} + 17m^{1/2}A_2^2\right)^2\tau^{9/8} \leq \frac{\lambda_{r^\star}(\bar{Z}^\star)m}{\kappa^2 8\cdot 10^4}.$$

**Step II:** We summarize what we obtain from Step I when $t = t_3$.

$$\|U^{t_3}\| \leq 3\sqrt{m}\|\bar{X}\|$$

$$\sigma_{\min}(U^{t_3}Q^{t_3}) \geq \frac{\sigma_{r^\star}(\bar{X})\sqrt{m}}{\sqrt{10}}$$

$$\|(V_{Z^\star}^\perp)^\top V_{U^{t_3}Q^{t_3}}\| \leq \frac{1}{1 - \alpha\frac{\lambda_{r^\star}(\bar{Z}^\star)}{4}}\tau^{1/2}m^{-1/4}\sqrt{r - r^\star} + \frac{\bar{c}_2}{2}\tau^{1/2}m^{-1/4}$$

$$\|U^{t_3}Q^{t_3,\perp}\| \leq \tau^{-1/4}m^{1/8}(r - r^\star)^{-1/4}A_2\tau^{21/32}$$

$$\|V_{Z^\star}^\top(\Delta^{t_3})\| \leq A_1\tau^{21/16} + 4A_2^2\tau^{13/16}$$

From here we establish by induction that

$$\|U^t\| \leq 3\sqrt{m}\|\bar{X}\|$$

$$\sigma_{\min}(U^tQ^t) \geq \frac{\sigma_{r^\star}(\bar{X})\sqrt{m}}{\sqrt{10}}$$

$$\|(V_{Z^\star}^\perp)^\top V_{U^tQ^T}\| \leq \left(1 - \frac{\alpha}{4}\lambda_{r^\star}(\bar{Z}^\star)\right)^{t-t_3}\left(\frac{\tau^{1/2}\sqrt{r-r^\star}}{1-\alpha\frac{\lambda_{r^\star}(\bar{Z}^\star)}{4}} + \frac{\bar{c}_2}{2}\tau^{1/2}\right)m^{-1/4} + C_1\tau^{1/8}$$

$$\|U^tQ^{t,\perp}\| \leq \left(1 + C_2\left(\frac{\tau}{m^{1/2}}\right)^{1/8}\right)^{t-t_3}\frac{A_2m^{1/8}}{(r-r^\star)^{1/4}}\tau^{13/32}$$

$$\|V_{Z^\star}(\Delta^T)\|_F \leq \left(1 - \frac{\alpha\lambda_{r^\star}(\bar{Z}^\star)}{400}\right)^{t-t_3}\left(A_1\tau^{21/16} + 4A_2^2\tau^{13/16}\right) + C_3\tau^{1/8},$$

for all $t \in [t_3, t_4]$

$$t_4 \triangleq t_3 + \left\lfloor\frac{\frac{1}{4}\log\left(\frac{\sqrt{m}}{\tau}\right)}{16\log\left(1 + C_2(\frac{\tau}{\sqrt{m}})^{1/8}\right)}\right\rfloor.$$

We start off in a similar way as in the previous phase. By induction hypothesis we can invoke Lemma B.4 in [34] and thus

$$\|\mathcal{W}\Delta^t\mathcal{W} - \mathcal{A}^*\mathcal{A}(\Delta^t)\| \leq 4\Delta\|V_{Z^\star}^\top(\Delta^t)\|_F + (4\Delta+1)\sqrt{r-r^\star}\|U^tQ^{t,\perp}\|^2$$

$$\|\Delta^t\| \leq \|V_{Z^\star}^\top\Delta^t\|_F + \sqrt{r-r^\star}\|U^tQ^{t,\perp}\|^2,$$

whereby induction hypothesis it follows that

$$\|\mathcal{W}\Delta^t\mathcal{W} - \mathcal{A}^*\mathcal{A}\| \le 4\Delta\left(A_1\tau^{21/16} + 4A_2^2\tau^{13/16} + C_3\tau^{1/8}\right)$$

$$+ (4\Delta + 1)A_2^2\tau^{13/16}\left(1 + C_2\left(\frac{\tau}{\sqrt{m}}\right)^{1/8}\right)^{2(t_4-t_3)} m^{1/4}$$

$$\|\Delta^t\| \le \left(A_1\tau^{21/16} + 4A_2^2\tau^{13/16} + C_3\tau^{1/8}\right) + A_2^2\tau^{13/16}\left(1 + C_2\left(\frac{\tau}{\sqrt{m}}\right)^{1/8}\right)^{2(t_4-t_3)} m^{1/4},$$

whereby using the expression for $t_4$ is that

$$\|\mathcal{W}\Delta^t\mathcal{W} - \mathcal{A}^*\mathcal{A}\| \le 4\Delta\left(A_1\tau^{21/16} + 4A_2^2\tau^{13/16} + C_3\tau^{1/8}\right) + (4\Delta + 1)A_2^2\tau^{25/32}\left(m\right)^{1/64}m^{1/4}$$

$$\|\Delta^t\| \le \left(A_1\tau^{21/16} + 4A_2^2\tau^{13/16} + C_3\tau^{1/8}\right) + A_2^2\tau^{25/32}m^{1/64}m^{1/4}.$$

A requirement for us to invoke elements of Lemma 9 is

$$4\Delta\left(A_1\tau^{21/16} + 4A_2^2\tau^{13/16} + C_3\tau^{1/8}\right) + (4\Delta + 1)A_2^2\tau^{25/32}m^{1/64}m^{1/4} \le \frac{m\lambda_{r^\star}(\bar{Z}^\star)}{24 \cdot 800}.$$

We follow the same steps as in the previous part and invoke Lemma 9.(i) yielding

$$\|U^{t+1}\| \le 3\sqrt{m}\|\bar{X}\|.$$

To invoke Lemma 9.(ii) we request that $\|(V_{\bar{Z}^\star}^\perp)^\top V_{U^t Q^t}\| \le \frac{\kappa^{-2}}{110}$ which is fulfilled by induction hypothesis as long as

$$\frac{\bar{c}_2}{2}\tau^{1/2}m^{-1/4} + \left(\frac{\sqrt{r-r^\star}}{1 - \alpha\frac{\lambda_{r^\star}(\bar{Z}^\star)}{4}}\right)\tau^{1/2}m^{-1/4} + C_1\tau^{1/8} \le \frac{\kappa^{-1}}{100}$$

from which we obtain

$$\sigma_{\min}(U^{t+1}Q^t) \ge \frac{\sigma_{r^\star}(\bar{X})\sqrt{m}}{\sqrt{10}},$$

following the same procedure as in Phase II. Similar as in Step I, to invoke Lemma 9.(iv) we require that

$$\frac{A_2}{(r-r^\star)^{1/4}}\tau^{12/32} \le \frac{\kappa^{-2}m\|\bar{X}\|}{24 \cdot 800}$$

yielding

$$\|(V_{\bar{Z}^\star}^\perp)^\top V_{U^{t+1}Q^{t+1}}\| \le \left(1 - \frac{\alpha}{4}\lambda_{r^\star}(\bar{Z}^\star)\right)\|(V_{\bar{Z}^\star}^\perp)^\top V_{U^t Q^t}\|$$

$$+ \frac{100}{m}\alpha\|\mathcal{W}\Delta^t\mathcal{W} - \mathcal{A}^*\mathcal{A}(\Delta^t)\| + \frac{500\alpha^2}{m^2}\|\Delta^t\|^2.$$

By induction hypothesis we have

$$\|(V_{\bar{Z}^\star}^\perp)^\top V_{U^{t+1}Q^{t+1}}\| \le \left(1 - \frac{\alpha}{4}\lambda_{r^\star}(\bar{Z}^\star)\right)^{t+1-t_3}\left(\frac{\tau^{1/2}\sqrt{r-r^\star}}{1 - \alpha\frac{\lambda_{r^\star}(\bar{Z}^\star)}{4}} + \frac{\bar{c}_2}{2}\tau^{1/2}\right)m^{-1/4} + C_1\tau^{1/8}$$

$$- \frac{\alpha}{4}\lambda_{r^\star}(\bar{Z}^\star)C_1\tau^{1/8} + \frac{500\alpha^2}{m^2}\left(A_1\tau^{21/16} + 4A_2^2\tau^{13/16} + C_3\tau^{1/8} + A_2^2\tau^{25/32}m^{1/64}m^{1/4}\right)^2$$

$$+ \alpha\frac{100}{m}\left(4\Delta\left(A_1\tau^{21/16} + 4A_2^2\tau^{13/16} + C_3\tau^{1/8}\right) + (4\Delta + 1)A_2^2\tau^{25/32}m^{1/64}m^{1/4}\right).$$

Consequently, for the statement to hold for $t+1$ it must hold that

$$\frac{\lambda_{r^\star}(\bar{Z}^\star)}{4}C_1\tau^{1/8} \ge \frac{500\alpha}{m^2}\left(A_1\tau^{21/16} + 4A_2^2\tau^{13/16} + C_3\tau^{1/8} + A_2^2\tau^{25/32}m^{1/64}m^{1/4}\right)^2$$

$$+ \frac{100}{m}\left(4\Delta\left(A_1\tau^{21/16} + 4A_2^2\tau^{13/16} + C_3\tau^{1/8}\right) + (4\Delta + 1)A_2^2\tau^{25/32}m^{1/64}m^{1/4}\right).$$

At this state, all conditions hold to invoke Lemma 9.(iii) yielding

$$\|U^{t+1}Q^{t+1,\perp}\| \leq \left(1 + \frac{2\alpha}{m}\|\mathcal{W}\Delta^{t+1}\mathcal{W} - \mathcal{A}^*\mathcal{A}^*(\Delta^t)\| + 9\alpha\|\bar{Z}^\star\|\|(V_{\bar{Z}^\star}^\perp)^\top V_{U^t Q^t}\|\right)\|U^t Q^{t,\perp}\|,$$

whereby we request that

$$\frac{2\alpha}{m}\left(4\Delta\left(A_1\tau^{21/16} + 4A_2^2\tau^{13/16} + C_3\tau^{1/8}\right) + (4\Delta+1)A_2^2\tau^{25/32}M^{1/64}m^{1/4}\right)$$

$$+ 9\alpha\|\bar{Z}^\star\|\left(\frac{\tau^{1/2}\sqrt{r-r^\star}}{1-\alpha\frac{\lambda_{r^\star}(\bar{Z})}{4}}m^{-1/4} + \frac{\bar{c}_2}{2}\tau^{1/2}m^{-1/4} + C_1\tau^{1/8}\right) \leq C_2\left(\frac{\tau}{\sqrt{m}}\right)^{1/8}$$

to yield

$$\|U^{t+1}Q^{t+1,\perp}\| \leq \left(1 + C_2\left(\frac{\tau}{\sqrt{m}}\right)^{1/8}\right)\|U^t Q^{t,\perp}\|,$$

and thus invoking the induction hypothesiswe obtain the desired result. Finally, we can invoke Lemma 9.(v) to yield

$$\|V_{\bar{Z}^\star}^\top\Delta^{t+1}\|_F \leq \left(1 - \frac{\alpha\lambda_{r^\star}(\bar{Z}^\star)}{200} + \alpha9\|\bar{Z}^\star\|\rho\right)\|V_{\bar{Z}^\star}^\top\Delta^t\|_F + \frac{\alpha\lambda_{r^\star}(\bar{Z}^\star)}{100}\sqrt{r-r^\star}\|U^t Q^{t,\perp}\|^2$$

$$\leq \left(1 - \frac{\alpha\lambda_{r^\star}(\bar{Z}^\star)}{200} + \alpha9\|\bar{Z}^\star\|\rho\right)\|V_{\bar{Z}^\star}^\top\Delta^t\|_F + \frac{\alpha\lambda_{r^\star}(\bar{Z}^\star)}{100}m^{1/64}m^{1/4}A_2^2\tau^{25/32}$$

$$\leq \left(1 - \frac{\alpha\lambda_{r^\star}(\bar{Z}^\star)}{400}\right)^{t-t_3}\left(1 - \frac{\alpha}{200}\lambda_{r^\star}(\bar{Z}^\star) + \alpha9\|\bar{Z}^\star\|\rho\right)\left(A_1\tau^{21/16} + 4A_2^2\tau^{13/16}\right)$$

$$+ C_3\tau^{1/8} - \left(\frac{\alpha}{200}\lambda_{r^\star}(\bar{Z}^\star) - 9\alpha\rho\|\bar{Z}^\star\|\right)C_3\tau^{1/8} + \frac{\alpha\lambda_{r^\star}(\bar{Z}^\star)}{100}m^{1/64}m^{1/4}A_2^2\tau^{25/32}.$$

Thus, requesting that

$$\rho \leq \frac{1}{3600\kappa^2} \qquad\qquad \frac{C_3}{4}\tau^{1/8} \geq m^{1/64}m^{1/4}A_2^2\tau^{25/32},$$

and the proof by induction follows with adequate choices of $C_1$, $C_2$ and $C_3$ which we provide shortly. For convenience define

$$B \triangleq 4(\Delta+1)\left(A_1\left(\frac{\tau}{\sqrt{m}}\right)^{21/16}m^{21/32} + 4A_2^2\left(\frac{\tau}{\sqrt{m}}\right)^{13/16}m^{13/32}\right.$$

$$\left. + C_3 + A_2^2\left(\frac{\tau}{\sqrt{m}}\right)^{25/32}m^{25/62}m^{17/64}\right)$$

Observe that we may choose

$$C_3 = 4m^{17/64}A_2^2\tau^{21/32}$$

$$C_1\tau^{1/8} \leq \frac{\kappa^{-1}}{100} - \frac{\bar{c}_2}{2}\tau^{1/2}m^{-1/4} - \left(\frac{\sqrt{r-r^\star}}{1-\alpha\frac{\lambda_{r^\star}(\bar{Z}^\star)}{4}}\right)\tau^{1/2}m^{-1/4}$$

$$C_1\frac{\lambda_{r^\star}(\bar{Z}^\star)}{4}\tau^{1/8} \geq \frac{500\alpha}{m}B^2 + \frac{100}{m}B$$

$$C_2\left(\frac{\tau}{\sqrt{m}}\right)^{1/8} \geq \frac{2\alpha}{m}(4\Delta+1)B + 9\alpha\|\bar{Z}^\star\|\left(\frac{\sqrt{r-r^\star}}{1-\alpha\frac{\lambda_{r^\star}(\bar{Z}^\star)}{4}} + \frac{\bar{c}_2}{2}\right)\tau^{1/2} + \alpha9\|\bar{Z}^\star\|C_1\tau^{1/2}.$$

Under the assumption that

$$\frac{500\alpha}{m}B \leq 1,$$

we set

$$C_1 = \frac{800}{m^{1/4}\lambda_{r^*}(\bar{Z}^\star)},$$

and

$$C_2 = \alpha \left( \frac{8(\Delta+1)}{m} B\left(\frac{\tau}{\sqrt{m}}\right)^{-1/8} + 9\|\bar{Z}^\star\| \left( \frac{\sqrt{r-r^*}}{1-\alpha\frac{\lambda_{r^*}(\bar{Z}^\star)}{4}} \frac{\bar{c}_2}{2} \right) (\tau/\sqrt{m})^{3/8} + \right.$$

$$\left. 9\|\bar{Z}^\star\| \frac{800}{\lambda_{r^*}(\bar{Z}^\star)} (\tau/\sqrt{m})^{3/8} \right).$$

For some universal constant

$$C_2 \le \alpha c \|\bar{Z}^\star\| \sqrt{r-r^*} (\tau/\sqrt{m})^{3/8} + \alpha c \kappa (\tau/\sqrt{m})^{3/8}$$

$$+ c\alpha \left( (r-r^\star)^{5/8}(r^\star)^{1/8}\|\bar{X}\|^{11/16} \left(\frac{\tau}{\sqrt{m}}\right)^{19/16} \right.$$

$$\left. + \left(\frac{r^\star}{4-r^\star}\right)^{1/8} \|\bar{X}\|^{11/16} \left( \left(\frac{\tau}{\sqrt{m}}\right)^{11/16} + \left(\frac{\tau}{\sqrt{m}}\right)^{21/32} \right) \right)$$

and consequently the window is of length at least

$$\left\lfloor \frac{1}{64} \left( \frac{1-\frac{\sqrt{m}}{\tau}}{C_2 \, (\tau/\sqrt{m})^{1/8}} \right) \right\rfloor$$

Observe that throughout both parts

$$\|\Delta^t\|_F \le c_3 \left( A_1 \tau^{21/16} + 4A_2^2 \tau^{13/16} + 4A_2^2 \tau^{1/4}\tau^{1/8} + A_2^2 \tau^{25/32} m^{1/64} m^{1/4} \right)$$

$$\le c_3 (r-r^\star)^{5/8}(r^\star)^{1/8}\|\bar{X}\|^{11/16} m \left(\frac{\tau}{\sqrt{m}}\right)^{21/16} + c_3 \|\bar{X}\|^{22/32} \left(\frac{r^\star}{r-r^\star}\right)^{1/8} \times$$

$$\times \left( m^{24/32} \left(\frac{\tau}{\sqrt{m}}\right)^{13/16} + \left(\frac{\tau}{\sqrt{m}}\right)^{25/32} m^{47/64} m^{17/64} \right)$$

Consequently,

$$\frac{\|\Delta^t\|_F}{m\|\bar{Z}^\star\|} \le c_3 (r-r^\star)^{5/8} (r^\star)^{1/8}\|\bar{X}\|^{-21/16} \left(\frac{\tau}{\sqrt{m}}\right)^{21/16}$$

$$+ c_3\|\bar{X}\|^{-21/16} \left(\frac{r^\star}{r-r^\star}\right)^{1/8} \left(\frac{\tau}{\sqrt{m}}\right)^{13/16}.$$

# E    Proofs of the Intermediate Results in Phase I (Appendix B.1)

## E.1    Proof of Lemma 5

Let

$$\tilde{E}^i \triangleq \frac{\alpha}{m} \mathcal{A}^* \mathcal{A} \left( U^{i-1} \left(U^{i-1}\right)^\top \right) U^{i-1}.$$

Then, it can be established by induction that for $t \ge 1$

$$\tilde{U}^t - U^t = -E^t = \sum_{i=1}^t \left( \mathcal{W}^2 + \frac{\alpha}{m} \mathcal{A}^* \mathcal{A}(\bar{Z}^\star) \right)^{t-i} \tilde{E}^i$$

Invoking Lemma 3 there holds that

$$\|\tilde{E}^i\| \le \frac{\alpha}{m} \left( 1 + 2\delta_4 + 2\left(\rho + (1+2\delta_4)\hat{\Delta}_4\right) \right) \|U^{i-1}(U^{i-1})^\top\|_* \|U^{i-1}\|$$

$$\le \frac{\alpha}{m} \left( 1 + 2\delta_4 + 2\left(\rho + (1+2\delta_4)\hat{\Delta}_4\right) \right) \|U^{i-1}\|_F^2 \|U^{i-1}\|.$$

Further, observe that

$$\left\| \left(\mathcal{W}^2 + \frac{\alpha}{m}\mathcal{A}^*\mathcal{A}(Z^\star)\right)^{t-i}\tilde{E}^i \right\| = \left\| \left(\mathcal{W}^2 + \frac{\alpha}{m}Z^\star + \frac{\alpha}{m}\left(\mathcal{A}^*\mathcal{A}(Z) - \mathcal{W}\mathcal{Z}^\star\mathcal{W}\right)\right) \right\|^{t-i} \|\tilde{E}^i\|.$$

Invoking Weyl's inequality and Lemma 1 we have

$$\lambda_1\left(\mathcal{W}^2 + \frac{\alpha}{m}\mathcal{A}^*\mathcal{A}(Z^\star)\right) \leq 1 + \alpha\lambda_1(\bar{Z}^\star) + 2\sqrt{r^\star}\alpha\delta_{2(r^\star+1)}\lambda_1(\bar{Z}^\star).$$

Consequently we have

$$\|E^t\| \leq \sum_{i=1}^{t}(1 + \alpha(1 + 2\delta_{2(r^\star+1)})\lambda_1(\bar{Z}^\star))^{t-i}\times$$
$$\times \frac{\alpha}{m}\left(1 + 2\delta_4 + 2\left(\rho + (1+2\delta_4)\hat{\Delta}_4\right)\right)\|U^{i-1}\|_F^2\|U^{i-1}\|.$$

We upper bound

$$\|U^{i-1}\|_F^2\|U^{i-1}\| \leq \min\{r, md\}\|U^{i-1}\|^3 \leq \min\{r, md\}\left(\|\tilde{U}^{i-1}\| + \|\tilde{U}^{i-1} - U^{i-1}\|\right)^3$$
$$\leq 8\min\{r, md\}\|\tilde{U}^{i-1}\|^3 \leq 8\min\{r, md\}\left(1 + \alpha(1 + 2\sqrt{r^\star}\delta_{2(r^\star+1)})\lambda_1(\bar{Z}^\star)\right)^{3(i-2)}\|\tilde{U}^1\|^3.$$

Observe that

$$\|\tilde{U}^1\| \leq \|\mathcal{J}U^0 + \frac{\alpha}{m}\mathcal{A}^*\mathcal{A}(Z^\star)\mathcal{J}U^0\| + \|(\mathcal{W}^2 - \mathcal{J})U^0 + \frac{\alpha}{m}\mathcal{A}^*\mathcal{A}(Z^\star)(\mathcal{I} - \mathcal{J})U^0\|$$
$$\overset{(a)}{\leq} \left(1 + \alpha(1 + 2\sqrt{r^\star}\delta_{2(r^\star+1)})\lambda_1(\bar{Z}^\star)\right)\|\mathcal{J}U^0\| + \left(\rho^2 + \alpha\rho^{1/2}(1 + \delta_{2r^\star}\sqrt{r^\star})\lambda_1(\bar{Z}^\star)\right)\|(\mathcal{I} - \mathcal{J})U^0\|$$
$$\leq \left(1 + \alpha(1 + 2\sqrt{r^\star}\delta_{2(r^\star+1)})\lambda_1(\bar{Z}^\star)\right)\left(\|\mathcal{J}U^0\|^2 + \rho^{1/2}\sqrt{r^\star}\|(\mathcal{I} - \mathcal{J})U^0\|^2\right)$$

where in (a) we invoke Lemmas 1 and 4. Thus,

$$\|U^{i-1}\|_F^2\|U^{i-1}\| \leq 8\min\{r, md\}\left(1 + \alpha(1 + 2\sqrt{r^\star}\delta_{2(r^\star+1)})\lambda_1(\bar{Z}^\star)\right)^{3(i-1)}\times$$
$$\times \left(\|\mathcal{J}U^0\| + \rho^{1/2}\sqrt{r^\star}\|(\mathcal{I} - \mathcal{J})U^0\|\right).$$

Combining we obtain

$$\|E^t\| \leq \sum_{i=1}^{t}\left(1 + \alpha(1 + 2\sqrt{r^\star}\delta_{2(r^\star+1)})\lambda_1(\bar{Z}^\star)\right)^{t+2i-3}\frac{\alpha}{m}\left(1 + 2\delta_4 + 2\left(\rho + (1+2\delta_4)\hat{\Delta}_4\right)\right)\times$$
$$\times 8\min\{r, md\}\left(\|\mathcal{J}U^0\|^2 + \rho^{1/2}\sqrt{r^\star}\|(\mathcal{I} - \mathcal{J})U^0\|\right)^3$$
$$\leq \frac{4m^{-1}}{(1 + 2\sqrt{r^\star}\delta_{2(r^\star+1)})\lambda_1(\bar{Z}^\star)}\mu^3\min\{r, md\}\left(1 + 2\delta_4 + 2\left(\rho + (1+2\delta_4)\hat{\Delta}_4\right)\right)\times$$
$$\times \left(1 + \alpha(1 + 2\sqrt{r^\star}\delta_{2(r^\star+1)})\lambda_1(\bar{Z}^\star)\right)^{3t}\left(\|\mathcal{J}U^0\| + \rho^{1/2}\sqrt{r^\star}\|(\mathcal{I} - \mathcal{J})U^0\|\right)^3.$$

### E.2 Proof of Lemma 6

Denote by $v_1$ the eigenvector corresponding to the largest eigenvalue of $\mathcal{W}^2 + \frac{\alpha}{m}\mathcal{A}^*\mathcal{A}(Z^\star)$. Then,

$$(\tilde{U}^t)^\top v_1 \geq (U^0)^\top\left(\lambda_1\left(\mathcal{W}^2 + \frac{\alpha}{m}\mathcal{A}^*\mathcal{A}(Z^\star)\right)\right)^t v_1.$$

Invoking Weyl's inequality and Lemma 2 yields

$$\lambda_1\left(\mathcal{W}^2 + \frac{\alpha}{m}\mathcal{A}^*\mathcal{A}(Z^\star)\right) \geq \lambda_1\left(\mathcal{W}^2 + \frac{\alpha}{m}Z^\star\right) - \frac{\alpha}{m}\|\mathcal{W}Z^\star\mathcal{W} - \mathcal{A}^*\mathcal{A}(Z^\star)\|$$
$$\geq 1 + \alpha(1 - 2\sqrt{r^\star}\delta_{2(r^\star+1)})\lambda_1(\bar{Z}^\star).$$

Consequently, combined with Lemma 5 we obtain

$$\frac{\|E^t\|}{\|\tilde{U}^t\|} \leq \frac{4m^{-1}}{\lambda_1(\bar{Z}^\star)(1+2\sqrt{r^\star}\delta_{2(r^\star+1)})}\mu^3 \min\{r, md\}\left(1 + 2\delta_4 + 2\left(\rho + (1+2\delta_4)\hat{\Delta}_4\right)\right) \times$$

$$\times \frac{\left(1 + \alpha\lambda_1(\bar{Z}^\star)(1+2\sqrt{r^\star}\delta_{2(r^\star+1)})\right)^{3t}}{\left(1 + \alpha\lambda_1(\bar{Z}^\star)(1-2\sqrt{r^\star}\delta_{2(r^\star+1)})\right)^t}\frac{\left(\|\mathcal{J}U^0\| + \rho^{1/2}\sqrt{r^\star}\|(\mathcal{I}-\mathcal{J})U^0\|\right)^3}{\|(U^0)^\top v_1\|}.$$

Therefore,

$$t^\star \geq \left\lceil \frac{\ln\left(\frac{(1+2\sqrt{r^\star}\delta_{2(r^\star+1)})\lambda_1(\bar{Z}^\star)}{4\left(1+2\delta_4+2\left(\rho+(1+2\delta_4)\hat{\Delta}_4\right)\right)\mu^2\frac{\left(\|\mathcal{J}U+\rho^{1/2}\sqrt{r^\star}\|(\mathcal{I}-\mathcal{J})U\|\right)^3}{m\|U^\top v_1\|}}\right)}{\ln\left(\frac{(1+\alpha\lambda_1(\bar{X}^\star)(1+2\sqrt{r^\star}\delta_{r^\star+1}))^3}{1+\alpha\lambda_1(\bar{X}^\star)(1-2\sqrt{r^\star}\delta_{2(r^\star+1)})}\right)} \right\rceil.$$

### E.3 Proof of Lemma 7

The condition on $\gamma$ implies, invoking Lemma 10 because $\gamma \leq \frac{\kappa^{-2}}{17}$ that

$$\sigma_{r^\star}(M^tU^0 + E^t) \geq (1-\delta_\rho)\mu\sigma_{r^\star}(M^t)\sigma_{\min}(V_{\mathcal{J}M\mathcal{J}}^\top U) - \|E^t\|,$$

and

$$\sigma_{r^\star}(M^tU^0 + E^t) \geq \frac{1-\delta_\rho}{2}\mu\sigma_{r^\star}(M^t)\sigma_{\min}(V_{\mathcal{J}M\mathcal{J}}^\top U).$$

Further,

$$\|(V_{\mathcal{J}M\mathcal{J}}^\perp)^\top V_{U^t}\| \leq \frac{\gamma}{(1-\delta_\rho)(1-\gamma)}$$

and consequently because

$$\|(V_{Z^\star}^\perp)^\top V_{U^t}\| \leq \|(V_{Z^\star}^\perp)^\top V_{\mathcal{J}M\mathcal{J}}\| + \|(V_{\mathcal{J}M\mathcal{J}}^\perp)^\top V_{U^t}\|.$$

To bound $\|(V_{Z^\star}^\perp)^\top V_{\mathcal{J}M\mathcal{J}}\|$ we invoke the Davis-Kahan $\sin(\Theta)$ theorem yielding

$$\|(V_{Z^\star}^\perp)^\top V_{\mathcal{J}M\mathcal{J}}\| \leq \frac{\|Z^\star - \mathcal{J}\mathcal{A}^*\mathcal{A}(Z^\star)\mathcal{J}\|}{m\lambda_{r^\star}(\bar{Z}^\star) - \|Z^\star - \mathcal{J}\mathcal{A}^*\mathcal{A}(Z^\star)\mathcal{J}\|} \overset{(a)}{\leq} \frac{\sqrt{r^\star}\delta_{r^\star}\lambda_1(\bar{Z}^\star)}{\lambda_{r^\star}(\bar{Z}) - \sqrt{r^\star}\delta_{r^\star}\lambda_1(\bar{Z}^\star)},$$

where in (a) have assumed that $\lambda_{r^\star}(\bar{Z}) > \sqrt{r^\star}\lambda_1(\bar{Z}^\star)\delta_{r^\star}$ and used that $\bar{\mathcal{A}}$ fulfills the RIP. Therefore, we conclude

$$\|(V_{Z^\star}^\perp)^\top V_{U^tQ^t}\| \leq \frac{\sqrt{r^\star}\delta_{r^\star}\lambda_1(\bar{Z}^\star)}{\lambda_{r^\star}(\bar{Z}) - \sqrt{r^\star}\delta_{r^\star}\lambda_1(\bar{Z}^\star)} + \frac{\gamma}{(1-\delta_\rho)(1-\gamma)}.$$

Further, from Lemma 11 and our assumption on $\gamma$ it follows that

$$\|U^tQ^{t,\perp}\| \leq \frac{\kappa^{-2}}{8}\mu\sigma_{r^\star}(M^t)\sigma_{\min}(V_{\mathcal{J}M\mathcal{J}}^\top U).$$

Finally, under our conditions on $\delta_{r^\star}$, $\rho$ and $\gamma$, it holds that $\|(V_{Z^\star}^\perp)^\top V_{U^t}\| \leq \frac{1}{8}$ implying that we can invoke Lemma 11 which yields

$$\sigma_{\min}(U^tQ^t) \geq \frac{1}{2}\sigma_{r^\star}(U^t) \geq \frac{1-\delta_\rho}{2}\mu\sigma_{r^\star}(\mathcal{M}^t)\sigma_{\min}(V_{\mathcal{J}}\mathcal{M}\mathcal{J}^\top U) - \|E^t\|.$$

### E.4 Proof of Lemma 8

Our goal is to find conditions under which for some $t$ there holds that for $c < \frac{1}{2}$

$$\gamma_t = \frac{\mu\sigma_{r^\star+1}(M^{t-1})\left(\sigma_{r^\star+1}(M)\|\mathcal{J}U\| + \left(\rho^2 + \alpha\rho^{1/2}\nu\right)\|(\mathcal{I}-\mathcal{J})U\|\right) + \|E^t\|}{(1-\delta_\rho)\mu\sigma_{r^\star}(M^t)\sigma_{\min}(V_{\mathcal{J}M\mathcal{J}}^\top U)} \leq c\kappa^{-2} \quad (63)$$

hold. Sufficient conditions for the above to hold are

$$\mu\sigma_{r^\star+1}(M^{t-1})\left(\sigma_{r^\star+1}(M)\|\mathcal{J}U\|+(\rho^2+\alpha\rho^{1/2}\nu)\|(\mathcal{I}-\mathcal{J})U\|\right) \tag{64}$$

$$\leq\frac{c}{2}\kappa^{-2}(1-\delta_\rho)\mu\sigma_{r^\star}(M^t)\sigma_{\min}(V_{\mathcal{J}M\mathcal{J}}^\top U)$$

$$\|E^t\|\leq\frac{c}{2}\kappa^{-2}(1-\delta_\rho)\mu\sigma_{r^\star}(M^t)\sigma_{\min}(V_{\mathcal{J}M\mathcal{J}}^\top U). \tag{65}$$

Using Weyl's inequality we have

$$\sigma_{r^\star+1}(M)\leq 1+\frac{\alpha}{m}\|\mathcal{W}Z^\star\mathcal{W}-\mathcal{A}^*\mathcal{A}(Z^\star)\|\overset{(a)}{\leq}1+\alpha 2\sqrt{r^\star}\delta_{2(r^\star+1)}\lambda_1(\bar{Z}^\star) \tag{66}$$

$$\sigma_{r^\star}(M)\overset{(a)}{\geq}1+\alpha\lambda_{r^\star}(\bar{Z}^\star)-\alpha 2\sqrt{r^\star}\delta_{2(r^\star+1)}\lambda_1(\bar{Z}^\star), \tag{67}$$

where in (a) we have used Lemma 2. Thus, sufficient conditions for (64) and (69) are

$$\mu\left(1+\alpha 2\sqrt{r^\star}\delta_{2(r^\star+1)}\lambda_1(\bar{Z}^\star)\right)^{t-1}\left((1+\alpha 2\sqrt{r^\star}\delta_{2(r^\star+1)}\lambda_1(\bar{Z}^\star))\|\mathcal{J}U\|\right) \tag{68}$$

$$+\mu\left(1+\alpha 2\sqrt{r^\star}\delta_{2(r^\star+1)}\lambda_1(\bar{Z}^\star)\right)^{t-1}\left(\rho^2+\alpha\rho^{1/2}\nu\right)\|(\mathcal{I}-\mathcal{J})U\|$$

$$\leq\frac{\tilde{c}_3}{2}\kappa^{-2}(1-\delta_\rho)\mu\left(1+\alpha\lambda_{r^\star}(\bar{Z}^\star)-\alpha 2\sqrt{r^\star}\delta_{2(r^\star+1)}\lambda_1(\bar{Z}^\star)\right)^t\sigma_{\min}(V_{\mathcal{J}M\mathcal{J}}^\top U).$$

$$\|E\|^t\leq\frac{\tilde{c}_3}{2}\kappa^{-2}(1-\delta_\rho)\mu\left(1+\alpha\lambda_{r^\star}(\bar{Z}^\star)-\alpha 2\sqrt{r^\star}\delta_{2(r^\star+1)}\lambda_1(\bar{Z}^\star)\right)^t\sigma_{\min}(V_{\mathcal{J}M\mathcal{J}}^\top U). \tag{69}$$

A sufficient condition for (64) is

$$\underbrace{2\kappa^2\frac{(1+\alpha 2\sqrt{r^\star}\delta_{2(r^\star+1)}\lambda_1(\bar{Z}^\star))\|\mathcal{J}U\|+(\rho^2+\alpha\rho^{1/2}\nu)\|(\mathcal{I}-\mathcal{J})U\|}{\tilde{c}_3\sigma_{\min}(V_{\mathcal{J}M\mathcal{J}}^\top U)\left(1+\alpha 2\sqrt{r^\star}\delta_{2(r^\star+1)}\lambda_1(\bar{Z}^\star)\right)}}_{=\sigma}$$

$$\leq\left(\frac{1+\alpha\lambda_{r^\star}(\bar{Z}^\star)-\alpha 2\sqrt{r^\star}\delta_{2(r^\star+1)}\lambda_1(\bar{Z}^\star)}{1+\alpha 2\sqrt{r^\star}\delta_{2(r^\star+1)}\lambda_1(\bar{Z}^\star)}\right)^t.$$

Thus, by definition of $t_\star$ the above holds for all $t\leq t_\star$.

We find next conditions under which (69) hold. Observe that (31) holds for $t\in[t^\star]$. Aiming at using Lemma 5, we find next conditions under which $t_\star\leq t^\star$. To this end, we ask

$$\frac{1}{8}\ln\left(\frac{(1+2\sqrt{r^\star}\delta_{2(r^\star+1)})\lambda_1(\bar{Z}^\star)}{4(1+\eta)\mu^2\frac{\min\{r,md\}(\|\mathcal{J}U\|+\rho^{1/2}\sqrt{r^\star}\|(\mathcal{I}-\mathcal{J})U\|)^3}{m\|U^\top v_1\|}}\right)\geq\ln(\sigma)\times$$

$$\times\underbrace{\frac{\ln\left(1+\alpha(1+2\sqrt{r^\star}\delta_{2(r^\star+1)})\lambda_1(\bar{Z}^\star)\right)+\frac{1}{2}\ln\left(\frac{1+\alpha\lambda_1(\bar{Z}^\star)(1+2\alpha\sqrt{r^\star}\delta_{2(r^\star+1)})}{1+\alpha\lambda_1(\bar{Z}^\star)(1-2\alpha\sqrt{r^\star}\delta_{2(r^\star+1)})}\right)}{\ln\left(\frac{1+\alpha\lambda_{r^\star}(\bar{Z}^\star)-\alpha 2\sqrt{r^\star}\delta_{2(r^\star+1)}\lambda_1(\bar{Z}^\star)}{1+\alpha 2\sqrt{r^\star}\delta_{2(r^\star+1)}\lambda_1(\bar{Z}^\star)}\right)}}_{\triangleq A},$$

where for convenience we defined $\eta\triangleq 2\delta_4+2(\rho+(1+2\delta_4)\hat{\Delta}_4)$. Using $\delta_{2(r^\star+1)}<\frac{11}{51\sqrt{r^\star}}\kappa^{-4}$ and $\alpha\leq\frac{1}{2}\kappa^{-2}\|\bar{Z}^\star\|^{-1}$, we have

$$A\leq\frac{\lambda_1(\bar{Z}^\star)+6\sqrt{r^\star}\delta_{2(r^\star+1)}\lambda_1(\bar{Z}^\star)}{\lambda_{r^\star}(\bar{Z}^\star)-4\sqrt{r^\star}\delta_{2(r^\star+1)}\lambda_1(\bar{Z}^\star)}\left(1+\alpha\lambda_{r^\star}(\bar{Z}^\star)\right)\leq 12\kappa^2.$$

Thus, for $t_\star\leq t^\star$ it suffices that

$$\ln\left(\frac{(1+2\sqrt{r^\star}\delta_{2(r^\star+1)})\lambda_1(\bar{Z}^\star)}{4(1+\eta)\mu^2\frac{\min\{r,md\}(\|\mathcal{J}U\|+\sqrt{r^\star}\rho^{1/2}\|(\mathcal{I}-\mathcal{J})U\|)^3}{m\|U^\top v_1\|}}\right)\geq 96\kappa^2\ln(\sigma),$$

which can be rearranged to yield

$$\mu^2 \leq \frac{m(1 + 2\sqrt{r^\star}\delta_{2(r^\star+1)})\lambda_1(\bar{Z}^\star)\|U^\top v_1\|}{4(1+\eta)\min\{r, md\}\left(\|\mathcal{J}U\| + \sqrt{r^\star}\rho^{1/2}\|(\mathcal{I} - \mathcal{J})U\|\right)^3}(\sigma)^{-96\kappa^2}.$$

We have thus established that up until $t_\star \|E^t\|$ can be bounded by invoking Lemma 5. Thus, we evaluate the upper bound on $\|E^t\|$ from Lemma 5 on the RHS of (69) to establish a sufficient condition for (69) to hold. By evaluating and rearranging we obtain

$$\frac{4\kappa^2}{m(1 + 2\sqrt{r^\star}\delta_{2(r^\star+1)})\lambda_1(\bar{Z}^\star)}\mu^3\min\{r, md\}(1+\eta)\times$$

$$\times \left(1 + \alpha\left(1 + 2\sqrt{r^\star}\delta_{2(r^\star+1)}\right)\lambda_1(\bar{Z}^\star)\right)^{3t}\left(\|\mathcal{J}U\| + \rho^{1/2}\sqrt{r^\star}\|(\mathcal{I} - \mathcal{J})U\|\right)^3$$

$$\leq \frac{c}{2}(1-\delta_\rho)\mu\left(1 + \alpha\lambda_{r^\star}(\bar{Z}^\star) - \alpha 2\sqrt{r^\star}\delta_{2(r^\star+1)}\lambda_1(\bar{Z}^\star)\right)^t\sigma_{\min}(V_{\mathcal{J}M\mathcal{J}}^\top U).$$

A sufficient conditions for the inequality above to hold is

$$\mu^2 \leq \frac{c(1-\delta_\rho)\left(1 + 2\sqrt{r^\star}\delta_{2(r^\star+1)}\right)\lambda_1(\bar{Z}^\star)m\sigma_{\min}(V_{\mathcal{J}M\mathcal{J}}^\top U)}{8\kappa^2\min\{r, md\}(1+\Delta)(\|\mathcal{J}U\| + \rho^{1/2}\sqrt{r^\star}\|(\mathcal{I} - \mathcal{J})U\|)^3}\times$$

$$\times\left(\frac{1 + \alpha\lambda_{r^\star}(\bar{Z}^\star) - 2\alpha\sqrt{r^\star}\delta_{2(r^\star+1)}\lambda_1(\bar{Z}^\star)}{\left(1 + \alpha\left(1 + 2\sqrt{r^\star}\delta_{2(r^\star+1)}\right)\lambda_1(\bar{Z}^\star)\right)^3}\right)^{t_\star},$$

whereby algebra there holds that

$$\left(\frac{1 + \alpha\lambda_{r^\star}(\bar{Z}^\star) - 2\alpha\sqrt{r^\star}\delta_{2(r^\star+1)}\lambda_1(\bar{Z}^\star)}{\left(1 + \alpha\left(1 + 2\sqrt{r^\star}\delta_{2(r^\star+1)}\right)\lambda_1(\bar{Z}^\star)\right)^3}\right)^{t_\star} \geq \exp(-12\kappa^2\ln(\sigma)).$$

Therefore, a sufficient requirement on $\mu^2$ for (69) is

$$\mu^2 \leq \frac{c(1-\delta_\rho)\left(1 + 2\sqrt{r^\star}\delta_{2(r^\star+1)}\right)\lambda_1(\bar{Z}^\star)m\sigma_{\min}(V_{\mathcal{J}M\mathcal{J}}^\top U)}{8\kappa^2\min\{r, md\}(1+\eta)\left(\|\mathcal{J}U\| + \rho^{1/2}\sqrt{r^\star}\|(\mathcal{I} - \mathcal{J})U\|\right)^3}\times(\sigma)^{-12\kappa^2}.$$

### E.5   Proof of Propositions 2 and 4

We start with the part that is common to both propositions and then particularize for each of the two. With probability at least $1 - \mathcal{O}\left(\exp(-c_0 d)\right)$ it holds that

$$\|\mathcal{J}U\| \leq \sqrt{\frac{dm}{r}} \quad \text{and} \quad \rho^{1/2}\|U\| \leq \sqrt{\frac{d}{r}}m\rho^{1/2}.$$

Further, with probability $1 - \mathcal{O}(e^{-r})$ there holds

$$\|U^\top v_1\| \geq \frac{\sqrt{m}}{2}.$$

Finally, observe that $\sigma_{\min}(V_{\mathcal{J}M\mathcal{J}}^\top U) = \frac{1}{\sqrt{m}}\sigma_{\min}(V_{\bar{M}}^\top \bar{U})$ where the elements of $\bar{U}$ are i.i.d. and distributed as $\mathcal{N}(0, 1/\sqrt{r})$, and consequently, we Theorem 8.8 in [34] from which we conclude that for every $\varepsilon > 0$ there holds with probability at least $1 - (c_1\varepsilon)^{r-r^\star+1} - \exp(-c_2 r)$ that

$$\sigma_{\min}(V_{\mathcal{J}M\mathcal{J}}^\top U) \geq \sqrt{m}\varepsilon\frac{\sqrt{r} - \sqrt{r^\star - 1}}{\sqrt{r}} \geq \begin{cases} \sqrt{m}\varepsilon\frac{2-\sqrt{2}}{2} & \text{if } r \geq 2r^\star \\ \sqrt{m}\frac{\varepsilon}{r} & \text{otherwise.} \end{cases}$$

**Case 1 (Proposition 2)** $r \geq 2r^\star$ Setting $\varepsilon = \frac{e^{-1/2}}{c_1}$ there yields with probability at least $1 - \exp(-r/4) - \exp(c_2 r)$ that

$$\sigma_{\min}(V_{\mathcal{J}M\mathcal{J}}^\top U) \geq \sqrt{m}\frac{e^{-1/2}(2 - \sqrt{2})}{2c_2}.$$

Consequently,

$$\mu^2 \leq \frac{(1 + 2\alpha\sqrt{r^\star}\delta_{2(r^\star+1)}\lambda_1(\bar{Z}^\star))}{4(1+\eta)\min\{r, md\}} \min\left\{\frac{m\sqrt{m}}{2m\sqrt{m}\frac{d}{r}\sqrt{\frac{d}{r}}\left(1 + \sqrt{\rho r^\star m}\right)^3}\sigma^{-96\kappa^2},\right.$$

$$\left.\frac{c(1-\delta_\rho)m\sqrt{m}e^{-1/2}(2-\sqrt{2})}{16c_2\kappa^2 m\sqrt{m}\frac{d}{r}\sqrt{\frac{d}{r}}\left(1 + \sqrt{\rho r^\star m}\right)^3}\sigma^{-12\kappa^2}\right\},$$

with

$$\sigma \leq 4c_2\kappa^2\frac{(1 + \alpha 2\sqrt{r^\star}\delta_{2(r^\star+1)}\lambda_1(\bar{Z}^\star))\sqrt{m}\sqrt{\frac{d}{r}} + (\rho^2 + \alpha\rho^{1/2}\nu)\sqrt{\frac{d}{r}}m}{c(1 + \alpha 2\sqrt{r^\star}\delta_{2(r^\star+1)}\lambda_1(\bar{Z}^\star)))\sqrt{m}e^{-1/2}(2-\sqrt{2})},$$

and

$$\frac{e^{-1/2}(2-\sqrt{2})}{2c_2}\sqrt{m} \leq \beta \leq 2\sqrt{m}\sigma^2.$$

**Case 2 (Proposition 4)** $r^\star \leq r \leq 2r^\star$. With probability $1 - (c_1\varepsilon)^{r-r^\star+1} - \exp(-c_2 r)$, it holds

$$\sigma_{\min}(V_{\mathcal{J}M\mathcal{J}}^\top U) \geq \sqrt{m}\frac{\varepsilon}{r}.$$

Consequently,

$$\mu^2 \leq \frac{(1 + \alpha 2\sqrt{r^\star}\delta_{2(r^\star+1)}\lambda_1(\bar{Z}^\star))}{4(1+\eta)\min\{r, md\}} \min\left\{\frac{m\sqrt{m}}{2m\sqrt{m}\frac{d}{r}\sqrt{\frac{d}{r}}(1 + \sqrt{\rho r^\star m})}\sigma^{-96\kappa^2},\right.$$

$$\left.\frac{c(1-\delta_\rho)m\sqrt{m}\varepsilon}{8c_2\kappa^2 m\sqrt{m}\frac{d}{r}\sqrt{\frac{d}{r}}(1 + \sqrt{\rho r^\star m})^3}\sigma^{-12\kappa^2}\right\},$$

with

$$\sigma \leq 2\kappa^2 r\frac{(1 + \alpha 2\sqrt{r^\star}\delta_{2(r^\star+1)}\lambda_1(\bar{Z}^\star))\sqrt{\frac{md}{r}} + (\rho^2 + \alpha\rho^{1/2}\nu)m\sqrt{\frac{d}{r}}}{c(1 + 2\alpha\sqrt{r^\star}\delta_{2(r^\star+1)}\lambda_1(\bar{Z}^\star))\sqrt{m}\varepsilon},$$

and

$$\sqrt{m}\frac{\varepsilon}{r} \leq \beta \leq 2\sqrt{m}\sigma^2.$$

### E.6 Technical lemmas: auxiliary to proving Lemma 7

**Lemma 10.** *Suppose that $\bar{\mathcal{A}}$ satisfies the $(\delta_{2(r^\star+1)}, 2(r^\star+1))-RIP$ property. Then, if*

$$\delta_\rho < 1,$$

*and $W$ satisfies Assumption 1, with $\rho$ such that*

$$\rho^2 \leq \frac{2\delta_{2(r^\star+1)^2}}{4m^5(1 + 2\delta_{2(r^\star+1)})^2},$$

*then,*

$$\sigma_{r^\star}(U^t) \geq (1 - \delta_\rho)\mu\sigma_{r^\star}(\mathcal{M}^t)\sigma_{\min}(V_{\mathcal{J}M\mathcal{J}}^\top U) - \|E^t\|$$

*and*

$$\sigma_{r^\star+1}(U^t) \leq \mu\sigma_{r^\star+1}(\mathcal{M}^{t-1})(1 + \alpha 2\sqrt{r^\star}\delta_{2(r^\star+1)}\lambda_1(\bar{Z}^\star))\|\mathcal{J}U\|$$
$$+ \mu\sigma_{r^\star+1}(\mathcal{M}^{t-1})\left(\rho^2 + \alpha\rho^{1/2}\nu\right)\|(\mathcal{I} - \mathcal{J})U\| + \|E^t\|.$$

*Then, if*

$$A \triangleq \mu\sigma_{r^\star}(\mathcal{M}^t)(1-\delta_\rho)\sigma_{\min}(V_{\mathcal{J}\mathcal{M}\mathcal{J}}^\top U) - \mu\sigma_{r^\star+1}(\mathcal{M}^{t-1})\left(1 + 2\alpha\sqrt{r^\star}\delta_{2(r^\star+1)}\lambda_1(\bar{Z}^\star)\right)\|\mathcal{J}U\|$$

$$- \mu\sigma_{r^\star+1}(\mathcal{M}^{t-1})\left(\rho^2 + \alpha\rho^{1/2}\nu\right)\|(\mathcal{I}-\mathcal{J})U\| - \|E^t\| > 0,$$

*it holds*

$$\left\|\left(V_{\mathcal{J}\mathcal{M}\mathcal{J}}^\perp\right)^\top V_{U^t}\right\| \leq \frac{\mu\sigma_{r^\star+1}(\mathcal{M}^t)\|\mathcal{J}U\| + \mu\sigma_{r^\star+1}(\mathcal{M}^{t-1})\left(\rho^2 + \alpha\rho^{1/2}\nu\right)\|(\mathcal{I}-\mathcal{J})U\| + \|E^t\|}{(1-\delta_\rho)A}.$$

*Proof.* From Weyl's inequality it follows that

$$\sigma_{r^\star}\left(\mathcal{M}^t U^0 + E^t\right) \geq \sigma_{r^\star}(\mathcal{M}^t U^0) - \|E^t\| \geq \sigma_{r^\star}(V_{\mathcal{M}}^\top \mathcal{M}^t U^0) - \|E^t\|,$$

$$\sigma_{r^\star}\left(V_{\mathcal{M}}^\top \mathcal{M}^t U^0\right) \geq \sigma_{\min}(V_{\mathcal{M}}^\top \mathcal{M}^t V_{\mathcal{M}} V_{\mathcal{M}}^\top U^0) \geq \sigma_{r^\star}(\mathcal{M}^t)\sigma_{\min}(V_{\mathcal{M}}^\top U^0).$$

Observe that

$$\sigma_{\min}(V_{\mathcal{M}}^\top U^0) \geq \sigma_{\min}(V_{\mathcal{M}}^\top V_{\mathcal{J}\mathcal{M}\mathcal{J}} V_{\mathcal{J}\mathcal{M}\mathcal{J}}^\top U^0) \geq \sigma_{\min}(V_{\mathcal{M}}^\top V_{\mathcal{J}\mathcal{M}\mathcal{J}})\sigma_{\min}(V_{\mathcal{J}\mathcal{M}\mathcal{J}}^\top U^0).$$

Further,

$$\sigma_{\min}(V_{\mathcal{M}}^\top V_{\mathcal{J}\mathcal{M}\mathcal{J}}) \geq 1 - \|(V_{\mathcal{M}}^\perp)^\top V_{\mathcal{J}\mathcal{M}\mathcal{J}}\|.$$

Invoking the David-Kahan $\sin(\Theta)-$ theorem we have

$$\|(V_{\mathcal{M}}^\perp)^\top V_{\mathcal{J}\mathcal{M}\mathcal{J}}\|_F \leq \frac{\sqrt{r^\star}\frac{\alpha}{m}\|(V_{\mathcal{M}}^\perp)^\top(\mathcal{I}-\mathcal{J})\mathcal{A}^*\mathcal{A}(Z^\star)\mathcal{J}V_{\mathcal{J}\mathcal{M}\mathcal{J}}\|}{\min_{i\in[r^\star],r^\star+1\leq j\leq md}\{|\lambda_i(\mathcal{J}\mathcal{M}\mathcal{J}) - \lambda_j(\mathcal{M})|\}}.$$

Observe that from Weyl's inequality and Lemma 2

$$\lambda_{r^\star}(\mathcal{J}\mathcal{M}\mathcal{J}) \geq 1 + \alpha\lambda_{r^\star}(\bar{Z}^\star) - \alpha\delta_{r^\star}\sqrt{r^\star}\lambda_1(\bar{Z}^\star),$$

$$\lambda_{r^\star+1}(\mathcal{M}) \leq 1 + \alpha 2\delta_{2r^\star}\sqrt{r^\star}\lambda_1(\bar{Z}^\star).$$

Further, invoking Lemma 4 we obtain

$$\|(V_{\mathcal{M}}^\perp)^\top V_{\mathcal{J}\mathcal{M}\mathcal{J}}\|_F \leq \underbrace{\frac{\sqrt{r^\star}\rho^{1/2}\lambda_1(\bar{Z}^\star)(1 + \sqrt{r^\star}\delta_{2r^\star})}{\lambda_{r^\star}(\bar{Z}^\star) - 3\delta_{2(r^\star+1)}\sqrt{r^\star}\lambda_1(\bar{Z}^\star)}}_{\triangleq\delta_\rho}.$$

With the choice of $\rho$ selected by the lemma there holds $\delta_\rho < 1$ and therefore

$$\sigma_{\min}(V_{\mathcal{M}}^\top U^0) \geq (1-\delta_\rho)\sigma_{\min}(V_{\mathcal{J}\mathcal{M}\mathcal{J}}^\top U^0).$$

Consequently, it follows that

$$\sigma_{r^\star}(\mathcal{M}^t U^0 + E^t) \geq (1-\delta_\rho)\mu\sigma_{r^\star}(\mathcal{M}^t)\sigma_{\min}(V_{\mathcal{J}\mathcal{M}\mathcal{J}}^\top U) - \|E^t\|.$$

Observe that

$$\sigma_{r^\star+1}(M^t U^0) \leq \sigma_{r^\star+1}(\mathcal{M}^{t-1}\mathcal{M}U^0) \leq \sigma_{r^\star}(\mathcal{M}^{t-1})\left(\sigma_1(\mathcal{M}\mathcal{J}U) + \sigma_1(\mathcal{M}(\mathcal{I}-\mathcal{J})U^0)\right).$$

Consequently invoking Lemma 4 we obtain

$$\sigma_{r^\star+1}(\mathcal{M}^t U^0 + E^t) \leq \mu\sigma_{r^\star+1}(\mathcal{M}^{t-1})\left(1 + \alpha 2\sqrt{r^\star}\delta_{2(r^\star+1)}\lambda_1(\bar{Z}^\star)\right)\|\mathcal{J}U\|$$

$$+ \mu\sigma_{r^\star+1}(\mathcal{M}^{t-1})\left((\rho^2 + \alpha\rho^{1/2}\lambda_1(\bar{Z}^\star)(1 + \sqrt{r^\star}\delta_{2r^\star}))\|(\mathcal{I}-\mathcal{J})U\|\right) + \|E^t\|.$$

Further, we write

$$\mathcal{M}^t U^0 + E^t = \mu Z^t V_{\mathcal{M}} V_{\mathcal{M}}^\top U + \underbrace{Z^t V_{\mathcal{M}}^\perp (V_{\mathcal{M}}^\perp)^\top U^0 + E^t}_{\triangleq H}.$$

Under the assumption that $V_{\mathcal{M}}^{\top}U$ is of rank $r^{\star}$, $\mathcal{M}^{t}V_{\mathcal{M}}V_{\mathcal{M}}^{\top}U$ is full ranked. Then, $V_{\mathcal{M}}^{\top}V_{\mathcal{M}}\mathcal{M}^{t}V_{\mathcal{M}}V_{\mathcal{M}}^{\top}U$ is of rank $r^{\star}$ and $V_{\mathcal{M}}$ is spanned by the left singular vectors of $Z^{t}V_{\mathcal{M}}V_{\mathcal{M}}^{\top}U$. Under the assumption that

$$A > 0,$$

it follows from the Davis-Kayan $\sin(\Theta)$ theorem that

$$\|(V_{\mathcal{M}}^{\perp})^{\top}V_{U^{t}}\| \leq \frac{\|H\|}{A},$$

whereby using $\sigma_{\min}(V_{\mathcal{M}}^{\top}U) \geq (1 - \delta_{\rho})\sigma_{\min}(V_{\mathcal{J}\mathcal{M}\mathcal{J}}^{\top}U)$ and that

$$\|H\| \leq \mu\|\mathcal{M}^{t}V_{\mathcal{M}}^{\perp}(V_{\mathcal{M}}^{\perp})^{\top}\mathcal{J}\| + \mu\|\mathcal{M}^{t}V_{\mathcal{M}}^{\perp}(V_{\mathcal{M}}^{\perp})^{\top}(\mathcal{I} - \mathcal{J})U\| + \|E^{t}\|$$
$$\leq \mu\sigma_{r^{\star}+1}(\mathcal{M}^{t})\|\mathcal{J}U\| + \mu\|V_{\mathcal{M}}^{\perp}(V_{\mathcal{M}}^{\perp})^{\top}\mathcal{M}^{t}(\mathcal{I} - \mathcal{J})U\| + \|E^{t}\|.$$

In this way,

$$\mu\|V_{\mathcal{M}}^{\perp}(V_{\mathcal{M}}^{\perp})^{\top}\mathcal{M}^{t}(\mathcal{I} - \mathcal{J})U\| \leq \mu\|V_{\mathcal{M}}^{\perp}(V_{\mathcal{M}}^{\perp})^{\top}\mathcal{M}^{t-1}\|\|\mathcal{M}(\mathcal{I} - \mathcal{J})U\|.$$

Invoking Lemma 4, we have

$$\mu\|V_{\mathcal{M}}^{\perp}(V_{\mathcal{M}}^{\perp})^{\top}\mathcal{M}^{t}(\mathcal{I} - \mathcal{J})U\| \leq \mu\sigma_{r^{\star}+1}(\mathcal{M}^{t-1}) \times \left(\rho^{2} + \alpha\rho^{1/2}\lambda_{1}(\bar{Z}^{\star})\left(1 + \sqrt{r^{\star}}\delta_{2r^{\star}}\right)\right)\|(\mathcal{I} - \mathcal{J})U\|,$$

yielding

$$\|(V_{\mathcal{M}}^{\perp})^{\top}V_{U^{t}}\| \leq$$
$$\frac{\mu\sigma_{r^{\star}+1}(\mathcal{M}^{t})\|\mathcal{J}U\| + \mu\sigma_{r^{\star}+1}(\mathcal{M}^{t-1})(\rho^{2} + \alpha\rho^{1/2}\lambda_{1}(\bar{Z}^{\star})(1 + \sqrt{r^{\star}}\delta_{2r^{\star}})\|(\mathcal{I} - \mathcal{J})U\|) + \|E^{t}\|}{A}.$$

Observe that

$$\|(V_{\mathcal{M}}^{\perp})^{\top}V_{U^{t}}\| \geq \|(V_{\mathcal{M}}^{\perp})^{\top}V_{\mathcal{J}\mathcal{M}\mathcal{J}}^{\perp}(V_{\mathcal{J}\mathcal{M}\mathcal{J}}^{\perp})^{\top}V_{U^{t}}\| \geq \sigma_{\min}((V_{\mathcal{M}}^{\perp})^{\top}V_{\mathcal{J}\mathcal{M}\mathcal{J}}^{\perp})\|(V_{\mathcal{J}\mathcal{M}\mathcal{J}}^{\perp})^{\top}V_{U^{t}}\|,$$

where

$$\sigma_{\min}((V_{\mathcal{M}}^{\perp})^{\top}V_{\mathcal{J}\mathcal{M}\mathcal{J}}^{\perp}) \geq 1 - \|(V_{\mathcal{M}}^{\perp})^{\top}V_{\mathcal{J}\mathcal{M}\mathcal{J}}\| \geq 1 - \delta_{\rho},$$

yielding the result. $\qquad\square$

**Lemma 11.** *Assume that $\|(V_{Z^{\star}}^{\perp})^{\top}V_{U^{t}}\| \leq \frac{1}{8}$ for some $t \geq 1$. Then, the following hold:*

$$\sigma_{r^{\star}}(U^{t}Q^{t}) \geq \frac{\sigma_{r^{\star}}(U^{t})}{2},$$
$$\|(V_{Z^{\star}}^{\perp})^{\top}V_{U^{t}Q^{t}}\| \leq 7\|(V_{Z^{\star}}^{\perp})^{\top}V_{U^{t}}\|,$$
$$\|U^{t}Q^{t,\perp}\| \leq 2\sigma_{r^{\star}+1}(U^{t}).$$

The proof of Lemma 11 follows readily from that of Lemma 8.4 in [34].

# F  Proofs of the Intermediate Results in Phase II (Appendix B.2)

## F.1  Proofs of Lemma 9

**(i)** The proof of this statement follows readily from that of Lemma 9.4 in [34] and thus is omitted.

**(ii)**

$$V_{Z^{\star}}^{\top}U^{t+1}Q^{t} = V_{Z^{\star}}^{\top}\left(\mathcal{W}^{2} + \frac{\alpha}{m}\mathcal{A}^{*}\mathcal{A}(Z^{\star} - U^{t}(U^{t})^{\top})\right)U^{t}Q^{t}$$
$$= V_{Z^{\star}}^{\top}\left(\mathcal{W}^{2} + \frac{\alpha}{m}\left(Z^{\star} - \mathcal{W}U^{t}(\mathcal{W}U^{t})^{\top}\right)\right)U^{t}Q^{t}$$
$$+ V_{Z^{\star}}^{\top}\left(\frac{\alpha}{m}(\mathcal{A}^{*}\mathcal{A}(Z^{\star} - U^{t}(U^{t})^{\top}) - \mathcal{W}(Z^{\star} - U^{t}(U^{t})^{\top})\mathcal{W}\right)U^{t}Q^{t}.$$

Observe that

$$V_{Z^\star}^\top \mathcal{W}^2 = \left(\frac{1}{\sqrt{m}}\mathbf{1}_m^\top \otimes V_{\bar{Z}^\star}^\top\right)(W^2 \otimes I_d) = \left(\frac{1}{\sqrt{m}}\mathbf{1}_m^\top \otimes V_{\bar{Z}^\star}^\top\right).$$

Further, $X = \sqrt{m}V_{Z^\star}\Sigma_{\bar{X}}Q_X^\top$ and consequently

$$
\begin{aligned}
V_{Z^\star}^\top U^{t+1}Q^t &= (I + \alpha\Sigma_{\bar{X}}^2)(V_{Z^\star})^\top U^t Q^t - \frac{\alpha}{m}V_{Z^\star}^\top U^t Q^t (Q^t)^\top (\mathcal{W}U^t)^\top U^t Q^t \\
&\quad + V_{Z^\star}^\top \frac{\alpha}{m}\left(\mathcal{A}^*\mathcal{A}(Z^\star - U^t(U^t)^\top) - \mathcal{W}(Z^\star - U^t(U^t)^\top)\mathcal{W}\right)U^t Q^t \\
&= (I + \alpha\Sigma_{\bar{X}}^2)V_{Z^\star}^\top U^t Q^t \left(I - \frac{\alpha}{m}(Q^t)^\top (U^t)^\top V_{Z^\star}V_{Z^\star}^\top U^t Q^t\right) \\
&\quad - \frac{\alpha}{m}\underbrace{V_{Z^\star}^\top U^t(U^t)^\top \mathcal{W}\left(V_{Z^\star}^\perp(V_{Z^\star}^\perp)^\top\right)U^t Q^t}_{\triangleq A_1} \\
&\quad + \frac{\alpha}{m}\underbrace{V_{Z^\star}^\top \left(\mathcal{A}^*\mathcal{A}(Z^\star - U^t(U^t)^\top) - \mathcal{W}(Z^\star - U^t(U^t)^\top)\mathcal{W}\right)U^t Q^t}_{\triangleq A_2} \\
&\quad + \frac{\alpha^2}{m}\underbrace{\Sigma_{\bar{X}}^2 V_{Z^\star}^\top U^t(U^t)^\top V_{Z^\star}V_{Z^\star}^\top U^t Q^t}_{\triangleq A_3}.
\end{aligned}
$$

We now rewrite each $A_i$ as

$$A_i = P_i V_{Z^\star}^\top U^t Q^t \left(I - \frac{\alpha}{m}(Q^t)^\top (U^t)^\top V_{Z^\star}V_{Z^\star}^\top U^t Q^t\right).$$

In order to proceed with $A_1$ we focus on the term $(V_{Z^\star}^\perp)^\top U^t Q^t$. Under the assumption that $V_{Z^\star}^\top U^t$ is of rank $r^\star$ we have

$$(V_{Z^\star}^\perp)^\top U^t Q^t = (V_{Z^\star}^\perp)^\top U^t Q^t \left(V_{Z^\star}^\top U^t Q^t\right)^{-1}(V_{Z^\star}^\top U^t Q^t).$$

Using the singular value decomposition $U^t Q^t = V_{U^t Q^t}\Sigma_{U^t Q^t}Q_{U^t Q^t}^\top$, yields

$$
\begin{aligned}
(V_{Z^\star}^\perp)^\top U^t Q^t &= (V_{Z^\star}^\perp)^\top V_{U^t Q^t}\Sigma_{U^t Q^t}Q_{U^t Q^t}^\top \left(V_{Z^\star}^\top V_{U^t Q^t}\Sigma_{U^t Q^t}Q_{U^t Q^t}^\top\right)^{-1}(V_{Z^\star}^\top V_{U^t Q^t}\Sigma_{U^t Q^t}Q_{U^t Q^t}^\top) \\
&= (V_{Z^\star}^\perp)^\top V_{U^t Q^t}\left(V_{Z^\star}^\top V_{U^t Q^t}\right)^{-1}V_{Z^\star}^\top U^t Q^t.
\end{aligned}
$$

Therefore, we write $A_1$

$$V_{Z^\star}^\top U^t(U^t)^\top \mathcal{W}(V_{Z^\star}^\perp(V_{Z^\star}^\perp)^\top)U^t Q^t = V_{Z^\star}^\top U^t(U^t)^\top \mathcal{W}V_{Z^\star}^\perp(V_{Z^\star}^\perp)^\top V_{U^t Q^t}(V_{Z^\star}^\top V_{U^t Q^t})^{-1}V_{Z^\star}^\top U^t Q^t$$

post-multiplying by $\left(I - \frac{\alpha}{m}(U^t Q^t)^\top V_{Z^\star}V_{Z^\star}^\top U^t Q^t\right)^{-1}$ and its inverse (a sufficient condition for the inverse to exist is given by $\alpha 9\|\bar{X}\|^2 \le 1$ which holds under the assumption on $\alpha$)

$$
\begin{aligned}
P_1 = {}&V_{Z^\star}^\top U^t(U^t)^\top \mathcal{W}(V_{Z^\star}^\perp(V_{Z^\star}^\perp)^\top)V_{U^t Q^t}(V_{Z^\star}^\top V_{U^t Q^t})^{-1}\times \\
&\left(I - \frac{\alpha}{m}V_{Z^\star}^\top U^t Q^t (U^t Q^t)^\top V_{Z^\star}\right)V_{Z^\star}^\top U^t Q^t.
\end{aligned}
$$

We proceed now similarly with $A_2$ yielding

$$A_2 = \frac{\alpha}{m}V_{Z^\star}^\top \left(\mathcal{A}^*\mathcal{A}(Z^\star - U^t(U^t)^\top) - \mathcal{W}(Z^\star - U^t(U^t)^\top)\mathcal{W}\right)V_{U^t Q^t}(V_{Z^\star}^\top U^t Q^t)^{-1}V_{Z^\star}^\top U^t Q^t$$

following the steps as for the case of $A_1$ we have

$$
\begin{aligned}
P_2 = {}&V_{Z^\star}^\top \left(\mathcal{A}^*\mathcal{A}(Z^\star - U^t(U^t)^\top) - \mathcal{W}(Z^\star - U^t(U^t)^\top)\mathcal{W}\right)V_{U^t Q^t}(V_{Z^\star}^\top V_{U^t Q^t})^{-1}\times \\
&\times \left(I - \frac{\alpha}{m}V_{Z^\star}^\top U^t Q^t (U^t Q^t)^\top V_{Z^\star}\right)^{-1}.
\end{aligned}
$$

Similarly,

$$
\begin{aligned}
A_3 = {}&\underbrace{\Sigma_{\bar{X}}^2 V_{Z^\star}^\top U^t Q^t \left(I - \frac{\alpha}{m}(Q^t)^\top (U^t)^\top V_{Z^\star}V_{Z^\star}^\top U^t Q^t\right)^{-1}(Q^t)^\top (U^t)^\top V_{Z^\star}}_{\triangleq P_3} \times \\
&\times V_{Z^\star}^\top U^t Q^t \left(I - \frac{\alpha}{m}(Q^t)^\top (U^t)^\top V_{Z^\star}V_{Z^\star}^\top U^t Q^t\right).
\end{aligned}
$$

Consequently, under the assumption that

$$\alpha \leq \frac{1}{8100}\|\bar{Z}^\star\|^{-1}\kappa^{-2} \quad \text{and} \quad \|U^t\| \leq 3\sqrt{m}\|\bar{X}\|$$

and using Weyl's inequality we have

$$\sigma_{\min}(V_{Z^\star}^\top U^{t+1}Q^t) \geq \sigma_{\min}\left(I + \alpha\Sigma_{\bar{X}}^2 - \frac{\alpha}{m}P_1 + \frac{\alpha}{m}P_2 + \frac{\alpha^2}{m}P_3\right) \times$$

$$= \sigma_{\min}\left(I + \alpha\Sigma_{\bar{X}}^2 - \frac{\alpha}{m}P_1 + \frac{\alpha}{m}P_2 + \frac{\alpha^2}{m}P_3\right)\sigma_{\min}(V_{Z^\star}^\top U^tQ^t)\left(1 - \frac{\alpha}{m}\sigma_{\min}^2(V_{Z^\star}^\top U^tQ^t)\right)$$

$$\geq \left(1 + \alpha\lambda_{r^\star}(\bar{Z}^\star) - \frac{\alpha}{m}\|P_1\| - \frac{\alpha}{m}\|P_2\| - \frac{\alpha^2}{m}\|P_3\|\right)\sigma_{\min}(V_{Z^\star}^\top U^t)\left(1 - \frac{\alpha}{m}\sigma_{\min}^2(V_{Z^\star}^\top U^t)\right).$$

We now bound each of the quantities $\|P_i\|$ with

$$\|P_1\| = \left\|V_{Z^\star}^\top U^tQ^t(U^tQ^t)^\top \mathcal{W}\left(V_{Z^\star}^\perp(V_{Z^\star}^\perp)^\top\right)V_{U^tQ^t}\left(V_{Z^\star}^\top V_{U^tQ^t}\right)^{-1}\left(I - \frac{\alpha}{m}V_{Z^\star}^\top U^t(U^t)^\top V_{Z^\star}\right)^{-1}\right\|.$$

By assumption $\|(V_{Z^\star}^\perp)^\top V_{U^tQ^t}\| \leq c_1\kappa^{-1}$ implying $\sigma_{\min}((V_{Z^\star}^\perp)^\top V_{U^tQ^t}) \geq 1 - c_1\kappa^{-1}$ and consequently observing that $Z^\star\mathcal{W} = \mathcal{W}Z^\star$

$$\|(V_{Z^\star}^\perp)^\top \mathcal{W}V_{U^tQ^t}\| \leq \|(V_{Z^\star}^\perp)^\top V_{U^tQ^t}\|$$

there holds

$$\|P_1\| \leq \frac{\|U^tQ^t\|^2\|V_{Z^\star}^\top V_{U^tQ^t}\|\|V_{Z^\star}^\top \mathcal{W}V_{U^tQ^t}\|}{\sigma_{\min}(V_{Z^\star}^\top V_{U^tQ^t})\left(1 - \frac{\alpha}{m}\|V_{Z^\star}^\top U^t\|^2\right)} \leq \frac{\|U^tQ^t\|^2\|(V_{Z^\star}^\perp)^\top V_{U^tQ^t}\|^2}{(1 - c_1\kappa^{-1})(1 - \frac{\alpha}{m}9\|\bar{X}\|^2)}$$

$$\leq \frac{\|U^tQ^t\|^2\|(V_{Z^\star}^\perp)^\top V_{U^tQ^t}\|^2}{(1 - c_1\kappa^{-1})(1 - 9\kappa^{-2}(1/81000))} \leq 4\|U^tQ^t\|^2\|(V_{Z^\star}^\perp)^\top V_{U^tQ^t}\|^2$$

$$\leq 4\|U^tQ^t\|^2 c_1\kappa^{-1} \leq 36m\|\bar{X}\|^2 c_1\kappa^{-2}.$$

Similarly,

$$\|P_2\| \leq \frac{\|\mathcal{A}^\ast\mathcal{A}(Z^\star - U^t(U^t)^\top) - \mathcal{W}(Z^\star - U^t(U^t)^\top)\mathcal{W}\|}{\sigma_{\min}(V_{Z^\star}^\top V_{U^tQ^t})\left(1 - \frac{\alpha}{m}\|V_{Z^\star}^\top U^t\|^2\right)}$$

$$\leq \frac{\|\mathcal{A}^\ast\mathcal{A}(Z^\star - U^t(U^t)^\top) - \mathcal{W}(Z^\star - U^t(U^t)^\top)\mathcal{W}\|}{(1 - c\kappa^{-1})(1 - 9\kappa^{-2}1/81000)}$$

$$\leq 4\|\mathcal{A}^\ast\mathcal{A}(Z^\star - U^t(U^t)^\top) - \mathcal{W}(Z^\star - U^t(U^t)^\top)\mathcal{W}\|$$

and

$$\|P_3\| \leq \frac{\|\bar{X}\|^2\|V_{Z^\star}^\top U^tQ^t\|^2}{1 - \frac{\alpha}{m}\|V_{Z^\star}^\top U^tQ^t\|^2} \leq \frac{\|\bar{X}\|^2\|V_{Z^\star}^\top U^tQ^t\|^2}{1 - \frac{\alpha}{9}\|\bar{X}\|^2} \leq \frac{\|\bar{X}\|^2\|V_{Z^\star}^\top U^tQ^t\|^2}{1 - 9(1/81000)\kappa^{-1}},$$

under the assumption that $\alpha$ there holds

$$\|P_3\| \leq 2\|\bar{X}\|^2\|V_{Z^\star}^\top U^tQ^t\|^2 \leq 18m\|\bar{X}\|^4.$$

Combining all estimates together with $\|U^tQ^t\|^2 \leq 9m\|\bar{X}\|^2$ yields

$$\sigma_{\min}(V_{Z^\star}^\top U^{t+1}Q^t) \geq \left(1 + \alpha\lambda_{r^\star}(\bar{Z}^\star) - \alpha36c_1\kappa^{-1}\|\bar{X}\|^2 - \alpha^2 18\|\bar{X}\|^4\right.$$

$$\left. - \frac{\alpha}{m}4\|\mathcal{A}^\ast\mathcal{A}(Z^\star - U^t(U^t)^\top) - \mathcal{W}(Z^\star - U^t(U^t)^\top)\mathcal{W}\|\right) \times$$

$$\times \sigma_{\min}(V_{Z^\star}^\top U^t)\left(1 - \frac{\alpha}{m}\sigma_{\min}^2(V_{Z^\star}^\top U^t)\right).$$

Observe that $\alpha \leq \frac{\lambda_{r^\star}(\bar{Z}^\star)}{81000\|\bar{X}\|^2}\|\bar{X}\|^{-2}$ and therefore

$$\sigma_{\min}(V_{Z^\star}^\top U^{t+1}Q^t) \geq \left(1 + \alpha\lambda_{r^\star}(\bar{Z}^\star) - \alpha c_1 36\kappa^{-2}\|\bar{X}\|^2 - \alpha\frac{18}{81000}\lambda_{r^\star}(\bar{Z}^\star)\right.$$

$$\left. - \frac{\alpha}{m}4\|\mathcal{A}^\ast\mathcal{A}(Z^\star - U^t(U^t)^\top) - \mathcal{W}(Z^\star - U^t(U^t)^\top)\mathcal{W}\|\right)\sigma_{\min}(V_{Z^\star}^\top U^t)\left(1 - \frac{\alpha}{m}\sigma_{\min}^2(V_{Z^\star}^\top U^t)\right).$$

Therefore

$$\sigma_{\min}(V_{Z^\star}^\top U^{t+1}Q^t) \geq \left(1 + \alpha\left(1 - 55c_1\right)\sigma_{\min}^2(\bar{X})\right)\sigma_{\min}(V_{Z^\star}^\top U^t)\left(1 - \frac{\alpha}{m}\sigma_{\min}^2(V_{Z^\star}^\top U^t)\right).$$

Under the assumption that $c_1 \leq \frac{1}{110}$, there holds

$$\sigma_{\min}(V_{Z^\star}^\top U^{t+1}Q^t) \geq \left(1 + \frac{\alpha}{2}\sigma_{\min}^2(\bar{X})\right)\sigma_{\min}(V_{Z^\star}^\top U^t)\left(1 - \frac{\alpha}{m}\sigma_{\min}^2(V_{Z^\star}^\top U^t)\right)$$

$$= \sigma_{\min}(V_{Z^\star}^\top U^t)\left(1 + \frac{\alpha}{2}\sigma_{\min}^2(\bar{X})\left(1 - \frac{\alpha}{m}\sigma_{\min}^2(V_{Z^\star}^\top U^t)\right) - \frac{\alpha}{m}\sigma_{\min}^2(V_{Z^\star}^\top U^t)\right)$$

$$\geq \sigma_{\min}\left(V_{Z^\star}^\top U^t\right)\left(1 + \frac{\alpha}{4}\sigma_{\min}^2(\bar{X}) - \frac{\alpha}{m}\sigma_{\min}^2(V_{Z^\star}^\top U^t)\right).$$

where we have used that

$$1 - \frac{\alpha}{m}\sigma_{\min}^2(V_{Z^\star}^\top U^t) \geq 1 - \frac{\kappa^{-1}\|\bar{X}\|^{-2}}{81000m}9m\|\bar{X}\|^2 \geq \frac{1}{2}.$$

The final result follows by noticing that $\sigma_{\min}(V_{Z^\star}^\top U^{t+1}) \geq \sigma_{\min}(V_{Z^\star}^\top U^{t+1}Q^t)$.

**(iii)** The proof of this statement is practically identical to that of [34, Lemma 9.2], and thus is omitted. However, [34, Lemma 9.2] builds upon [34, Lemma B.1] of which we provide a suitable version for our purpose in Lemma 12.

**(iv)** For convenience, define

$$U^{t+1}Q^{t+1} = \left(\mathcal{W}^2 + \frac{\alpha}{m}\mathcal{M}\right)U^tQ^{t+1} \tag{70}$$

$$= \left(\mathcal{W}^2 + \frac{\alpha}{m}\mathcal{M}\right)\left(V_{U^tQ^t}V_{U^tQ^t}^\top U^tQ^t(Q^t)^\top Q^{t+1} + U^tQ^{t,\perp}(Q^{t,\perp})^\top Q^{t+1}\right). \tag{71}$$

Due to the assumption on $\|U^tQ^{t,\perp}\|$ and because $(Q^t)^\top Q^{t+1}$ is invertible which follows from Lemma 13, we have that the matrix $V_{U^tQ^t}^\top U^tQ^t(Q^t)^\top Q^{t+1}$ is invertible and we may consequently write

$$U^tQ^{t,\perp}(Q^{t,\perp})^\top Q^{t+1}$$

$$= \underbrace{U^tQ^{t,\perp}(Q^{t,\perp})^\top Q^{t+1}\left(V_{U^tQ^t}^\top U^tQ^t(Q^t)^\top Q^{t+1}\right)^{-1}V_{U^tQ^t}^\top}_{\triangleq P} \times$$

$$\times V_{U^tQ^t}V_{U^tQ^t}^\top U^tQ^t(Q^t)^\top Q^{t+1}.$$

Therefore, we have

$$U^{t+1}Q^{t+1} = \left(\mathcal{W}^2 + \frac{\alpha}{m}\mathcal{M}\right)(I + P)V_{U^tQ^t}V_{U^tQ^t}V_{U^tQ^t}^\top U^tQ^t(Q^t)^\top Q^{t+1}.$$

Define the matrices

$$Z' \triangleq \left(\mathcal{W}^2 + \frac{\alpha}{m}\mathcal{M}\right)(I + P)V_{U^tQ^t}$$

$$Z \triangleq \left(I + \frac{\alpha}{m}\mathcal{M}\right)(I + P)V_{U^tQ^t}.$$

Because the matrix $U^tQ^t$ is full rank, there holds $\|(V_{Z^\star}^\perp)^\top V_{U^{t+1}Q^{t+1}}\| = \|(V_{Z^\star}^\perp)^\top V_{Z'}\|$. We now argue that $\mathrm{span}(V_Z') \subseteq \mathrm{span}(V_Z)$ and consequently $\|(V_{Z^\star}^\perp)^\top V_{Z'}\| \leq \|(V_{Z^\star}^\perp)^\top V_Z\|$. We observe that the main deviating factor between the matrices $Z$ and $Z'$ are the terms $\mathcal{W}^2 + \frac{\alpha}{m}\mathcal{M}$ and $I + \frac{\alpha}{m}\mathcal{M}$ respectively. It suffices then to establish that $\mathrm{span}(\mathcal{W}^2 + \frac{\alpha}{m}\mathcal{M}) \subseteq \mathrm{span}(I + \frac{\alpha}{m}\mathcal{M})$ as the term $(I + P)V_{U^tQ^t}$ is common to both matrices. Observe that because $\mathcal{M}$ is symmetric by construction, it suffices to choose $\alpha$ sufficiently small such that

$$\left\|\frac{\alpha}{m}\mathcal{M}\right\| \leq \frac{1}{2} \tag{72}$$

for $I + \frac{\alpha}{m}\mathcal{M} \succeq \frac{1}{2}I$. This implies that $\mathrm{span}(I + \frac{\alpha}{m}\mathcal{M}) = \mathbb{R}^{md}$. This implies that

$$\|(V_{Z^\star}^\perp)^\top V_{U^{t+1}Q^{t+1}}\| \leq \|(V_{Z^\star}^\perp)^\top V_Z\| = \|(V_{Z^\star}^\perp)^\top V_Z Q_Z^\top\| = \|V_\perp^\top Z(Z^\top Z)^{-1/2}\|. \tag{73}$$

From here we proceed similarly to the proof of [34, Lemma 9.3] with the following definitions of $B_i$

$$B_1 \triangleq \frac{\alpha}{m}\left(XX^\top - \mathcal{W}U^t(\mathcal{W}U^t)^\top\right), \tag{74}$$

$$B_2 \triangleq \frac{\alpha}{m}\left(\mathcal{A}^*\mathcal{A}(Z^\star - U^t(U^t)^\top) - \mathcal{W}(Z^\star - U^t(U^t)^\top)\mathcal{W}\right), \tag{75}$$

$$B_3 \triangleq U^tQ^{t,\perp}(Q^{t,\perp})^\top Q^{t+1}\left(V_{U^tQ^t}^\top U^tQ^t(Q^t)^\top Q^{t+1}\right)^{-1}V_{U^tQ^t}^\top, \tag{76}$$

$$B_4 \triangleq \frac{\alpha}{m}\mathcal{M}P, \tag{77}$$

$$B \triangleq \frac{\alpha}{m}\mathcal{M} + P + \frac{\alpha}{m}\mathcal{M}P. \tag{78}$$

Suppose the following holds (a fact that will be proved later)

$$-\frac{1}{2}I \preceq V_{U^tQ^t}^\top BV_{U^tQ^t} + V_{U^tQ^t}^\top B^\top V_{U^tQ^t} + V_{U^tQ^t}^\top B^\top BV_{U^tQ^t} \preceq \frac{1}{2}I. \tag{79}$$

Following an identical procedure to that in [34, Lemma 9.3], we obtain

$$Z(Z^\top Z)^{-1/2} = V_{U^tQ^t} + BV_{U^tQ^t} - \frac{1}{2}(I+B)V_{U^tQ^t}V_{U^tQ^t}^\top(B+B^\top)V_{U^tQ^t} - D$$

$$D \triangleq (I+B)V_{U^tQ^t}\left(\frac{1}{2}V_{U^tQ^t}^\top B^\top BV_{U^tQ^t} - C\right).$$

The expansion of the term $(V_{Z^\star}^\perp)^\top Z(Z^\top Z)^{-1/2}$ becomes analogous to that in [34, Lemma 9.3]. To upper bound the term $(I)$ defined analogously as in [34, Lemma 9.3] we have

$$(I) = V_\perp^\top\left(I + B_1 - \frac{1}{2}V_{U^tQ^t}V_{U^tQ^t}^\top(B_1 + B_1^\top)\right)V_{U^tQ^t}$$

$$= (V_{Z^\star}^\perp)^\top\left(I + \frac{\alpha}{m}\left(I - V_{U^tQ^t}V_{U^tQ^t}^\top\right)(Z^\star - \mathcal{W}U^t(\mathcal{W}U^t)^\top)\right)V_{U^tQ^t}$$

$$= (V_{Z^\star}^\perp)^\top V_{U^tQ^t} - \frac{\alpha}{m}(V_{Z^\star}^\perp)^\top V_{U^tQ^t}V_{U^tQ^t}^\top Z^\star$$

$$\quad - \frac{\alpha}{m}(V_{Z^\star}^\perp)^\top(I - V_{U^tQ^t}V_{U^tQ^t}^\top)\mathcal{W}U^t(Q^t(Q^t)^\top + Q^{t,\perp}(Q^{t,\perp})^\top)(\mathcal{W}U^t)^\top V_{U^tQ^t}.$$

Observe that $\mathrm{span}(\mathcal{W}U^tQ^t(Q^t)^\top) \subseteq \mathrm{span}(V_{U^tQ^t})$ and consequently,

$$(I - V_{U^tQ^t}V_{U^tQ^t}^\top)\mathcal{W}U^tQ^t = 0.$$

Therefore

$$(I) = (V_{Z^\star}^\perp)^\top V_{U^tQ^t} - \frac{\alpha}{m}(V_{Z^\star}^\perp)^\top V_{U^tQ^t}V_{U^tQ^t}^\top Z^\star$$

$$\quad - \frac{\alpha}{m}(V_{Z^\star}^\perp)^\top(I - V_{U^tQ^t}V_{U^tQ^t}^\top)\mathcal{W}U^tQ^{t,\perp}\left(\mathcal{W}U^tQ^{t,\perp}\right)^\top V_{U^tQ^t}$$

$$= (V_{Z^\star}^\perp)^\top V_{U^tQ^t}\left(I - \frac{\alpha}{m}V_{U^tQ^t}^\top Z^\star V_{U^tQ^t}\right)$$

$$\quad - \frac{\alpha}{m}(V_{Z^\star}^\perp)^\top(I - V_{U^tQ^t}V_{U^tQ^t}^\top)\mathcal{W}U^tQ^{t,\perp}(\mathcal{W}U^tQ^{t,\perp})^\top(V_{Z^\star}V_{Z^\star}^\top + V_{Z^\star}^\perp(V_{Z^\star}^\perp)^\top)V_{U^tQ^t}.$$

Observe that $V_{Z^\star}^\top\mathcal{W}U^tQ^{t,\perp} = V_{Z^\star}^\top U^tQ^{t,\perp} = V^t\Sigma^t(Q^t)^\top Q^{t,\perp} = 0$. Therefore,

$$\|(I)\| \leq \|(V_{Z^\star}^\perp)^\top V_{U^tQ^t}\|\left(1 - \frac{\alpha}{m}\sigma_{\min}(V_{U^tQ^t}^\top Z^\star V_{U^tQ^t})\right) + \frac{\alpha}{m}\|(V_{Z^\star}^\perp)^\top V_{U^tQ^t}\|\|U^tQ^{t,\perp}\|^2 \tag{80}$$

where $\sigma_{\min}(V_{U^tQ^t}^\top Z^\star V_{U^tQ^t}) = \sigma_{\min}(X)\sigma_{\min}(V_{U^tQ^t}^\top V_{Z^\star})$. By assumption $\sigma_{\min}(V_{Z^\star}^\top V_{U^tQ^t}) \geq 1 - c\kappa^{-1}$ setting $c \leq \frac{1}{2}$. We thus obtain

$$\|(I)\| \leq \|(V_{Z^\star}^\perp)^\top V_{U^tQ^t}\|\left(1 - \frac{\alpha}{2m}\lambda_{r^\star}(\bar{Z}^\star)m + \frac{\alpha}{m}\|U^tQ^{t,\perp}\|^2\right) \tag{81}$$

$$\leq \|(V_{Z^\star}^\perp)^\top V_{U^tQ^t}\|\left(1 - \frac{\alpha}{2}\lambda_{r^\star}(\bar{Z}^\star) + \alpha c\lambda_{r^\star}(\bar{Z}^\star)\right). \tag{82}$$

Setting $c \leq \frac{1}{6}$,

$$\|(I)\| \leq \|(V_{Z^\star}^\perp)^\top V_{U^t Q^t}\| \left(1 - \frac{\alpha}{3}\lambda_{r^\star}(\bar{Z}^\star)\right). \tag{83}$$

For $(II)$ it follows in the same way as in [34] (c.f. page 59)

$$\|(II)\| \leq \frac{\alpha}{m}\|\mathcal{A}^*\mathcal{A}(Z^\star - U^t(U^t)^\top) - \mathcal{W}(Z^\star - U^t(U^t)^\top)\mathcal{W}\|. \tag{84}$$

$$\|(III)\| \leq \left\|(V_{Z^\star}^\perp)^\top \left(B_3 - \frac{1}{2}V_{U^t Q^t}V_{U^t Q^t}^\top \left(B_3 + B_3^\top\right)\right) V_{U^t Q^t}\right\|, \tag{85}$$

with

$$B_3 = U^t Q^{t,\perp}(Q^{t,\perp})^\top Q^{t+1}(V_{U^t Q^t}^\top U^t Q^t (Q^t)^\top Q^{t+1})^{-1} V_{U^t Q^t}^\top \tag{86}$$

$$= U^t Q^{t,\perp}(Q^{t,\perp})^\top Q^{t+1}((Q^t)^\top Q^{t+1})^{-1}(V_{U^t Q^t}^\top U^t Q^t)^{-1} V_{U^t Q^t}^\top, \tag{87}$$

and

$$\|B_3\| \leq \|U^t Q^{t,\perp}\|\|(Q^{t,\perp})^\top Q^{t+1}\|\|((Q^t)^\top Q^{t+1})^{-1}\|\|(V_{U^t Q^t}^\top U^t Q^t)^{-1}\|. \tag{88}$$

Using Lemma 13 under which it holds that $\sigma_{\min}((Q^t)^\top Q^{t+1}) \geq \frac{1}{2}$

$$\|B_3\| \leq \frac{\|(Q^{t,\perp})^\top Q^{t+1}\|\|U^t Q^{t,\perp}\|}{\sigma_{\min}((Q^t)^\top Q^{t+1})\sigma_{\min}(U^t Q^t)} \leq 4\|(Q^{t,\perp})^\top Q^{t+1}\| \tag{89}$$

From the same lemma it follows that

$$\|(Q^{t,\perp})^\top Q^{t+1}\| \leq \alpha \left(\frac{\lambda_{r^\star}(\bar{Z}^\star)}{6400} + \frac{\|U^t Q^t\|\|U^t Q^{t,\perp}\|}{m}\right)\|(V_{Z^\star}^\perp)^\top V_{U^t Q^t}\| \tag{90}$$

$$+ \frac{4\alpha}{m}\|(\mathcal{A}^*\mathcal{A})(Z^\star - U^t(U^t)^\top) - \mathcal{W}(Z^\star - U^t(U^t)^\top)\mathcal{W}\| \tag{91}$$

therefore, using the above and that $\|U^t Q^{t,\perp}\| \leq c\kappa^{-2}\sqrt{m}\lambda_{r^\star}(\bar{Z}^\star)$, $\|U^t Q^t\| \leq 3\sqrt{m}\|\bar{X}\|$ we have for $c \leq \frac{1}{24 \cdot 800}$

$$\|(III)\| \leq 2\|B_3\| \leq 8\|(Q^{t,\perp})^\top Q^{t+1}\| \leq \frac{\alpha}{400}\sigma_{\min}^2(\bar{X})\|(V_{Z^\star}^\perp)^\top V_{U^t W^t}\| \tag{92}$$

$$+ 32\frac{\alpha}{m}\|\mathcal{A}^*\mathcal{A}(Z^\star - U^t(U^t)^\top) - \mathcal{W}(Z^\star - U^t(U^t)^\top)\mathcal{W}\|. \tag{93}$$

Similarly,

$$\|(IV)\| \leq 2\|B_4\| = \frac{2\alpha}{m}\|\mathcal{M}P\| \leq \frac{2\alpha}{m}\|\mathcal{A}^*\mathcal{A}(Z^\star - U^t(U^t)^\top)\|\|B_3\|. \tag{94}$$

We have already obtained a bound on $\|B_3\|$ meaning that we are to bound the remaining matrix yielding under the lemma's assumptions that

$$\|\mathcal{A}^*\mathcal{A}(Z^\star - U^t(U^t)^\top)\| \leq 11c\kappa^{-2},$$

and therefore with $c \leq \frac{1}{24 \cdot 800}$

$$\|(IV)\| \leq \frac{\alpha}{50}\kappa^{-2}\lambda_{r^\star}(\bar{Z}^\star)\|(V_{Z^\star}^\perp)^\top V_{U^t Q^t}\|$$

$$+ 352c\frac{\alpha}{m}\|\mathcal{A}^*\mathcal{A}(Z^\star - U^t(U^t)^\top) - \mathcal{W}(Z^\star - U^t(U^t)^\top)\mathcal{W}\|$$

and

$$\|(V)\| \leq \frac{1}{2}\left\|(V_{Z^\star}^\perp)^\top B V_{U^t Q^t} V_{U^t Q^t}^\top (B + B^\top) V_{U^t Q^t}\right\| \leq \frac{1}{2}\|B\|\|B + B^\top\| \leq \|B\|^2.$$

Observe that

$$\|B\| \leq \left\|\frac{\alpha}{m}\mathcal{M} + P + \frac{\alpha}{m}\mathcal{M}P\right\| \leq \|B_4\| + \frac{\alpha}{m}\|\mathcal{M}\| + \|B_3\|$$

$$\leq \left(1 + \frac{\alpha}{m}\|\mathcal{A}^*\mathcal{A}(Z^\star - U^t(U^t)^\top) - \mathcal{W}(Z^\star - U^t(U^t)^\top)\mathcal{W}\|\right)\|B_3\| + \frac{\alpha}{m}\|\mathcal{M}\|$$

$$\leq \frac{\alpha}{800}\lambda_{r^\star}(\bar{Z}^\star)\|(V_{Z^\star}^\perp)^\top V_{U^t Q^t}\| + 17\frac{\alpha}{m}\|\mathcal{A}^*\mathcal{A}(Z^\star - U^t(U^t)^\top) - \mathcal{W}(Z^\star - U^t(U^t)^\top)\mathcal{W}\|$$

$$+ \frac{\alpha}{m}\|Z^\star - (\mathcal{W}U^t)(\mathcal{W}U^t)^\top\|.$$

Therefore,

$$
\begin{aligned}
\|(V)\| \le\; & 3\frac{\alpha^2}{(800)^2}\lambda_{r^\star}^2(\bar{Z}^\star)\|(V_{Z^\star}^\perp)^\top V_{U^t Q^t}\|^2 \\
& + \frac{3(17)^2\alpha^2}{m^2}\|\mathcal{W}(Z^\star - U^t(U^t)^\top)\mathcal{W} - \mathcal{A}^*\mathcal{A}(Z^\top - U^t(U^t)^\top)\|^2 \\
& + \frac{3\alpha^2}{m^2}\|Z^\top - (\mathcal{W}U^t)(\mathcal{W}U^t)^\top\|^2 \le \frac{3\alpha^2}{m^2}\|Z^\star - (\mathcal{W}U^t)(\mathcal{W}U^t)^\top\|^2 \\
& + \frac{3\alpha c\kappa^{-4}}{(800)^2}\lambda_{r^\star}(\bar{Z}^\star)\|(V_{Z^\star}^\perp)^\top V_{U^t Q^t}\|^2 \\
& + \frac{3(17)^2 c\kappa^{-4}}{24\cdot 800 m}\alpha\|\mathcal{W}(Z^\star - U^t(U^t)^\top)\mathcal{W} - \mathcal{A}^*\mathcal{A}(Z^\star - U^t(U^t)^\top)\|.
\end{aligned}
$$

Similarly as in [34] we have

$$
\begin{aligned}
\|(VI)\| \le \|(V_{Z^\star}^\perp)^\top D\| &\le \left\|(V_{Z^\star}^\perp)^\top (I+B)V_{U^t Q^t}\left(\frac{1}{2}V_{U^t Q^t}^\top B^\top B V_{U^t Q^t} - C\right)\right\| \\
&\le 2\left(\|B\|^2 + \|C\|\right) \\
\|C\| &\le 3\|V_{U^t Q^t}^\top B V_{U^t Q^t} + V_{U^t Q^t}^\top B^\top V_{U^t Q^t} + V_{U^t Q^t}^\top B^\top B V_{U^t Q^t}\|.
\end{aligned}
$$

Following [34, Lemma 9.3] there holds that

$$
\begin{aligned}
\|(VI)\| \le 56\Bigg( &3\frac{\alpha^2}{m^2}\|Z^\star - (\mathcal{W}U^t)(\mathcal{W}U^t)^\top\|^2 + 3c\frac{\alpha\kappa^{-4}}{2\cdot 400^2}\lambda_{r^\star}(\bar{Z}^\star)\|(V_{Z^\star}^\perp)^\top V_{U^t Q^t}\|^2 \\
& + \frac{3\alpha}{2m}c\kappa^{-4}\|\mathcal{A}^*\mathcal{A}(Z^\star - U^t(U^t)^\top) - \mathcal{W}(Z^\star - U^t(U^t)^\top)\mathcal{W}\|\Bigg).
\end{aligned}
$$

The final results follows readily combining all the above bounds.

We are left to establish that (79) holds.

Observe that by definition the matrix is symmetric. Consequently it suffices to prove that

$$
2\|B\| + \|B^\top B\| \le \frac{1}{2}.
$$

From the upper bound on (VI) it follows from the derivations in [34, Lemma 9.3] that

$$
\|B\| \le 10\alpha\|\bar{X}\|^2 + \frac{\alpha}{400}\lambda_{r^\star}(\bar{Z}^\star) + 33c\alpha\lambda_{r^\star}(\bar{Z}^\star).
$$

Consequently, for the above to hold it suffices to chose $c$ such that

$$
10c\kappa^{-2} + \frac{c}{400}\kappa^{-2} + 33c^2\kappa^{-2} \le \frac{1}{6},
$$

which follows under the restriction on $c$ as stated in the lemma.

**(v)** We write

$$Z^\star - U^{t+1}(U^{t+1})^\top = Z^\star - \mathcal{W}^2 U^t (\mathcal{W}^2 U^t)^\top - \frac{\alpha}{m} \mathcal{W}^2 U^t (U^t)^\top (Z^\star - \mathcal{W} U^t (\mathcal{W} U^t)^\top)$$

$$- \frac{\alpha}{m} (Z^\star - \mathcal{W} U^t (\mathcal{W} U^t)^\top) U^t (U^t)^\top \mathcal{W}^2$$

$$+ \frac{\alpha}{m} \mathcal{W}^2 U^t (U^t)^\top \left( \mathcal{W}(Z^\star - U^t(U^t)^\top)\mathcal{W} - \mathcal{A}^* \mathcal{A}(Z^\star - U^t(U^t)^\top) \right)$$

$$+ \frac{\alpha}{m} \left( \mathcal{W}(Z^\star - U^t(U^t)^\top)\mathcal{W} - \mathcal{A}^* \mathcal{A}(Z^\star - U^t(U^t)^\top) \right) U^t (U^t)^\top \mathcal{W}^2$$

$$- \frac{\alpha^2}{m^2} (\mathcal{A}^* \mathcal{A}(Z^\star - U^t(U^t)^\top)) U^t (U^t)^\top (\mathcal{A}^* \mathcal{A}(Z^\star - U^t(U^t)^\top))$$

$$= \mathcal{W} \left( I - \frac{\alpha}{m} \mathcal{W} U^t (U^t)^\top \mathcal{W} \right) (Z^\star - U^t(U^t)^\top) \left( I - \frac{\alpha}{m} \mathcal{W} U^t (U^t)^\top \mathcal{W} \right) \mathcal{W} + \mathcal{W} U^t (U^t)^\top \mathcal{W}$$

$$- \mathcal{W}^2 U^t (U^t)^\top \mathcal{W}^2 - \frac{\alpha^2}{m^2} \mathcal{W}^2 U^t (U^t)^\top \mathcal{W}(Z^\star - U^t(U^t)^\top)\mathcal{W} U^t (U^t)^\top \mathcal{W}$$

$$+ \frac{\alpha}{m} \mathcal{W}^2 U^t (U^t)^\top (\mathcal{W}(Z^\star - U^t(U^t)^\top)\mathcal{W} - \mathcal{A}^* \mathcal{A}(Z^\star - U^t(U^t)^\top))$$

$$+ \frac{\alpha}{m} (\mathcal{W}(Z^\star - U^t(U^t)^\top)\mathcal{W} - \mathcal{A}^* \mathcal{A}(Z^\star - U^t(U^t)^\top)) U^t (U^t)^\top \mathcal{W}^2$$

$$- \frac{\alpha^2}{m^2} \mathcal{A}^* \mathcal{A}(Z^\star - U^t(U^t)^\top) U^t (U^t)^\top \mathcal{A}^* \mathcal{A}(Z^\star - U^t(U^t)^\top).$$

We are interested in bounding the following terms

$$V_{Z^\star}^\top (Z^\star - U^{t+1}(U^{t+1})^\top) =$$

$$\underbrace{V_{Z^\star}^\top \left( I - \frac{\alpha}{m} U^t (U^t)^\top \right) (Z^\star - \mathcal{W} U^t (\mathcal{W} U^t)^\top) \left( I - \frac{\alpha}{m} U^t (U^t)^\top \right) \mathcal{W}}_{\triangleq (I)}$$

$$+ \underbrace{\frac{\alpha}{m} V_{Z^\star}^\top \left( Z^\star - \mathcal{W} U^t (\mathcal{W} U^t)^\top \right) U^t (U^t)^\top (\mathcal{W} - \mathcal{W}^2)}_{\triangleq (II)}$$

$$- \underbrace{\frac{\alpha^2}{m^2} V_{Z^\star}^\top U^t (U^t)^\top \mathcal{W}(Z^\star - U^t(U^t)^\top)\mathcal{W} U^t (U^t)^\top \mathcal{W}}_{\triangleq (III)}$$

$$+ \underbrace{\frac{\alpha}{m} V_{Z^\star}^\top U^t (U^t)^\top (\mathcal{W}(Z^\star - U^t(U^t)^\top)\mathcal{W} - \mathcal{A}^* \mathcal{A}(XX^\top - U^t(U^t)^\top))}_{\triangleq (IV)}$$

$$+ \underbrace{\frac{\alpha}{m} V_{Z^\star}^\top ((\mathcal{W}(Z^\star - U^t(U^t)^\top)\mathcal{W} - \mathcal{A}^* \mathcal{A}(Z^\star - U^t(U^t)^\top)) U^t (U^t)^\top) \mathcal{W}^2}_{\triangleq (V)}$$

$$- \underbrace{\frac{\alpha^2}{m^2} V_{Z^\star}^\top \mathcal{A}^* \mathcal{A}(Z^\star - U^t(U^t)^\top) U^t (U^t)^\top \mathcal{A}^* \mathcal{A}(Z^\star - U^t(U^t)^\top)}_{\triangleq (VI)}.$$

We begin with term $(I)$:

$$(I) = V_{Z^\star}^\top \left(I - \frac{\alpha}{m} U^t(U^t)^\top\right)(Z^\star - (U^t)(U^t)^\top)(I - \frac{\alpha}{m} U^t(U^t)^\top)\mathcal{W}$$

$$= V_{Z^\star}^\top \left(I - \frac{\alpha}{m} U^t(U^t)^\top\right) V_{Z^\star} V_{Z^\star}^\top (Z^\star - (U^t)(U^t)^\top)(I - \frac{\alpha}{m} U^t(U^t)^\top)\mathcal{W}$$

$$+ V_{Z^\star}^\top \left(I - \frac{\alpha}{m} U^t(U^t)^\top\right) (V_{Z^\star}^\perp(V_{Z^\star}^\perp)^\top)(Z^\star - (U^t)(U^t)^\top)(I - \frac{\alpha}{m} U^t(U^t)^\top)\mathcal{W}$$

$$= \underbrace{V_{Z^\star}^\top \left(I - \frac{\alpha}{m} U^t(U^t)^\top\right) V_{Z^\star} V_{Z^\star}^\top (Z^\star - (U^t)(U^t)^\top)(I - \frac{\alpha}{m} U^t(U^t)^\top)\mathcal{W}}_{\triangleq (I.a)}$$

$$- \underbrace{V_{Z^\star}^\top \left(I - \frac{\alpha}{m} U^t(U^t)^\top\right) (V_{Z^\star}^\perp(V_{Z^\star}^\perp)^\top)U^t(U^t)^\top(I - \frac{\alpha}{m} U^t(U^t)^\top)\mathcal{W}}_{\triangleq (I,b)}.$$

Therefore, we obtain

$$\|(I,a)\| \leq \left\|I - \frac{\alpha}{m} V_{Z^\star}^\top U^t(U^t)^\top V_{Z^\star}\right\| \|V_{Z^\star}^\top(Z^\star - U^t(U^t)^\top)\| \|(I - \frac{\alpha}{m} U^t(U^t))\mathcal{W}\|$$

$$\leq \left(1 - \frac{\alpha}{m}\sigma_{\min}^2(V_{Z^\star}^\top U^t Q^t)\right) \|V_{Z^\star}^\top(Z^\star - U^t(U^t)^\top)\|.$$

By assumption $\sigma_{\min}^2(U^t Q^t) \geq \frac{m}{10}\lambda_{r^\star}(\bar{Z}^\star)$, which leads to

$$\|(I,a)\| \leq \left(1 - \frac{\alpha}{20}\lambda_{r^\star}(\bar{Z}^\star)\right) \|V_{Z^\star}^\top(Z^\star - U^t(U^t)^\top)\|.$$

We now proceed with the quantity

$$\|(I,b)\| \leq \frac{\alpha}{m} \|V_{Z^\star}^\top U^t(U^t)^\top \mathcal{W}(V_{Z^\star}^\perp(V_{Z^\star}^\perp)^\top)U^t(U^t)^\top\|$$

$$= \frac{\alpha}{m} \|V_{Z^\star}^\top U^t Q^t(U^t Q^t)^\top \mathcal{W}(V_{Z^\star}^\perp(V_{Z^\star}^\perp)^\top)^\top U^t(U^t)^\top\|$$

$$\leq \frac{\alpha}{m} \|V_{Z^\star}^\top U^t Q^t\| \|(V_{Z^\star}^\perp)^\top \mathcal{W} U^t Q^t\| \|(V_{Z^\star}^\perp)^\top U^t(U^t)^\top\|,$$

where because $Z^\star \mathcal{W} = \mathcal{W} Z^\star$, there holds

$$\|(V_{Z^\star}^\perp)^\top \mathcal{W} U^t Q^t\| \leq \|(V_{Z^\star}^\perp)^\top U^t Q^t\|.$$

We can now use the same procedure as in the proof of [34, Lemma 9.5] and obtain, under $c \leq \frac{1}{9\cdot 400}$, that

$$\|(I,b)\| \leq \frac{\lambda_{r^\star}(\bar{Z}^\star)}{100} \|V_{Z^\star}^\top(Z^\star - U^t(U^t)^\top)\| + \frac{\lambda_{r^\star}(\bar{Z}^\star)}{400} \|U^t Q^{t,\perp}(U^t Q^{t,\perp})^\top\|.$$

Proceeding to $(II)$, we have

$$\|(II)\| \leq \frac{\alpha}{m} \|V_{Z^\star}^\top(Z^\star - U^t(U^t)^\top)\| \|U^t(U^t)(\mathcal{W} - \mathcal{W}^2)\|$$

$$\leq \frac{\alpha 9\|\bar{X}\|^2 m}{m} \|\mathcal{W} - \mathcal{W}^2\| \|V_{Z^\star}^\top(Z^\star - U^t(U^t)^\top)\| \leq 9\alpha\|\bar{X}\|^2 \rho \|V_{Z^\star}^\top(Z^\star - U^t(U^t)^\top)\|.$$

Combining

$$\|I\| + \|II\| \leq \left(1 - \frac{\alpha\lambda_{r^\star}(\bar{Z}^\star)}{40} + 9\alpha\|\bar{X}\|^2\rho\right) \|V_{Z^\star}^\top(Z^\star - U^t(U^t)^\top)\|$$

$$+ \frac{\alpha\lambda_{r^\star}(\bar{Z}^\star)}{400} \|U^t Q^{t,\perp}(U^t Q^{t,\perp})^\top\|.$$

Following the proof of [34, Lemma 9.5], we obtain, under $c \leq \frac{1}{81000}$,

$$\|(III)\| \leq \alpha\lambda_{r^\star}(\bar{Z}^\star)\left(\frac{1}{200}\|V_{Z^\star}^\top(Z^\star - U^t(U^t)^\top)\| + \frac{1}{1000}\|U^t Q^{t,\perp}(U^t Q^{t,\perp})^\top\|\right).$$

Similarly, we have

$$\||(IV)\|| + \||(V)\|| \le \frac{\alpha \lambda_{r^\star}(\bar{Z}^\star)}{50} \left( \||V_{Z^\star}^\top (U^t(U^t)^\top - Z^\star)\|| + \||U^t Q^{t,\perp}(U^t Q^{t,\perp})^\top\|| \right).$$

Lastly,

$$\||(VI)\|| \le \frac{\alpha^2}{m^2} \|U^t\|^2 \||\mathcal{A}^*\mathcal{A}(Z^\star - U^t(U^t))\||^2 \le \frac{9\alpha^2}{m} \|\bar{X}\|^2 c^2 \kappa^{-4} \||Z^\star - U^t(U^t)\||^2$$

$$\le \frac{9\alpha^2}{m} \|\bar{X}\|^2 c\kappa^{-4} \||Z^\star - U^t(U^t)^\top\|| \left( \|X\|^2 + \|U^t\|^2 \right)$$

$$\le \frac{9}{m} \|\bar{X}\|^2 10m \|\bar{X}\|^2 c\kappa^{-4} \||Z^\star - U^t(U^t)^\top\||$$

$$\le 90\alpha \lambda_{r^\star}^2(\bar{Z}^\star) \||Z^\star - U^t(U^t)^\top\||.$$

Using [34, Lemma B.4], yields

$$\||(VI)\| \le 90\alpha \lambda_{r^\star}^2(\bar{Z}^\star) \left( 4\||V_{Z^\star}^\top(Z^\star - U^t(U^t)^\top)\|| + \||U^t Q^{t,\perp}(U^t Q^{t,\perp})^\top\|| \right).$$

Combining all bounds, we finally obtain

$$\||V_{Z^\star}^\top(Z^\star - U^{t+1}(U^{t+1})^\top)\|| \le \left( 1 - \frac{\alpha}{200}\lambda_{r^\star}(\bar{Z}^\star) + 9\alpha\|\bar{X}\|^2 \rho \right) \||V_{Z^\star}^\top(Z^\star - U^t(U^t)^\top)\||$$

$$+ \alpha \frac{\lambda_{r^\star}(\bar{Z}^\star)}{100} \||U^t Q^{t,\perp}(U^t Q^{t,\perp})^\top\||.$$

### F.2 Technical lemmas: auxiliary to proving Lemma 9

**Lemma 12.** *Assume that* $\alpha \le \alpha \min\{\|\bar{Z}^\star\|^{-1}, m\|\mathcal{W}\Delta^t\mathcal{W} - \mathcal{A}^*\mathcal{A}(\Delta^t)\|\}$ *for some* $c \in (0, 1/1200)$ *and that* $\|U^t\| \le 3\sqrt{m}\|\bar{X}\|$. *Moreover assume that* $V_{Z^\star}^\top U^{t+1}Q^t$ *is full rank. Then, if* $\|(V_{Z^\star}^\perp)^\top V_{U^tQ^t}\| \le c\kappa^{-1}$ *there holds that*

$$\|(V_{Z^\star}^\perp)^\top V_{U^{t+1}Q^t}\| \le 2\left( \|(V_{Z^\star}^\perp)^\top V_{U^tQ^t}\| + \frac{\alpha}{m}\|\mathcal{A}^*\mathcal{A}(\Delta^t)\| \right) \tag{95}$$

$$\|(V_{Z^\star}^\perp)^\top V_{U^{t+1}Q^t}\| \le \frac{1}{50}. \tag{96}$$

*Proof.*

$$U^{t+1}Q^t = \left( \mathcal{W}^2 + \frac{\alpha}{m}\mathcal{A}^*\mathcal{A}(Z^\star - U^t(U^t)^\top) \right) U^t Q^t, \tag{97}$$

and denote by $V_{U^tQ^t}\Sigma_{U^tQ^t}Q_{U^tQ^t}^\top$ the singular value decomposition of $U^tQ^t$. Denote by

$$Y \triangleq \left( \mathcal{W}^2 + \frac{\alpha}{m}\mathcal{A}^*\mathcal{A}(XX^\top - U^t(U^t)^\top) \right) V_{U^tQ^t}. \tag{98}$$

By assumption the matrix $\Sigma_{U^tQ^t}Q_{U^tQ^t}^\top$ is full rank and the matrix $Y = V_Y\Sigma_Y Q_Y^\top$ has the same column space as $U^{t+1}Q^t$. It follows that

$$\|(V_{Z^\star}^\perp)^\top V_{U^{t+1}Q^t}\| = \|(V_{Z^\star}^\perp)^\top V_Y\| \le \|(V_{Z^\star}^\perp)^\top V_Y\Sigma_Y Q_Y^\top\|\|(\Sigma_Y Q_Y^\top)^{-1}\| = \frac{\|(V_{Z^\star}^\perp)^\top Y\|}{\sigma_{\min}(Y)} \tag{99}$$

Then,

$$\sigma_{\min}(Y) \ge \sigma_{\min}(V_{Z^\star}^\top Y) = \sigma_{\min}(V_{Z^\star}^\top V_{U^tQ^t}) - \frac{\alpha}{m}\| \left( V_{Z^\star}^\top \left( \mathcal{A}^*\mathcal{A}(Z^\star - U^t(U^t)^\top) \right) V_{U^tQ^t} \right) \| \tag{100}$$

Observe that by assumption $\|(V_{Z^\star}^\perp)^\top V_{U^tQ^t}\| \le c\kappa^{-1}$ if $c < 1$ there holds that $\sigma_{\min}(V_{Z^\star}^\top V_{U^tQ^t}) \ge 1 - c\kappa^{-1}$ implying that the matrix $V_{Z^\star}^\top V_{U^tQ^t}$ is of rank $r^\star$ (full-rank). Observe that there holds that

$$\|\mathcal{A}^*\mathcal{A}(\Delta^t)\| \le \|\mathcal{A}^*\mathcal{A}(\Delta^t) - \mathcal{W}\Delta^t\mathcal{W}\| + \|Z^\star - \mathcal{W}U^t(\mathcal{W}U^t)^\top\| \tag{101}$$

$$\frac{\alpha}{m}\|\mathcal{A}^*\mathcal{A}(\Delta^t)\| \le c + \frac{\alpha}{m}\|Z^\star\| + \alpha 9\|\bar{X}\|^2 \le 11c \tag{102}$$

where we have used that

$$\alpha \le c\min\{\|\bar{X}\|^{-2}, m\|(\mathcal{W}^2 - \mathcal{A}^*\mathcal{A})(\Delta^t)\|^{-1}\} \tag{103}$$

and

$$\|U^t\| \le 3\sqrt{m}\|\bar{X}\|. \tag{104}$$

Thus, there holds

$$\sigma_{\min}(Y) \ge 1 - c\kappa^{-1} - 11c. \tag{105}$$

Under the assumption that $c \le \frac{1}{24}$ there holds that

$$\sigma_{\min}(Y) \ge \frac{1}{2}. \tag{106}$$

In this way we Further,

$$\|(V_{Z^\star}^\perp)^\top Y\| \le \|(V_{Z^\star}^\perp)^\top V_{U^t Q^t}\| + \frac{\alpha}{m}\|\mathcal{A}^*\mathcal{A}(\Delta^t)\|. \tag{107}$$

Observe that by construction

$$\|(V_{Z^\star}^\perp)^\top V_{U^t Q^t}\| = \|(V_{Z^\star}^\perp)^\top V_{U^{t+1} Q^t}\|, \tag{108}$$

and therefore

$$\|(V_{Z^\star}^\perp)^\top V_{U^{t+1} Q^t}\| \le 2\left(\|(V_{Z^\star}^\perp)^\top V_{U^t Q^t}\| + \frac{\alpha}{m}\|\mathcal{A}^*\mathcal{A}(\Delta^t)\|\right). \tag{109}$$

Similarly,

$$\|(V_{Z^\star}^\perp)^\top Y\| \le \|(V_{Z^\star}^\perp)^\top \mathcal{W}^2 V_{U^t Q^t}\| + \frac{\alpha}{m}\|\mathcal{A}^*\mathcal{A}(\Delta^t)\| \tag{110}$$

where using that $\mathcal{W}^2 Z^\star = Z^\star \mathcal{W}^2$ we obtain

$$\|(V_{Z^\star}^\perp)^\top \mathcal{W}^2 V_{U^t Q^t}\| \le \|(V_{Z^\star}^\perp)^\top V_{U^t Q^t}\|, \tag{111}$$

consequently,

$$\|(V_{Z^\star}^\perp)^\top V_{U^{t+1} Q^t}\| \le \frac{\|(V_{Z^\star}^\perp)^\top Y\|}{\sigma_{\min}(Y)} \le 2\left(\|(V_{Z^\star}^\perp)^\top V_{U^t Q^t}\| + \frac{\alpha}{m}\|\mathcal{A}^*\mathcal{A}(Z^\star - U^t(U^t)^\top)\|\right) \tag{112}$$

We can further bound

$$\frac{\alpha}{m}\|\mathcal{A}^*\mathcal{A}(Z^\star - U^t(U^t)^\top)\| \le \frac{\alpha}{m}\|Z^\star - U^t(U^t)^\top\| + \frac{\alpha}{m}\|\mathcal{A}^*\mathcal{A}(\Delta^t) - \mathcal{W}\Delta^t\mathcal{W}\| \tag{113}$$

$$\le \frac{\alpha}{m}\left(m\|\bar{X}\|^2 + 9m\|\bar{X}\|^2\right) + c \le 11c. \tag{114}$$

Consequently,

$$\|(V_{Z^\star}^\perp)^\top V_{U^{t+1} Q^t}\| \le 2c\kappa^{-1} + 22c \le \frac{1}{50} \tag{115}$$

under the assumption that $c < \frac{1}{1200}$.

$\square$

**Lemma 13.** *Assume that* $\|U^t Q^{t,\perp}\| \le 2\sigma_{\min}(U^t Q^t)$ *and* $\|U^t\| \le 3\sqrt{m}\|\bar{X}\|$ *holds. Further, assume that*

$$\|\mathcal{W}\Delta^t\mathcal{W} - \mathcal{A}^*\mathcal{A}(\Delta^t)\| \le cm\sigma_{\min}^2(\bar{X}) \tag{116}$$

$$\|(V_{Z^\star}^\perp)^\top V_{U^t Q^t}\| \le c \tag{117}$$

$$\alpha \le c\kappa^{-1}\|\bar{Z}^\star\|^{-1} \tag{118}$$

$$\|U^t Q^{t,\perp}\| \le c\kappa^{-2}\sqrt{m}\|\bar{X}\| \tag{119}$$

*for some* $c \le \frac{1}{240}$. *Then, there holds that*

$$\|(Q^{t,\perp})^\top Q^{t+1}\| \le \alpha\left(\frac{\lambda_{r^\star}(\bar{Z}^\star)}{6400} + \frac{\|U^t Q^t\|\|U^t Q^{t,\perp}\|}{m}\right)\|(V_{Z^\star}^\perp)^\top V_{U^t Q^t}\| + 4\frac{\alpha}{m}\|\mathcal{W}\Delta^t\mathcal{W} - \mathcal{A}^*\mathcal{A}(\Delta^t)\| \tag{120}$$

*and*

$$\sigma_{\min}((Q^{t,\perp})^\top Q^{t+1}) \ge 1/2 \tag{121}$$

*Proof.* Follows identically to that in [34]. $\square$

# G Proofs of the RIP Properties and Consequences

## G.1 Proof of Lemma 1

By definition of the map $\mathcal{A}$ we have

$$\|\mathcal{A}(Z)\|_2^2 = \frac{1}{mn} \sum_{\ell=1}^N \left( \left\langle \left( m w_{\mathcal{V}(\ell)} w_{\mathcal{V}(\ell)}^\top - \frac{1}{m} 1_m 1_m^\top \right) \otimes A_\ell, Z \right\rangle + m \langle A_\ell, \bar{Z} \rangle \right)^2 \tag{122}$$

$$\overset{(a)}{\leq} (1 + \varepsilon_u) \, m^2 \|\bar{\mathcal{A}}(\bar{Z})\|_2^2 + \left( 1 + \frac{1}{\varepsilon_u} \right) \underbrace{\frac{1}{mn} \sum_{\ell=1}^N \left( \left\langle A_\ell, \sum_{j=1}^m \sum_{k=1}^m \left( m w_{\mathcal{V}(\ell)j} w_{\mathcal{V}(\ell)k} - \frac{1}{m} \right) [Z]_{jk} \right\rangle \right)^2}_{\triangleq A}, \tag{123}$$

where in $(a)$ we use Young's inequality for any $\varepsilon_u > 0$. Similarly, for any $\varepsilon_\ell \in (0, 1)$

$$\|\mathcal{A}(Z)\|_2^2 \geq (1 - \varepsilon_\ell) \, m^2 \|\bar{\mathcal{A}}(\bar{Z})\|_2^2 + \left( 1 - \frac{1}{\varepsilon_\ell} \right) A. \tag{124}$$

We now proceed by upper bounding $A$. Because the matrix $W$ is doubly stochastic we can write

$$A = \frac{1}{mn} \sum_{\ell=1}^N \left( \left\langle A_\ell, \sum_{j=1}^m \sum_{k=1}^m \left( m w_{\mathcal{V}(\ell)j} w_{\mathcal{V}(\ell)k} - \frac{1}{m} \right) ([Z]_{jk} - \bar{Z}) \right\rangle \right)^2. \tag{125}$$

consequently, we write

$$A \leq \frac{1}{mn} \sum_{\ell=1}^N \left( \sum_{j=1}^m \sum_{k=1}^m \left| m w_{\mathcal{V}(\ell)j} w_{\mathcal{V}(\ell)k} - \frac{1}{m} \right| |\langle A_\ell, [Z]_{jk} - \bar{Z} \rangle| \right)^2. \tag{126}$$

Because $w_{ij} \geq 0$ for all $i$ and $j$ there holds

$$\max_{i,l,k} \left| w_{il} w_{ik} m - \frac{1}{m} \right| \leq m \max_{ik} \left| w_{ik}^2 - \frac{1}{m^2} \right| = m \max_{ik} \left( w_{ik} + \frac{1}{m} \right) \left| w_{ik} - \frac{1}{m} \right| \tag{127}$$

$$\leq m \left( 1 + \frac{1}{m} \right) \|W - J\|_\infty \leq 2\sqrt{m} m \rho. \tag{128}$$

Thus,

$$A \leq \frac{1}{mn} \sum_{\ell=1}^N 4 m^3 \rho^2 \left( \sum_{j=1}^m \sum_{k=1}^m |\langle A_\ell, [Z]_{jk} - \bar{Z} \rangle| \right)^2 \tag{129}$$

$$\leq \frac{1}{mn} \sum_{\ell=1}^N \sum_{j=1}^m \sum_{k=1}^m 4 m^5 \rho^2 \left( \langle A_\ell, [Z]_{jk} - \bar{Z} \rangle \right)^2 \leq 4 m^5 \rho^2 \sum_{j=1}^m \sum_{k=1}^m \left\| \bar{\mathcal{A}}([Z]_{jk} - \bar{Z}) \right\|^2. \tag{130}$$

Using that $\bar{\mathcal{A}}$ fulfills the $(\delta_{2r}, 2r)$ RIP we obtain

$$\|\mathcal{A}(Z)\|_2^2 \leq (1 + \varepsilon_u) \, (1 + \delta_{2r}) \, m^2 \|\bar{Z}\|_F^2 + \left( 1 + \frac{1}{\varepsilon_u} \right) 4 m^5 \rho^2 (1 + \delta_{2r}) \|Z - \mathcal{J} Z \mathcal{J}\|_F^2 \tag{131}$$

$$\|\mathcal{A}(Z)\|_2^2 \geq (1 - \varepsilon_\ell) \, (1 - \delta_{2r}) \, m^2 \|\bar{Z}\|_F^2 - \left( \frac{1}{\varepsilon_\ell} - 1 \right) 4 m^5 \rho^2 (1 + \delta_{2r}) \|Z - \mathcal{J} Z \mathcal{J}\|_F^2. \tag{132}$$

Set $\varepsilon_u = \varepsilon_\ell = \frac{\delta_{2r}}{1 + \delta_{2r}}$, then

$$(1 + \varepsilon_u)(1 + \delta_{2r}) = 1 + 2\delta_{2r} \tag{133}$$

$$(1 - \varepsilon)(1 - \delta_{2r}) \geq 1 - 2\delta_{2r}, \tag{134}$$

and

$$\left(1 + \frac{1}{\varepsilon_u}\right) = \frac{1 + 2\delta_{2r}}{\delta_{2r}} \tag{135}$$

$$\left(\frac{1}{\varepsilon_\ell} - 1\right) = \frac{1}{\delta_{2r}} \leq \frac{1 + 2\delta_{2r}}{\delta_{2r}}, \tag{136}$$

and consequently

$$\|\mathcal{A}(Z)\|_2^2 \leq (1 + 2\delta_{2r}) \|\mathcal{J}Z\mathcal{J}\|_F^2 + 4m^5\rho^2 \frac{1 + 2\delta_{2r}}{\delta_{2r}}(1 + \delta_{2r})\|Z - \mathcal{J}Z\mathcal{J}\|^2 \tag{137}$$

$$\|\mathcal{A}(Z)\|_2^2 \geq (1 - 2\delta_{2r}) \|\mathcal{J}Z\mathcal{J}\|_F^2 - 4m^5\rho^2 \frac{1 + 2\delta_{2r}}{\delta_{2r}}(1 + \delta_{2r})\|Z - \mathcal{J}Z\mathcal{J}\|^2. \tag{138}$$

## G.2 Proof of Lemma 2

due to linearly of the operators we have

$$\|\mathcal{W}Z\mathcal{W} - \mathcal{A}^*\mathcal{A}(Z)\| \leq \underbrace{\|\mathcal{J}Z\mathcal{J} - \mathcal{A}^*\mathcal{A}(\mathcal{J}Z\mathcal{J})\|}_{\triangleq (I)} + \underbrace{\|\mathcal{W}(Z - \mathcal{J}Z\mathcal{J})\mathcal{W} - \mathcal{A}^*\mathcal{A}(Z - \mathcal{J}Z\mathcal{J})\|}_{\triangleq (II)}. \tag{139}$$

We start off with $(I)$ which can be rewritten as

$$(I) = \max_{v:\|v\|_2 \leq 1} \langle \mathcal{J}vv^\top \mathcal{J}, \mathcal{J}Z\mathcal{J} \rangle - \langle \mathcal{A}(vv^\top), \mathcal{A}(\mathcal{J}Z\mathcal{J}) \rangle \tag{140}$$

$$= \max_{v:\|v\|_2 \leq 1} \left( \frac{1}{4} \|\mathcal{J}(Z + vv^\top)\mathcal{J}\|_F^2 - \frac{1}{4} \|\mathcal{J}(Z - vv^\top)\mathcal{J}\|_F^2 - \frac{1}{4} \|\mathcal{A}(vv^\top + \mathcal{J}Z\mathcal{J})\|_F^2 \right. \tag{141}$$

$$\left. + \frac{1}{4} \|\mathcal{A}(vv^\top - \mathcal{J}Z\mathcal{J})\|_F^2 \right) \overset{(a)}{\leq} \max_{v:\|v\|_2 \leq 1} \left( \frac{1}{4} \|\mathcal{J}(Z + vv^\top)\mathcal{J}\|_F^2 - \frac{1}{4} \|\mathcal{J}(Z - vv^\top)\mathcal{J}\|_F^2 \right. \tag{142}$$

$$\left. - \frac{(1 + 2\delta_{2(r+1)})}{4} \|\mathcal{J}(Z + vv^\top)\mathcal{J}\|_F^2 + \frac{1 - 2\delta_{2(r+1)}}{4} \|\mathcal{J}(Z - vv^\top)\mathcal{J}\|_F^2 \right. \tag{143}$$

$$\left. + \frac{\hat{\Delta}_{2(r+1)}}{2} \|vv^\top - \mathcal{J}vv^\top \mathcal{J}\|_F^2 \right) \leq \max_{v:\|v\|_2 \leq 1} \left( \delta_{2(r+1)} \|\mathcal{J}(Z + vv^\top)\mathcal{J}\|_F^2 \right. \tag{144}$$

$$\left. - \delta_{2(r+1)} \|\mathcal{J}(Z - vv^\top)\mathcal{J}\|_F^2 + \frac{\hat{\Delta}_{2(r+1)}}{2} \|vv^\top - \mathcal{J}vv^\top \mathcal{J}\|_F^2 \right) \tag{145}$$

$$= \max_{v:\|v\|_2 \leq 1} \left( \delta_{2(r+1)} \left( \|\mathcal{J}Z\mathcal{J}\|_F^2 + \|\mathcal{J}vv^\top \mathcal{J}\|_F^2 \right) + \frac{\hat{\Delta}_{2(r+1)}}{2} \|vv^\top - \mathcal{J}vv^\top \mathcal{J}\|_F^2 \right) \tag{146}$$

where in (a) we have used Lemma 1. Under the assumption that $\hat{\Delta}_{2(r+1)} \leq \delta_{2(r+1)}$ there holds

$$(I) \leq \delta_{2(r+1)} \|\mathcal{J}Z\mathcal{J}\|_F^2 + \delta_{2(r+1)} \overset{(a)}{\leq} 2\delta_{2(r+1)}, \tag{147}$$

where we have used in (a) that $\|\mathcal{J}Z\mathcal{J}\|_f^2 \leq 1$. Consequently, due to the linearity of the operators involved, there holds for any $Z$ fulfilling the conditions of the lemma that

$$\|\mathcal{W}Z\mathcal{W} - \mathcal{A}^*\mathcal{A}(Z)\| \leq 2\delta_{2(r+1)} \|Z\|_F. \tag{148}$$

We now proceed with term $(II)$ and write

$$(II) = \max_{v:\|v\|_2 \leq 1} \left( \underbrace{\langle vv^\top, \mathcal{W}(Z - \mathcal{J}Z\mathcal{J})\mathcal{W} \rangle}_{\triangleq (II.a)} - \underbrace{\langle \mathcal{A}(vv^\top), \mathcal{A}(Z - \mathcal{J}Z\mathcal{J}) \rangle}_{\triangleq (II.b)} \right), \tag{149}$$

where

$$- (II.b) = -\langle \mathcal{A}(vv^\top - \mathcal{J}vv^\top\mathcal{J}), \mathcal{A}(Z - \mathcal{J}Z\mathcal{J})\rangle - \langle\mathcal{A}(\mathcal{J}vv^\top\mathcal{J}),\mathcal{A}(Z-\mathcal{J}Z\mathcal{J})\rangle \tag{150}$$

$$\leq \|\mathcal{A}(vv^\top - \mathcal{J}vv^\top\mathcal{J})\|\|\mathcal{A}(Z-\mathcal{J}Z\mathcal{J})\| + \|\mathcal{A}(\mathcal{J}vv^\top\mathcal{J})\|\|\mathcal{A}(Z-\mathcal{J}Z\mathcal{J})\| \tag{151}$$

$$\overset{(a)}{\leq} \hat{\Delta}_{2(r+1)}\|vv^\top - \mathcal{J}vv^\top\mathcal{J}\|_F\|Z-\mathcal{J}Z\mathcal{J}\|_F \tag{152}$$

$$+ \sqrt{(1+2\delta_{2(r+1)})\hat{\Delta}_{2(r+1)}}\|\mathcal{J}vv^\top\mathcal{J}\|_F\|Z-\mathcal{J}Z\mathcal{J}\|_F \leq \frac{\hat{\Delta}_{2(r+1)}}{2}\|vv^\top - \mathcal{J}vv^\top\mathcal{J}\|_F^2 \tag{153}$$

$$+ \frac{\hat{\Delta}_{2(r+1)}}{2}\|Z-\mathcal{J}Z\mathcal{J}\|_F^2 + \sqrt{(1+2\delta_{2(r+1)})\hat{\Delta}_{2(r+1)}}\|\mathcal{J}vv^\top\mathcal{J}\|_F\|Z-\mathcal{J}Z\mathcal{J}\|_F, \tag{154}$$

where in (a) we have invoked Lemma 1 and

$$(II.a) = \langle vv^\top, \mathcal{W}Z(\mathcal{I}-\mathcal{J})\mathcal{W} + \mathcal{J}Z(\mathcal{I}-\mathcal{J})\mathcal{W}\rangle \leq 2\rho\|(\mathcal{I}-\mathcal{J})Z\|_F \leq 2\rho\|Z-\mathcal{J}Z\mathcal{J}\|_F. \tag{155}$$

combining

$$(II) \leq 2\rho\|Z-\mathcal{J}Z\mathcal{J}\|_F + (1+2\delta_{2(r+1)})\hat{\Delta}_{2(r+1)}\|Z-\mathcal{J}Z\mathcal{J}\|_F^2 \tag{156}$$

$$+ \max_{v:\|v\|_2\leq 1}\left(\frac{\hat{\Delta}_{2(r+1)}}{2}\|vv^\top - \mathcal{J}vv^\top\mathcal{J}\|_F^2 + \frac{(1+2\delta_{2(r+1)})\hat{\Delta}_{2(r+1)}}{2}\|\mathcal{J}vv^\top\mathcal{J}\|_F^2\right). \tag{157}$$

Under the assumption that $\|Z - \mathcal{J}Z\mathcal{J}\|_F^2 \leq 1$ there holds

$$(II) \leq 2\rho + 2(1+2\delta_{2(r+1)})\hat{\Delta}_{2(r+1)}, \tag{158}$$

and using the linearity of the operator, there holds for all matrices fulfilling the constraints of the lemma that

$$(II) \leq 2\left(\rho + (1+2\delta_{2(r+1)})\hat{\Delta}_{2(r+1)}\right)\|Z-\mathcal{J}Z\mathcal{J}\|_F. \tag{159}$$

Gathering both results the desired results follows.

### G.3   Proof of Lemma 3

Denote by $Z = \sum_{i=1}^{md}\lambda_i v_i v_i^\top$

$$\|(\mathcal{W}Z\mathcal{W} - \mathcal{A}^*\mathcal{A})(Z)\|_F \leq \sum_{i=1}^{md}\lambda_i\|Wv_iv_i^\top W - \mathcal{A}^*\mathcal{A}(v_iv_i^\top)\|_F. \tag{160}$$

Invoking Lemma 1 there holds

$$\|(\mathcal{W}Z\mathcal{W} - \mathcal{A}^*\mathcal{A})(Z)\|_F \leq \tag{161}$$

$$\leq \sum_{i=1}^{md}\lambda_i\left(2\delta_4 + 2\left(\rho + (1+2\delta_4)\hat{\Delta}_4\right)\right) = \left(2\delta_4 + 2\left(\rho + (1+2\delta_4)\hat{\Delta}_4\right)\right)\|Z\|_*. \tag{162}$$

### G.4   Proof of Lemma 4

$$\|(\mathcal{I}-\mathcal{J})\mathcal{A}^*\mathcal{A}(Z^\star)\| = \|\mathcal{A}^*\mathcal{A}(Z^\star)(\mathcal{I}-\mathcal{J})\|, \tag{163}$$

and

$$\mathcal{A}^*\mathcal{A}(Z^\star) = \frac{1}{mn}\sum_{s=1}^N\left(w_{\mathcal{V}(s)}w_{\mathcal{V}(s)}^\top\right)\otimes A_s\langle A_s, \bar{Z}^\star\rangle \tag{164}$$

$$= \mathcal{W}\frac{1}{mn}\sum_{s=1}^N\left(e_{\mathcal{V}(s)}e_{\mathcal{V}(s)}^\top\right)\otimes A_s\langle A_s, \bar{Z}^\star\rangle\mathcal{W}. \tag{165}$$

Thus, because $\mathcal{W}$ is p.s.d. there holds

$$(\mathcal{I} - \mathcal{J})\mathcal{A}^*\mathcal{A}(Z^\star) = (\mathcal{W} - \mathcal{J})\frac{1}{mn}\sum_{s=1}^{N}\left(m^2 e_{\mathcal{V}(s)}e_{\mathcal{V}(s)}^\top\right) \otimes A_s\langle A_s, \bar{Z}^\star\rangle\mathcal{W} \tag{166}$$

$$= (\mathcal{W}^{1/2} - \mathcal{J})\frac{1}{mn}\sum_{s=1}^{m}\left(m^2 w_{\mathcal{V}(s)}^{1/2}\left(w_{\mathcal{V}(s)}^{1/2}\right)^\top\right) \otimes A_s\langle A_s, \bar{Z}^\star\rangle\mathcal{W}^{1/2}. \tag{167}$$

Consequently, we have

$$\|(\mathcal{I} - \mathcal{J})\mathcal{A}^*\mathcal{A}(Z^\star)\| \le \rho^{1/2}\left\|\frac{1}{mn}\sum_{s=1}^{m}\left(m^2 w_{\mathcal{V}(s)}^{1/2}\left(w_{\mathcal{V}(s)}^{1/2}\right)^\top\right) \otimes A_s\langle A_s, \bar{Z}^\star\rangle\right\| \tag{168}$$

Under the assumptions of the lemma we can invoke Lemma 2 yielding

$$\|(\mathcal{I} - \mathcal{J})\mathcal{A}^*\mathcal{A}(Z^\star)\| \le \rho^{1/2}\left(\|Z^\star\| + \delta_{2r^\star}\|Z^\star\|_F\right) \tag{169}$$

Following the same procedure

$$\|(\mathcal{I} - \mathcal{J})\mathcal{A}^*\mathcal{A}(Z^\star)(\mathcal{I} - \mathcal{J})\| \le \rho\left(\|Z^\star\| + \delta_{2r^\star}\|Z^\star\|_F\right). \tag{170}$$