# OpenReview forum: "Decentralized Matrix Sensing: Statistical Guarantees and Fast Convergence"
_NeurIPS.cc/2023/Conference — NeurIPS 2023 poster_

### Official Review · Reviewer_XCka · 2023-06-28

**Soundness:** 3 good
**Presentation:** 4 excellent
**Contribution:** 3 good
**Rating:** 6
**Confidence:** 4

**Summary:**

In this work, the authors proved the convergence of the distributed GD method on the over-parameterized matrix sensing problem. The convergence is based on the proposed in-network RIP condition. Numerical experiments are conducted to verify the theoretical findings.

**Strengths:**

The paper is well-written and easy to follow. In my opinion, the results are novel and should be interesting to audiences in optimization and machine learning fields.

**Weaknesses:**

Although I believe that the paper is theoretically sound, I feel that the proof techniques are mostly the same as [34], except the concept of in-network RIP condition. This may weaken the theoretical contribution of this work.

**Questions:**

(1) Equation (1): I think it should be trace(A_j * Z^*) instead of trace(A_j, Z^*) in the right term.

(2) Line 124: I think w_{ij} > 0 if and only if i and j can communicate?

(3)

(3) Theorem 1 and Remark on Line 333: it would be better if the authors could elaborate on the technical difficulty in proving the theorem, besides introducing the in-network RIP condition and modifying the proofs in [34].

(4) Corollary 1: the denominator in the left-hand side should be \|Z^*\| instead of \|Z^*\|^2?

(5) Line 355: A_i = (1/2) (S_i + S_i^T).

**Limitations:**

Please see my questions in the previous section.

---

> ### Author Rebuttal · Authors · 2023-08-07
>
> We acknowledge the query about the novelty of our convergence analysis relative to [34], providing an opportunity to highlight the unique features of our approach. While both methods share a two-phase structure, our proof contains technical details that substantially diverge from [34], a distinction recognized by the other Reviewers and elaborated below.
> - **Different algorithms and consequences**:  [34] employs the standard gradient algorithm,
> $$ \bar{U}^{t+1} = \Big(I + \alpha \bar{\mathcal{A}}^*\bar{\mathcal{A}}(\bar{Z}^* - \bar{U}^t(\bar U^t)^{\top})\Big)\bar{U}^t, \qquad (1)$$
> while the proposed scheme performs a *double* mixing update that intermingles iterates and local gradients:
> $$U^{t+1} = \Big(\mathcal{W}^2 + \frac{\alpha}{m}\mathcal{A}^*\mathcal{A}(Z^{\star} - U^t(U^t)\Big)U^t.	\qquad (2)$$
> The difference in the updates (despite the lifting of variables), in particular the dissimilarity of the maps  $\mathcal{A}$ and $\bar{\mathcal{A}}$,  has the following consequences on the analysis
>
>     -(i) *Lack of RIP*:  The proofs in [34] heavily rely on the RIP of $\bar{\mathcal{A}}$.   **Notably,  $\mathcal{A}$  does not in general fulfill the RIP**, except under an undesirable stringent condition on the agents' local sample size, which would break the *centralized* sample complexity achievable by our scheme following our analysis. This difference renders some steps of the proofs in [34] inapplicable to our algorithm and complicate the translation of analyses between the centralized and distributed contexts.
>
>     -(ii) *Which power-like  method in Phase I for (2)?*  Using the RIP of $\bar{\mathcal{A}}$, Phase I in [34] demonstrates that,  for small $t$,  $\bar{U}^{t+1}$ in (1) remains sufficiently close to the power method iterate $\bar{\mathcal{M}}^t U^0$, (i.e.,  $\||\bar{U}^{t+1}-\bar{\mathcal{M}}^t U^0\||=\mathcal O(\mu^3)$), which can be proved to be  'well-aligned' to the ground truth $\bar{Z}^\star$, where  $$\bar{\mathcal{M}}:={I} + \alpha \bar{\mathcal{A}}^*\bar{\mathcal{A}}(\bar{Z}^{\star}) \qquad (3)$$ and $\mu$ is the size of the initialization.
>
>     While the choice of  $\bar{\mathcal{M}}$ is straightforward for the gradient algorithm (1),  **the counterpart of  $\bar{\mathcal{M}}$ for the iterates (2)** (denoted hereafter by $\mathcal M$) **remains unclear**. An observation under the RIP of $\bar{\mathcal{A}}^*\bar{\mathcal{A}}(\bar{Z}^{\star})$ is that $\mathcal{J}\mathcal{A}^*\mathcal{A}(Z^{\star})\mathcal{J}$ fulfills the RIP as well. Consequently,  a potential choice of   $\mathcal M$  in line  with (3) that allows us to closely emulate the proof in [34] could be   $$\mathcal M=\mathcal{J} + \frac{\alpha}{m}\mathcal{J}\mathcal{A}^*\mathcal{A}(Z^{\star})\mathcal{J} \quad \text{or}\quad \mathcal M=\mathcal{W}^2 + \frac{\alpha}{m}\mathcal{J}\mathcal{A}^*\mathcal{A}(Z^{\star})\mathcal{J}, \quad (4) $$ enabling reliance on the RIP of the mappings appearing in such  $\mathcal M$'s.
>
>     Following the steps of [34] with either of these choices leads to an unsatisfactory outcome: the discrepancy of the trajectories,  $\||{U}^{t+1}-{\mathcal{M}}^t U^0\||$, becomes uncontrollable by the initialization size $\mu$ only (as instead in [34]), due to the network dependency of the mapping  $\mathcal{A}$. This breaks the condition in [34] to exit Phase I.
>
>     Therefore,  if one aims to avoid imposing a local RIP (thus a local sample size) condition, the proofs of [34] are not directly applicable. To circumvent this challenge, our key innovation lies in identifying the 'right' mapping $\mathcal M$,  expressed as $$\mathcal M=\mathcal{W}^2 + \frac{\alpha}{m}\mathcal{A}^*\mathcal{A}(Z^{\star}),\qquad (5)$$ which in conjunction with the *new concept*  of in-network RIP,  enables us to demonstrate that   $\||{U}^{t+1}-{\mathcal{M}}^t U^0\||$ is controllable solely by the initialization size  $\mu$    without necessitating the RIP of $\mathcal{A}^*\mathcal{A}(Z^{\star})$ in $\mathcal M$ (thereby avoiding constraints on the local sample size). In essence, the new RIP serves as the linchpin to manage the distortion that the mapping  $\mathcal M$ in (5) induces on  the eigenvectors and eigenvalues of $Z^\star$ via a condition on the network connectivity $\rho$ while  maintaining a decoupling from the size $\mu$ of the initialization. This subtle yet powerful modification illuminates a path forward, distinct from  [34], that captures the  decentralized nature of the algorithm.
>
>     -(iii) *Phase II: handling  extra error terms:* The aforementioned difference of the mappings,  along with the distributed nature of our algorithm, introduce additional complexity into the analysis of Phase II, due to extra error terms (such as those stemming from consensus errors) that are absent in [34]. These error components necessitate careful control and management, adding an additional layer of intricacy to our analysis
> - **In-network RIP**:We feel  that the new concept of in-network RIP holds more than mere technical significance; it represents an essential bridge between centralized and distributed worlds. It provides a tool for analyzing decentralized algorithms under *centralized* sample complexity, marking a considerable departure from [34]. Far from a minor contribution, it may pave the way for broader applications and insights in distributed computation
> - **On the final stage of convergence:** Our new analysis offers convergence assurances within a specific time **window** post a defined point in time. This is an enhancement over [34], whose proofs only ensure convergence at a particular instance and not all subsequent iterations. This expansion of the convergence window (consequence of our proof)  is sensitive in distributed settings, where synchronizing termination at a singular moment is not easily enforceable.
>
> Please let us know if our assessment of the technical novelty is satisfactory. Happy to provide more details
>
> We will also fix the typos raised in your questions. Thanks

---

> > ### Comment · Reviewer_XCka · 2023-08-16
> >
> > I would like to thank the authors for the detailed response! I am happy to increase my rating.

---

### Official Review · Reviewer_1WwW · 2023-07-02

**Soundness:** 3 good
**Presentation:** 3 good
**Contribution:** 2 fair
**Rating:** 6
**Confidence:** 4

**Summary:**

This paper proposes a decentralized gradient algorithm for the matrix sensing problem via Burer-Monteiro type decomposition. A new concept of RIP termed in-network RIP is introduced for the proposed algorithm, which harnesses the RIP of the measurement operator and intertwines it with the network's connectivity. The paper demonstrates the effectiveness of the proposed algorithm by providing numerical simulations.

**Strengths:**

- The paper provides a new decentralized gradient algorithm that solves the low-rank matrix sensing problem via Burer-Monteiro type decomposition. The study covers various aspects of the algorithm, including statistical guarantees, communication complexity, and sample complexity.

- The paper introduces a new concept of in-network RIP, which harnesses the RIP of the measurement operator and intertwines it with the network's connectivity to derive favorable attributes of the new, overarching network-wide measurement operator. This concept provides a new perspective on the analysis of decentralized matrix sensing.



**Weaknesses:**

- Lack of decentralized applications. The paper only provides some applications for centralized matrix sensing problems and presents the shortcomings of common centralized applications. It would be helpful to list some practical applications for decentralized matrix sensing problems to motivate the proposed algorithms.

- The article is innovative but not too big. The algorithm is a simple combination of [25] and [45], but it's not straightforward to extend the analysis in [34] from the centralized case to the decentralized case. In-network RIP is proposed to close the gap.

- The condition of $\rho$ in (12) seems too strict. How many communication rounds per iteration are needed to satisfy (12)?  It would be helpful to present the number of communication rounds per iteration in numerical experiments.

**Questions:**

Please explain the concept "generalization error". As far as I know, generalization error is a concept in machine learning that measures the performance of a trained model on new, unseen data. What does it mean in this paper?

---

> ### Author Rebuttal · Authors · 2023-08-08
>
> We thank the Reviewer for their insightful comments  and the overall positive assessment of our work. Our response to her/his comments/questions follows, starting from those listed under the `Weaknesses'.
>
> -$\textbf{1.}$ Solving problems in a decentralized fashion is essential in situations in which high dimensionality data is gathered in multiple locations. In these situations gathering all data in a single machine can be prohibitive. However, with only partial access to the data the problem may not be solvable. A feasible approach in this circumstance is to opt for a master-client architecture (star topology with a server node connected to all the clients). However, this introduces a single point of failure and the server can easily become a communication bottleneck. Thus, in circumstances in which high-dimensional data is gathered in multiple/many locations, in terms of robustness and efficiency, decentralized computation over mesh networks is the preferred option, which has become a popular choice in several ML applications and computational architectures. Concrete matrix factorization/sensing applications in which high dimensional data is gathered in multiple locations naturally include the Netflix problem [Koren, Bell, Volinski 2019], for which schemes working on the master-client architecture have been proposed [Teflioudi, Makari, Gemulla, 2012] , and Seismic data interpolation [Aravkin, Kumar, Mansour, et. al. 2014].  Observe that in these applications data is both high dimensional and gathered in a decentralized fashion.
>
>
>
> -$\textbf{2.}$ We agree with the Reviewer that the proposed scheme builds on [25]. However, it is important to remark that the algorithm [25]  has been designed to deal with **convex** problem. A direct application of the same approach here would have suggested the algorithm in [25] applied to a  **convex formulation** of the matrix sensing problem  over the network (e.g., minimizing the nuclear norm subject to  linear constraint, e.g.,  [3]). However, by doing so, the resulting distributed algorithm would have incurred in the unaffordable communication cost of  $\mathcal{O}(d^2)$ elements per iteration. This motivates the application of a double-mixing based  decentralized algorithm to the distributed **non-convex** problem in the form  (2), for which there was no study in the literature. We refer the Reviewer to the reply to Reviewer XCka for  details also on the *technical* novelties of our convergence analysis.
>
> Referring to [45], the authors therein established that for unconstrained problems the algorithm in [25] and a variant of decentralized gradient descent are equivalent. Because the matrix sensing problem is indeed unconstrained, our proposed scheme can be cast into a variant of decentralized gradient descent (up to a change of variable), which we mentioned in our manuscript. However, since the analysis in [45] does not help to  establish  convergence guarantees in our setting, we believe that the aforementioned change of variable is not convenient for our purposed.
>
> More generally, we believe that for the matrix sensing  problem proposed algorithm is most suitable among other distributed algorithms in the literature applicable to nonconvex problems. This is for two reasons, (i) contrary to the problems handled in [25] and [45] $\bar{Z}^{\star}$ is a solution to the global problem and to each individual problem, which removes the need of more involved updates, like incorporating correction mechanisms of the local gradients (such as gradient tracking mechanisms), even if arbitrarily close to 0 precision guarantees are requested; (ii) gradient tracking or other correction mechanisms would complicate the analysis further, and as argued, to no benefit.
>
> -$\textbf{3.}$
> In order to obtain the number of communication rounds required to satisfy (12) with a mixing matrix/graph with a given connectivity $\bar{\rho}$ we require $K$ communication rounds where $K$ is such that
> $$
>     \bar{\rho}^K \leq C \frac{\delta}{m^6 \kappa^4 r^{\star}}
> $$
> which is fulfilled by setting for some universal constant $c_0 > 0$
> $$
>     K = \left\lceil \frac{c_0 \log\left( m \kappa r^{\star}/\delta \right)}{1-\bar{\rho}} \right\rceil.
> $$
> Observe that the dependence on the number of communication rounds is logarithmic with respect to $m$ $\kappa$ and $r^{\star},$ and consequently, scales beneficially with quantities that are moderate in size and is independent of potentially large quantities such as $d.$
>
> $\textbf{Questions:}$ We agree with the reviewer that the terminology 'generalization error' is typically used in the ML community to evaluate the performance of a trained model (typically trained on the empirical risk) $U^t$ in terms of the population risk value at $U^t$. We referred to the quantity in Theorem 1 as generalization error because when particularized to the data model discussed in lines 193-197 the quantity on the RHS of (15) corresponds to the population risk value for the distributed problem.
>
> We understand that this terminology can be misleading and we will change it in the final version of the paper.
>
> Please let us know if further clarifications are needed.

---

> > ### Comment · Reviewer_1WwW · 2023-08-16
> > **After review**
> >
> > I appreciate the response, most of my concerns have been addressed. I have already scored based on this article's potential, so the score remains unchanged.

---

### Official Review · Reviewer_qpo5 · 2023-07-08

**Soundness:** 3 good
**Presentation:** 4 excellent
**Contribution:** 4 excellent
**Rating:** 7
**Confidence:** 3

**Summary:**

This paper studies the problem of decentralized low-rank matrix sensing.  The paper presents novel theoretical results on the convergence and generalization of a standard decentralized learning algorithm. In particular, they provide convergence and generalization guarantees for  decentralized gradient descent algorithm over mesh networks without any central server. To do so, the authors define a new "in-network" RIP, which serves as an equivalent quantity to the standard RIP used to establish similar results in the fully centralized setting. This quantity is shown to capture both the network connectivity and the measurement operator characteristics. Using this quantity, they establish an upper bound on the number of communication iterations and the resulting estimation error. The paper also provides simulation results that demonstrate the recoverability of the ground truth matrices using the algorithm.

**Strengths:**

1. The paper is very well-written and is easy to follow despite being dense.

2. The formulation of the in-network RIP is interesting and captures the complexity of the problem well.

3. The paper makes solid technical contributions to an interesting problem.

4. The interpretation of the theoretical results is much appreciated and helps the readability of the paper.

**Weaknesses:**

1. The figures in the experimental section could be improved in terms of presentation. Particularly, please enlarge the figure font sizes.

2. The authors could provide a refinement of equation 15 in terms of the sample complexity required to achieve epsilon relative error. The current error bound makes it hard to get a sense of how many measurements are required to get a desired error, when all other parameters are fixed.

3. In lines 95-98, the authors mention that distributed spectral methods cannot guaranty exact recovery and seem to indicate that the current algorithm can. However, it is not clear from the theory or the experiments if exact recovery is possible in the noiseless case even when the required RIP properties are obeyed. Can the authors provide further clarification on this?


**Questions:**

Please see the weaknesses section.

**Limitations:**

N/A.

---

> ### Author Rebuttal · Authors · 2023-08-08
>
> We thank the Reviewer for their insightful comments and the overall positive assessment of our work. We are happy the Reviewer liked it. We answer the Reviewer's concerns in order.
>
> -$\textbf{1.}$  We will enlarge the font of the text in the figures and re scale them for visibility. We can change the color choices/grid choices to increase the figure visibility.
>
> -$\textbf{2:}$  We agree that the result can be alternatively stated in terms of iterations towards an $\varepsilon>0$ accuracy. We opted for the format of presentation of the convergence of the same type as in [34] to facilitate the comparison, showing that our distributed algorithm  achieves centralized statistical guarantees.  Addressing the Reviewer's question, an $\varepsilon>0$ solution,   $$ \frac{ \||U^{\hat{t}}(U^{\hat{t}})^\top-Z^{\star}\||_F}{\||Z^{\star}\||}\leq \varepsilon,$$ is achieved by  choosing the size of the initialization $\mu$ as
>
> $$\mu^2 \precsim \min \left\\{\frac{\varepsilon^2 r^2}{\kappa^4 d^2 (r - r^{\star})^2(r^{\star})^{4/21}} \\|\bar{X}\\|^2, \frac{\sqrt{rm}}{d \sqrt{d}\kappa^9} , \frac{\sqrt{r}}{d \sqrt{d}}\left(\kappa^2 \sqrt{\frac{d}{r}} \right)^{-96\kappa^2} \right\\} \quad \text{(A)}$$
>
> Notice that this result holds for **any** sample size  $N \geq c_1 d (r^{\star})\kappa^8$, which is **independent** on $\varepsilon$, This condition is required to guarantee the RIP condition (with   $\delta \precsim \kappa^{-4}(r^{\star})^{-1/2}$).
>
> To summarize, the condition on the sample size $N$ does not depend on the precision of the estimates the algorithm can achieve, which   instead is controlled via the initialization size $\mu$ (affecting thus the convergence rate).
>
> -$\textbf{3: }$ Contrary to our algorithm that achieves arbitrarily small  $\varepsilon>0$ precision, under the $\delta$-RIP, for **any**   $\delta \precsim \kappa^{-4}(r^{\star})^{-1/2}$ (resulting in a **fixed**  $N \geq C_1 d (r^{\star})\kappa^8$ for **any** $\varepsilon>0$),  this is not the case if spectral methods are employed to solve the matrix sensing problem.  Such methods  would provably estimate  the eigenvalues and eigenvectors of the matrix $\mathcal{J}\mathcal{A}^*\mathcal{A}(Z^{\star})\mathcal{J}$ with arbitrary precision, which however do not match the eigenvalues and eigenvectors  of $Z^{\star}$ with arbitrarily precision, unless $\delta$  becomes arbitrarily small (which cannot be for fixed $N$). More precisely, assuming the centralized mapping $\bar{\mathcal{A}}$ fulfills the $\delta$-RIP, with fixed $\delta\neq 0$ (thus fixed, proper $N$), the estimate  $Y$ produced by such procedures  satisfies  [Chi, Lu, Chen 2019]
> $$\\|Y-\bar{Z}^{\star}\\| \leq 	\delta \sqrt{r^{\star}}\\|\bar{Z}^{\star}\\|.$$ Clearly, for a given  $\delta\neq 0$, the right hand side cannot be made arbitrarily small. As discussed in the comment 2 above, this is not the case for our algorithm.
>
> Please let us know if further clarification are needed. We will update the paper to point out the above aspects.

---

> > ### Comment · Reviewer_qpo5 · 2023-08-18
> >
> > Thanks for the clarifications, I appreciate it. I do not have any further comments.

---

### Author Rebuttal · Authors · 2023-08-07

We thank the Reviewers for their careful reading and insightful comments. We are glad that there is a consensus that the paper is well written and contains novel results that are of interest   to audiences in optimization and machine learning. Our reply to their specific questions and key comments follow.  In our replies we use the same notation as in the paper.

---

### Decision · Program_Chairs · 2023-09-21

**Decision:**

Accept (poster)

**Comment:**

This paper offers a first analysis for solving low-rank matrix sensing in a decentralized network, providing a comprehensive statistical, computational and communication complexity analysis. In particular, it bypasses several challenges, including but not limited to how to characterize the algorithm dynamics from small random initializations.